# Multi-omics analysis of green lineage osmotic stress pathways unveils crucial roles of different cellular compartments

Josep Vilarrasa-Blasi [1,2] ✉, Tamara Vellosillo[1,2], Robert E. Jinkerson [2,3], Friedrich Fauser[2,4], Tingting Xiang[2,5], Benjamin B. Minkoff[6], Lianyong Wang[4], Kiril Kniazev[1], Michael Guzman[2], Jacqueline Osaki[2], Gregory A. Barrett-Wilt[7], Michael R. Sussman [6], Martin C. Jonikas [2,4] & José R. Dinneny [1,2] ✉

Maintenance of water homeostasis is a fundamental cellular process required by all living organisms. Here, we use the single-celled green alga *Chlamydomonas reinhardtii* to establish a foundational understanding of osmotic-stress signaling pathways through transcriptomics, phosphoproteomics, and functional genomics approaches. Comparison of pathways identified through these analyses with yeast and Arabidopsis allows us to infer their evolutionary conservation and divergence across these lineages. 76 genes, acting across diverse cellular compartments, were found to be important for osmotic-stress tolerance in Chlamydomonas through their functions in cytoskeletal organization, potassium transport, vesicle trafficking, mitogen-activated protein kinase and chloroplast signaling. We show that homologs for five of these genes have conserved functions in stress tolerance in Arabidopsis and reveal a novel PROFILIN-dependent stage of acclimation affecting the actin cytoskeleton that ensures tissue integrity upon osmotic stress. This study highlights the conservation of the stress response in algae and land plants, and establishes Chlamydomonas as a unicellular plant model system to dissect the osmotic stress signaling pathway.

Maintaining cellular water homeostasis is vital for all living organisms and requires mechanisms to counteract differences in water availability between the intracellular and extracellular environment occurring during fluctuating environmental conditions[1]. The increasing need to provide sustainable food sources for a growing population, combined with the effects of climate change, urges efforts to improve our understanding of such mechanisms to support the development of crops more resilient to water scarcity[2].

Simplified model systems have been used to gain mechanistic insight for molecular functions associated with acclimation to osmotic stress[3], which can later be characterized in more complex organisms[4] to reveal the evolutionary relationships of such pathways[5]. A prominent example is the HIGH OSMOLARITY GLYCEROL (HOG) signaling pathway, responsible for the response to high osmolarity, initially uncovered in yeast and later shown to be conserved in a variety of more complex eukaryotes[1]. The HOG pathway has established a

[1]Department of Biology, Stanford University, Stanford, CA 94305, USA. [2]Carnegie Institution for Science, Department of Plant Biology, Stanford, CA 94305, USA. [3]Department of Chemical and Environmental Engineering, University of California Riverside, Riverside, CA 92521, USA. [4]Department of Molecular Biology, Princeton University, Princeton, NJ 08544, USA. [5]Department of Biological Sciences, University of North Carolina at Charlotte, Charlotte, NC 28223, USA. [6]Department of Biochemistry and Center for Genomics Science Innovation, University of Wisconsin, Madison, WI 53706, USA. [7]Biotechnology Center, University of Wisconsin, Madison, WI 53706, USA. ✉e-mail: pituvilarnadal@gmail.com; dinneny@stanford.edu

paradigm for environmental signal transduction in eukaryotes, with cell membrane osmosensory receptors that convey the signal through a MAPK kinase (HOG) cascade. Downstream of perception, molecular programs required for cellular acclimation to hyperosmotic stress are orchestrated including temporary cell cycle arrest, transcription and translational remodeling, transport and synthesis of compatible osmolytes, among others[1]. Despite the high degree of conservation across many eukaryotes, photosynthetic organisms do not have a canonical HOG pathway[6].

Land plants are multicellular organisms with organs below and above the soil, with different sensitivities to water-fluctuating environments, and composed of several cell types acting differently in response to osmotic stress[7]. Knowledge of osmotic signaling pathways and growth adaptations in photosynthetic organisms mostly comes from plant model species such as Arabidopsis thaliana, where decades of research have yielded knowledge of specific signaling components and growth adaptations[8,9]. The emerging picture is a complex molecular suite of pathways initiated through hormone-dependent and independent pathways with cell type specificity[8]. Despite this, we are still lacking a systematic overview of molecular components associated with osmotic stress, at the cellular level, in any photosynthetic organism.

We sought to advance our understanding of osmotic signaling pathways by establishing the unicellular alga Chlamydomonas (*Chlamydomonas reinhardtii*) as a system that allows for rapid genetic and molecular investigation of the process. During the dominant phase of the lifecycle, Chlamydomonas exists as a haploid single cell, which facilitates the characterization of genetic mutants[10]. Recent large-scale characterization of mutant collections for growth defects under diverse stresses have demonstrated the importance of genes with homologs in animal and land-plant lineages[11]. Being a freshwater alga, structural and physiological differences with land plants also exist, such as the lack of a rigid cell wall to counteract cellular turgor generated by intracellular solutes[12], or the presence of an ancient contractile vacuole used to expel excess water out of the cell[13]. Furthermore, while initial responses to osmotic stress in land plants involve $Ca^{2+}$-mediated signaling[14], Chlamydomonas lacks several $Ca^{2+}$ channels present in algae and animal cells such as Voltage-dependent $Ca^{2+}$ channels, the transient receptor potential channels and the inositol triphosphate receptor, which may be important in mediating physiological responses to the environment[15–17].

Based on our understanding in other systems, we hypothesize that the regulation of osmotic stress tolerance occurs across different cellular and molecular pathways and requires a multifaceted approach to develop a comprehensive understanding of the system. Thus, we applied multi-omics profiling to dissect the different layers of the osmotic response in algae and compared it with the land plant, *Arabidopsis thaliana*, and fungi, *Saccharomyces cerevisiae*, to identify genetic components that may have conserved functions. Our transcriptomic analysis of osmotic stress responses in Chlamydomonas revealed similarities with plants in the regulation of cell cycle activity, but without a canonical abscisic acid (ABA) hormonal response common to land plants but with algal-specific responses, such as regulation of flagella structure. Using high-throughput genetic screens of barcoded mutants, we identified a gene network necessary for the survival and growth of Chlamydomonas under different osmotic conditions, which highlighted the different strategies used to cope with ionic and non-ionic osmoticum. We present various instances that support earlier hypotheses derived from the phylogenetic analysis within the green lineage, involving the need for transporters that were lost during the transition to land, rendering terrestrial plants more vulnerable to ionic stress when compared to freshwater and marine algae. To explore whether shared genetic mechanisms operate across Chlamydomonas and land plants, we characterized T-DNA mutants in orthologous Arabidopsis genes and uncovered 5 conserved cellular processes including; actin dynamics, potassium transport, palmitoylation, mitogen-activated protein kinase signaling and chloroplast signaling. Though the examination of developmental phenotypes in the roots of our Arabidopsis mutants, we revealed different spatio-temporal patterns of tissue damage occurring after osmotic stress, suggesting that the identified biological processes have different spatial and temporal contributions to acclimation. Further characterization of PROFILIN5, an actin binding protein, uncovered a major role for the regulation of actin cytoskeleton dynamics upon osmotic stress. Our work highlights the functional conservation of osmotic signaling pathways across kingdoms and provides tools to further explore the cellular strategies organisms use to survive water scarcity.

## Results

### A temporal map for the Chlamydomonas transcriptional response to osmotic stress

To understand the temporal progression of pathways regulated during acclimation to osmotic stress, we established a time-course transcriptional map in Chlamydomonas using the wild-type strain CC-4533 using two osmotic challenges, NaCl and mannitol. We used treatments that substantially reduce cell growth but are non-lethal (100 mM NaCl and 300 mM Mannitol, Supplementary Fig. 1a, b). We identified a total of 1,456 differentially regulated genes (FC > 2, FDR < 0.01), with a peak of regulation after 1 h (Supplementary Fig. 1c, d, Supplementary Data 1, 2). Up-regulated genes were enriched (FDR < 0.05) in annotations associated with response to cellular stimuli, vesicle-mediated-transport, and amyloplast organization, while downregulated genes were associated with the cell cycle, translation, chloroplast fission, and deoxyribonucleotide metabolism (Fig. 1a). Consistent with these data, we found that an osmotic challenge induced an increase in the number of vesicles and starch puncta in Chlamydomonas cells (Fig. 1b, Supplementary Fig. 2a) based on the use of fluorescent reporters to mark these compartments[18] and starch staining[19]. In agreement with previous transcriptomic analyses using different Chlamydomonas strains (GY-D55 and CC-503) and solute concentrations[20,21], we found that Chlamydomonas accumulates glycerol and proline as the main compatible osmolytes during stress. Interestingly, our time-course data suggests an earlier role for glycerol 15–60 min after stress, while proline may have a later role, since induction of proline metabolic gene transcription is not observed until 6 h after stress is applied (Fig. 1a, Supplementary Fig. 2b).

The gene regulatory patterns observed suggest that osmotic stress induces a pause in general cellular functions associated with growth, a pattern also observed in Saccharomyces and Arabidopsis upon osmotic stress[7,22]. The Chlamydomonas genome includes homologous gene families found in both Saccharomyces, and Arabidopsis[10]. Chlamydomonas osmotic stress-responsive genes have homologs in both Arabidopsis and Saccharomyces, although the similarity of the osmotic stress response is greater with Arabidopsis (24% of genes overlap in their osmotic stress response compared with 6% in Saccharomyces) (Material and methods, Supplementary Fig. 3, Supplementary Data 2).

While the abscisic acid (ABA) hormone signaling pathway mediates much of the transcriptional response downstream of osmotic stress perception in Arabidopsis[23], and exogenous application of ABA has been shown to increase tolerance to salinity in Chlamydomonas[24,25], we did not find enrichment of gene annotations associated with ABA-regulated genes nor related cis-regulatory elements (e.g., ABA RESPONSE ELEMENT or G-box) in the promoters of genes regulated by stress (Supplementary Fig. 1e, f, Supplementary Data 2)[26]. We hypothesize that the use of high concentrations of ABA in previous studies[24,25], ranging from 50 to 500 micromolar ABA, which are significantly higher than the nanomolar range of receptor affinity[27], may induce non-physiological responses in Chlamydomonas. This, in concordance with the lack of orthologous genes to the PYL/PYR/RCAR

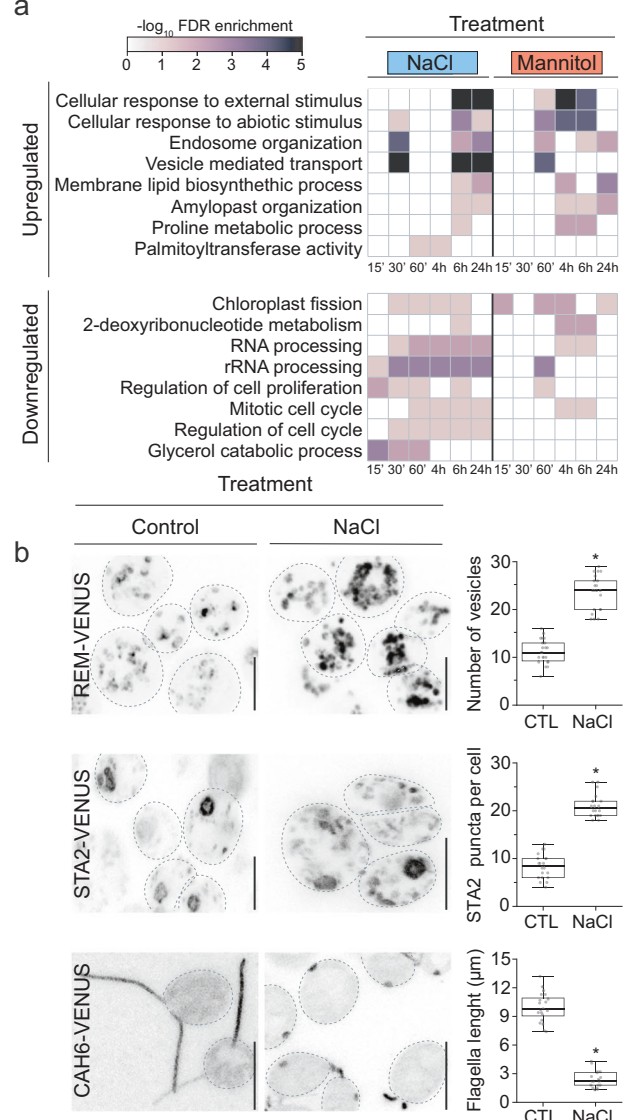

**Fig. 1 | *Chlamydomonas reinhardtii* transcriptional response to osmotic stress.**
**a** Gene ontology term analysis of the transcriptional response of Chlamydomonas cells treated with 100 mM NaCl or 300 mM mannitol at the indicated time points ('and h indicate minutes and hours respectively). See also Supplementary Figs. 1–3 and Supplementary Data 1, 2. **b** Confocal images of fluorescently tagged Chlamydomonas cells with different fluorescent markers under control conditions and treatment with 100 mM NaCl for 30 h. Cellular markers used: Endosomes, Cre16.g661700: REMORIN, labels the Golgi apparatus and secretory pathway (REM:VENUS); Starch, Cre17.g721500: STARCH SYNTHASE 2 (STA2), labels pyrenoid starch and starch granules (STA2:VENUS); flagella, Cre12.g485050: CARBONIC ANHYDRASE 6 (CAH6) labels flagella (CAH6:VENUS). (Right) Quantification of the number of vesicles in REMORIN cells, STA2 puncta and flagella length in control and cells treated with NaCl. $n = 20$ cells per condition. Cell images were taken in two independent experiments. Scale bars = 10 μm. Center lines show medians and box limits indicate the 25th and 75th percentiles. Whiskers represent minima and maxima, *P* values are from a one-way ANOVA test between control condition and treatments, $p < 0.05$. See also Source Data.

ABA receptors identified in land plants[10,28] suggests that canonical ABA-mediated osmotic response is not a conserved feature between Chlamydomonas and Arabidopsis.

A cellular compartment of Chlamydomonas cells that is shared with animals and certain plant lineages is the formation of motile cilia. Within land plants, cilia are exclusively produced in sperm cells of all nonseed plants, while the vast majority of seed plants are non-

ciliated[29]. We found an enrichment of genes encoding ciliary proteins[30] being differentially expressed in Chlamydomonas under osmotic stress (Fisher's Exact Test<1E-05), as well as a substantial reduction in flagellar length upon osmotic stress (Fig. 1b, Supplementary Data 2), in agreement with previous reports[31,32]. Together, these data demonstrate that osmotic stress induces rapid changes in the Chlamydomonas transcriptional landscape that are likely under the control of signaling pathways distinct from the dominant ABA-dependent pathway of land plants.

## Phosphoproteomic characterization of the early osmotic stress response

To uncover early signaling components of the Chlamydomonas osmotic stress response pathway, we adapted untargeted high-resolution mass spectrometry in conjunction with metabolic labeling to examine the effect of 5 min NaCl or mannitol treatment (Fig. 2, Supplementary Data 3, 4). We identified 33 Chlamydomonas differentially phosphorylated proteins (FC > 2, FDR < 0.01) encompassing a variety of functions including transcriptional regulation (MYB and bZIP family proteins), flagellar machinery (Flagella-associated proteins (FAP), FAP 69 and FAP 88), among others. Interestingly a large number of proteins (23/33) are dephosphorylated upon early osmotic stress suggesting a predominant role for protein phosphatases in the initial responses to osmotic stress in Chlamydomonas. We found several proteins with homologs (homology based on Reciprocal Best BLAST hit [RBH], see material and methods), that when mutated in Arabidopsis, have osmotic sensitive phenotypes such as; Ubiquitin protease UBIQUITIN-SPECIFIC PROTEASE 16 (UBP16)[33] and ENHANCED EM LEVEL (EEL), a bZIP transcription factor homologous to ABA-INSENSITIVE 5 (ABI5)[34]. Similarly, we identified several proteins with Saccharomyces homologs previously shown to be involved in osmotic stress, including; ribosomal stalk protein RIBONUCLEASE P (RPP1A)[35], VESICLE-ASSOCIATED MEMBRANE PROTEIN-ASSOCIATED PROTEIN (SCS2)[36], and INOSITOL POLYPHOSPHATE 5-PHOSPHATASE (IP 5-P), a phosphoinositol binding protein involved in cytoskeletal reorganization[37]. Among the Saccharomyces orthologous proteins, 8 out of 14 were shown to be phosphorylated upon osmotic stress, while only 3 out of 16 Arabidopsis orthologues showed phosphorylation, based on previous phosphoproteomic characterization in the respective species (Fig. 2, Supplementary Data 4). Using a less stringent significance threshold (FC > 1.5, FDR < 0.01), we identified 56 proteins with orthologs that are also phosphorylated in Arabidopsis (Supplementary Fig. 4, Supplementary Data 4). Our dataset includes 11 unannotated proteins which we renamed PHOSPHORYLATED UPON OSMOTIC STRESS IN CHLAMYDOMONAS (POC).

Overall, we found limited overlap between the candidates identified in our transcriptomic and phosphoproteomic analysis with only 5 genes/proteins shared between the two data sets. This list included a MYB transcription factor, IP 5-P, NAP57, and 2 Chlamydomonas-specific proteins (Fig. 2, Supplementary Data 4). Similarly limited overlap between proteomic and transcriptomic data sets examining the same process has been previously shown in Arabidopsis[38] and Saccharomyces[39] and is not entirely unexpected considering the different stress durations used with each assay. Nevertheless, our transcriptional and phosphoproteomic analysis has established the immediate targets of osmotic stress signaling in Chlamydomonas and highlights components most likely to be conserved across the green lineage and with non-photosynthetic eukaryotes.

## High-throughput genetic screens identify novel genes with roles in osmotic stress

Microbial model systems provide advantages for genome-wide, forward genetic screens due to the ease of culturing thousands of genotypes simultaneously. We utilized a recently developed barcoded genome-wide mutant library[40] and subjected Chlamydomonas cells to

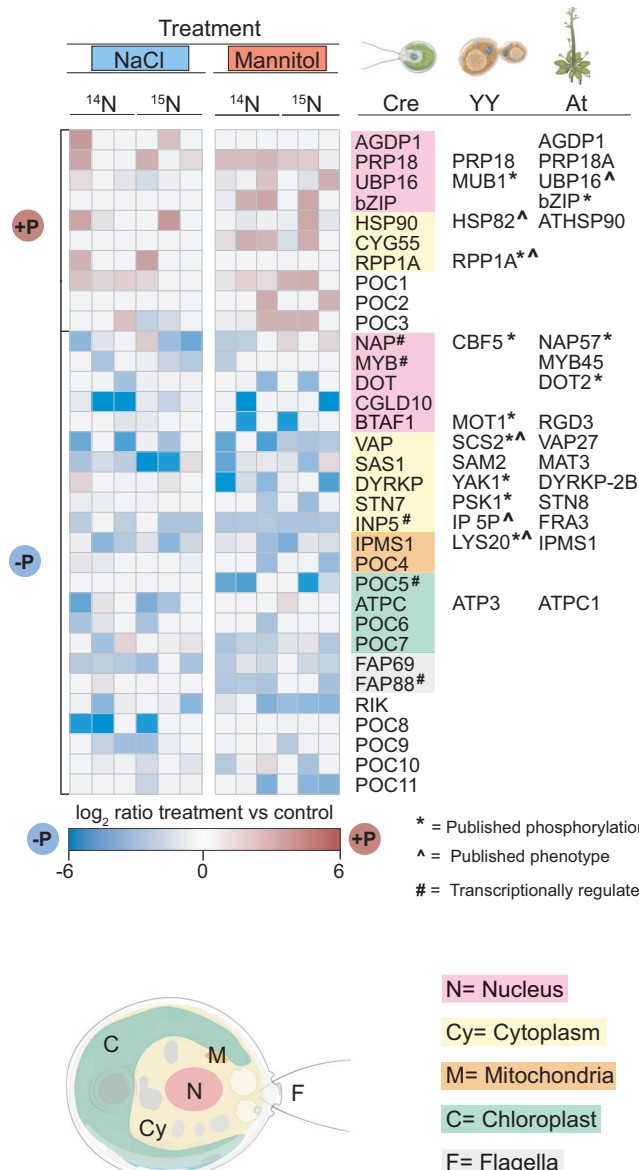

**Fig. 2 | Chlamydomonas early phosphorylation events upon osmotic stress.**
Heat map representing proteins differentially phosphorylated (FC > 2, FDR < 0.01) upon 5 min treatment with 100 mM NaCl or 300 mM mannitol. Color code represents protein localization according to PredAlgo[114] or associated functions. * Represents published phosphorylated proteins upon osmotic stress, ^ represents proteins with a described role in osmotic pathways. # Represents genes regulated in our Chlamydomonas osmotic treatments. See also Supplementary Data 3, 4 and Source Data.

such as DNA damage, chemical stress, photosynthesis, among others (Supplementary Data 6). These data suggest that some of the identified genes may play a role in multiple stress pathways with important roles in osmotic stress and/or other biological processes such as DNA damage, protein biosynthesis or photosynthesis, among others (Supplementary Data 6).

We validated our mutant phenotypes by performing secondary screens of 140 mutants selected according to different statistical criteria (Supplementary Data 7, Supplementary Fig. 5), confirming 55% of the hits. We identified genes encoding: 9 transporters, 4 flagella-associated proteins, 7 kinases, 2 phosphatases, 11 nuclear-localized proteins, among others. From this list of high confidence hits we identified 34 and 53 Saccharomyces and Arabidopsis orthologs, respectively (Fig. 3a) (FDR < 0.3, Supplementary Data 5). Interestingly, 24 genes did not have any previous annotation in Chlamydomonas, Arabidopsis, or Saccharomyces, and therefore we renamed them _OSMOTIC GROWTH DEFECTIVE IN CHLAMYDOMONAS (OSMO)_. Among our top hits, we identified _HYPEROSMOLALITY-INDUCED CALCIUM INCREASE 1 (CreOSCA1)_ and _MscS-LIKE (CreMSL)_ that mediate the initial events of osmotic stress signaling in Arabidopsis[42,43]. Mutants of both genes showed sensitivity to NaCl and mannitol, acclimation to osmotic stress and flagella defects, and for CreMSL, a cellular localization comparable to the Arabidopsis homolog, as previously reported (Supplementary Fig. 6)[42,44]. Below, we describe several examples exploring our functional genomics results by cellular functions.

Eukaryotic cells use two systems to actively extrude intracellular toxic sodium: Cation/H⁺ antiporters[45] (CPAs) and Na⁺-P-ATPases[46]. Our screens uncovered sodium chloride sensitivity to; (i) a conserved CPA transporter SODIUM/HYDROGEN EXCHANGER (NHX, Cre01.g034150) previously described in plants[47] and fungi[48] to play a role in sodium toxicity; and (ii) SODIUM/POTASSIUM-EXCHANGING ATPase (CrNAK1) (Cre06.g263950)[49], a P2C Na+/K+-ATPase lost at the base of the land plant lineage. Only two chlorophytes retain NAK1 transporters, a marine algae (_Ostreococcus tauri_) and Chlamydomonas (freshwater), suggesting that the high sensitivity of land plants to Na⁺ arose from the loss of an effective system to remove intracellular Na⁺[49]. Our screens have osmolyte specificity as shown by the number of membrane transporters necessary for Chlamydomonas growth under NaCl, since these transporters were not identified in the other conditions screened.

A central role for protein kinases and phosphatases in response to high osmolarity have been shown in fungi, mammals and plants[1,50]. We identified sodium chloride sensitive phenotypes for signaling components such as kinases: CALCIUM-DEPENDENT PROTEIN KINASE/ MICROTUBLE ASSOCIATE PROTEIN 2 KINASE (CDPK, Cre03.g153150) and MITOGEN-ACTIVATED PROTEIN KINASE KINASE KINASE (MAPKKK, Cre13.g576600), and phosphatase: PROTEIN PHOSPHATASE 2C (PP2C, Cre03.g211073) (Fig. 3a, Supplementary Data 6). Kinase orthologs in Arabidopsis (CDPK4, At4g09570; M3KDELTA 7, At1g18160) are required for the activation of ABA and osmotic stress signaling in plants[51,52], while Saccharomyces orthologs (SKM1, YOL113W; CMK1, YFR014C) are necessary for the activation of the response to hyperosmotic stress (HOG pathway) in yeast[1]. Similarly, PP2C orthologs (ABI, At4g26080; PTC1, YDL006W) are negative regulators of (i) ABA signaling in plants[53] and (ii) the HOG pathways in yeast[54]. In addition, we identified 3 kinases involved in non-ionic osmotic stress including (i) a DUAL-SPECIFICITY TYROSINE REGULATED PROTEIN KINASE (DYRKP, Cre07.g337300) present in our high confidence list of differentially phosphorylated proteins (Fig. 2a) and sensitive to hyperosmotic stress, (ii) OSMO17 a protein containing a POLO-DOMAIN KINASE (OSMO17, Cre03.g190050) involved in the cell cycle[55] and sensitive to mannitol treatment, and (iii) OSMO22 an uncharacterized Chlamydomonas specific protein with a protein kinase domain and sensitive to hypoosmotic stress.

four different osmotic perturbations: NaCl, mannitol, Polyethylene glycol (PEG), and hypo-osmotic stresses. After analysis of the data using a previously defined stringent confidence threshold[11] we identified 76 candidate genes/hits with associated mutants that exhibit a growth defect in at least one stress condition: 27 for NaCl, 34 for mannitol, 13 for hypoosmotic, 2 for PEG (FDR < 0.3; Fig. 3a, b, Supplementary Data 5, 6, see Material and methods). While several replicates were performed for each condition, only 10 significant genes were consistently identified across replicates, suggesting that the screens had not been performed to saturation (Fig. 3b, Supplementary Data 6). Comparison of our osmotic-related phenotypes with a larger survey for phenotypes using 121 conditions using the same mutant library and analysis pipeline[41] revealed that 34 out of 76 high confidence hits showed a growth phenotype in another stress condition

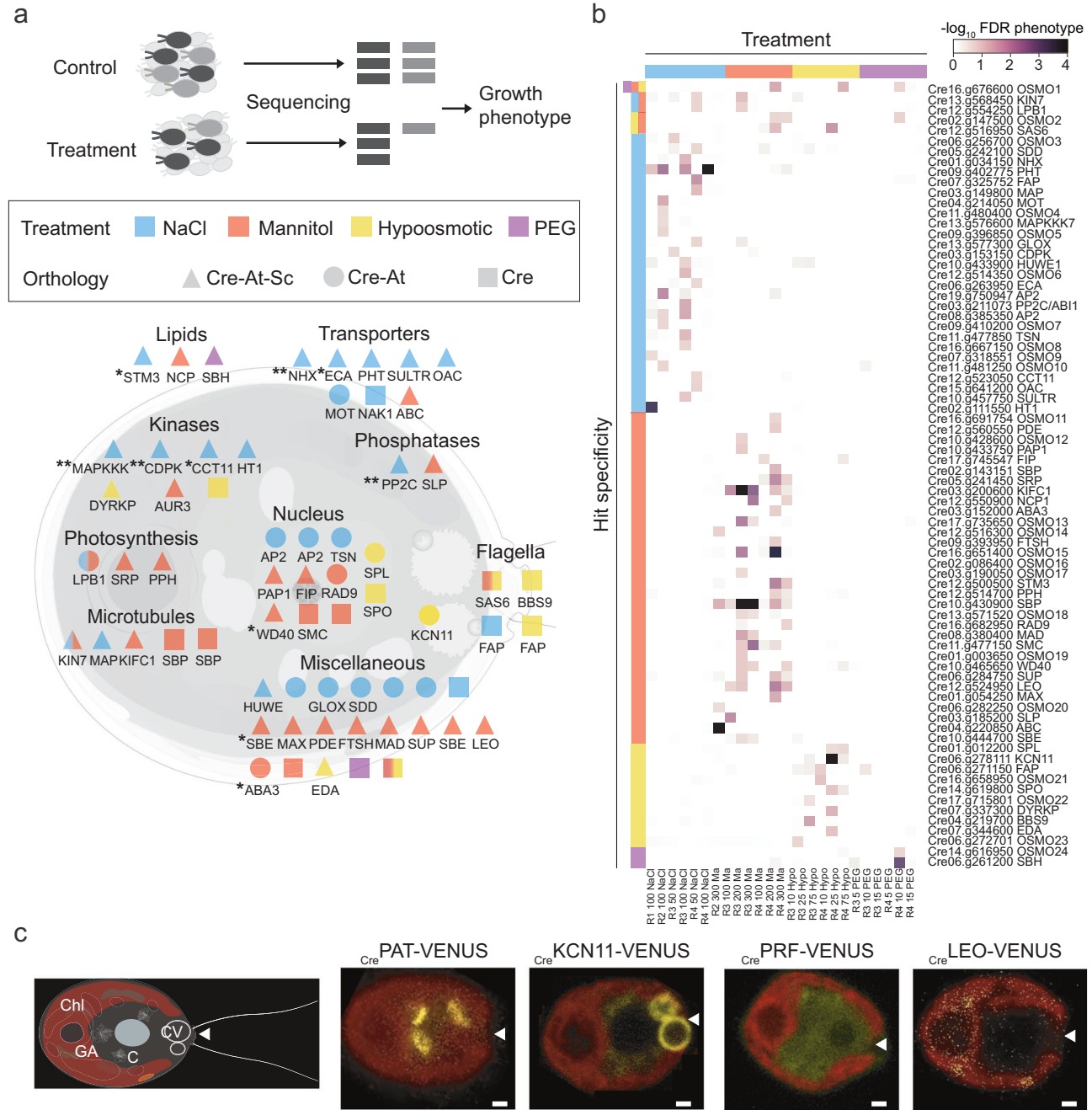

**Fig. 3 | Identification of genes with roles in osmotic stress in Chlamydomonas.**
**a** Upper panel, diagrammatic representation of barcode mutant screens. Unique barcodes allow genome-wide screening of a Chlamydomonas mutant pool. Mutants sensitive to osmotic stress can be identified because their barcodes will be less abundant after growth under treatment (NaCl, Mannitol, PEG, hypoosmotic) compared to control conditions. Lower panel, diagrammatic representation of high confidence hits (FDR < 0.3). Cre indicates Chlamydomanas specific genes. Hit orthology to S. cerevisiae (Sc) or A. thaliana (At) based on reciprocal BLAST (<1E-10). * and ** represents an osmotic phenotype previously described in Sc or At homologous genes. See also Supplementary Data 5, 6 and Source Data. **b** Heat map representing the significance of high confidence hits (FDR < 0.3). The x-axis indicates the different conditions screened, NaCl, Mannitol, PEG, and hypoosmotic stress grouped in the different replicas performed. Y-axis contains the gene ID and annotation. See also Supplementary Data 6 and Source Data. **c** Confocal images of Chlamydomonas cells showing the subcellular localization of proteins fused to VENUS fluorescent protein. Cartoon represents different organelles in Chlamydomonas: Chl chloroplast, GA Golgi apparatus, C cytoplasm, CV contractile vacuole. Red color shows chloroplast autofluorescence and yellow shows VENUS fluorescence. White arrows indicate the base of the flagella. Scale bar; 1 μm.

Nuclear localized factors identified included: transcription factors with homology to the APETALLA-2 family (AP2, Cre19.g750947 and Cre08.g385350) and SQUAMOSA PROMOTER PROTEIN-LIKE (SPL, Cre03.g185200); a WD-40 repeat-containing protein, Cre10.g465650, previously linked to drought stress in Arabidopsis[56]; and a core component of the homologous recombination pathway RADIATION DEFECTIVE (RAD, Cre16.g682950), among others. Interestingly we

found two genes whose Arabidopsis homologs were previously shown to function in acclimation to osmotic stress: a STARCH BRANCHING ENZYME (SBE, Cre10.g444700) and MOLYBDENUM COFACTOR SUL-FURTRANSFERASE/ABA DEFICIENT 3 (ABA3, Cre03.g152000) a gene encoding a molybdenum cofactor sulfurase that has been demonstrated to participate in both ABA and ABA-independent stress pathways, regulating the accumulation of anthocyanins during stress

responses within Arabidopsis[57,58]. The lack of ABA receptors in Chlamydomonas suggests that anthocyanins may play a role in osmotic stress in algae as previous shown in plants[59]. Not surprisingly, we also found limited overlap in the gene lists identified through transcriptomics and functional genomics, with just five genes (0.34%, 5 of our functional genomics overlap with 1456 genes differentially regulated upon osmotic stress) exhibiting a significant growth defect and transcriptional response upon osmotic stress: an AP2 transcription factor, SBE, a motor microtubule protein KINESIN 7 (KIN7, Cre13.g568450), a proprotein convertase subtilisin (Cre05.g242100) and Cre17.g735650, a gene without any known domain.

To determine if candidates were part of common protein complexes, we used the Arabidopsis orthologous genes to generate a protein-protein interaction network with the Arabidopsis Interactions Viewer 2.0. We identified 32 proteins out of our 56 candidates that have common interactors, suggesting that they are part of the same protein complexes (Supplementary Data 8). Overall, our functional genomic approaches identified high-confidence genes involved in osmotic stress signaling and corroborates the low overlap between transcriptomics and functional genomics previously reported in yeast for other stresses; pH (3%), 1 M NaCl (0.88%), 1.5 M Sorbitol (0.34%)[11,60,61].

Together our multi-omics characterization of Chlamydomonas identifies components with high confidence for future studies and points to the conservation of several osmotic signaling pathway components between algae, fungi, and land plants.

## Cellular processes conserved across the green lineage upon osmotic stress

To understand how the candidate genes identified in Chlamydomonas may function in a multicellular context, we grew Arabidopsis seedlings with mutations in genes homologous to our hits from the functional genomics in Chlamydomonas (homology based on Reciprocal Best BLAST hit [RBH], see material and methods) and analyzed the ability of their roots to acclimate to osmotic stress induced by NaCl or mannitol containing media. We uncovered five genes with mutants exhibiting osmotic phenotypes in both Chlamydomonas and Arabidopsis; including genes encoding for: MITOGEN ACTIVATED PROTEIN KINASE KINASE (MAPKK, Cre08.g384900), a POTASSIUM CHANNEL (KCN11, Cre06.g278111), S-PALMITOTRANSFERASE (PAT, Cre06.g277000), PROFILIN (PRF, Cre10.g427250), and LETHAL EMBRYONIC OSMOTIC (LEO1, Cre12.g524950), a gene that encodes for 50S ribosome-binding GTPase, an uncharacterized protein (Fig. 4a, b, Supplementary Fig. 7). Interestingly, in Chlamydomonas we found that these genes showed mutant phenotypes under osmotic stress but few other conditions (Supplementary Data 6), suggesting specificity in their function. We were able to localize four out of five proteins to various cellular organelles in Chlamydomonas, including KCN11 to the contractile vacuole, PAT to the Golgi apparatus, LEO to the chloroplast, and PRF to the cytoplasm and nuclear envelope (Fig. 3c, Supplementary Fig. 8), suggesting the involvement of multiple organelles in the osmotic stress signaling pathway.

Time-lapse imaging of root growth using an introgressed plasma membrane marker, LTI6b:YFP[62], revealed that several mutants exhibited distinct patterns of root growth cessation and tissue damage upon osmotic stress (Fig. 4c–f). Under standard conditions, all mutant primary roots grew similarly to wild-type except gork-1, which exhibited a loss of root hair anisotropic growth, producing rounded hair cells. Abnormal root hair development was suppressed when gork-1 mutants were grown on either media lacking supplemented potassium, at low temperatures, or by removing sucrose from the media, suggesting that the abnormal growth and loss of cellular integrity in this mutant may be dependent on intracellular accumulation of potassium or growth rate (Supplementary Fig. 9c, d).

Wild-type plants transferred to 140 mM NaCl exhibited apparent cell death (visualized by the loss of LTI6b:YFP fluorescence signal in a cell) in the late elongation and early differentiation zones, ~400–600 μm from the root tip within 4–8 h after transfer to stress media. In comparison, treatment with 300 mM mannitol did not affect cell viability. Different spatiotemporal patterns of cell-viability loss were observed in several mutants (Fig. 4c–f). The mapkk-1 mutant showed cell death in the elongation-differentiation zone upon NaCl or mannitol treatment. The pat-1 mutants showed cell death in the meristematic zone, 0–200 μm from the root tip, and elongation zone with a peak of cell death occurring 9 h after transfer to stress treatment, which resulted in roots without epidermal cells within the first 400 μm from the root tip. Similarly, mannitol treatment induced cell death to a minor degree in pat-1 mutants, starting 4 h after transfer. Early cell death occurred in the meristematic and elongation zones of prf5 mutants upon transfer to NaCl, leaving most epidermal cells dead. Comparably, mannitol treatment promoted cell death at the meristematic-elongation zone boundary (a.k.a. transition zone) as early as 2 h after treatment in prf5 mutants. Continuous cell death in the transition zone occurred in gork-1 mutants after NaCl treatment, while mannitol promoted cell death in the early elongation zone. Finally, cell death occurred in the leo-1-amiR mutant at late timepoints upon NaCl treatment, while no cell death happened upon mannitol treatment. Together our phenotypic characterization indicates that all mutants exhibit sensitivity to both NaCl and mannitol-mediated osmotic stress, though greater sensitivity was generally observed with NaCl. Mutants exhibited distinct patterns of cell viability defects, which suggests different cellular mechanisms may be mediating their sensitivity to osmotic stress.

## Profilin-mediated actin remodeling ensures cell viability under osmotic stress

The Arabidopsis prf5 mutant exhibited one of the earliest and most severe defects in cell viability amongst our characterized mutants, with most meristematic and elongating cells dying within the first 5 h after transfer to stress. In addition, prf5 mutant root growth is hypersensitive to the actin depolymerization drug Latrunculin B (LatB) (Supplementary Fig. 10). Similarly, Chlamydomonas prf1-1 mutant is hypersensitive to LatB treatment (Supplementary Data 6). Profilin is an actin interacting protein that determines the dynamics of the actin cytoskeleton by controlling the rate of polymerization, bundling, and cable formation[63,64]. Osmotic stress promotes actin reorganization in leaf and differentiated root tissues[65,66], but the cellular consequences and molecular components controlling this process remain unknown. We therefore hypothesized that prf5 mutant roots exhibit hypersensitivity to stress due to the inability to properly control this actin reorganization. To test this hypothesis we monitored actin dynamics in elongating cells of Arabidopsis roots using the actin-binding domain of fimbrin fluorescent reporter, ABD2:GFP[67]. Upon transfer to osmotic stress, both 140 mM NaCl and 300 mM mannitol, root cells in the elongation zone underwent massive actin reorganization (Fig. 5a, b). An increase in the asymmetric distribution of actin filaments, skewness, revealed an increase in actin bundling together with a decrease in the filament number. Concomitantly, actin filament angle switched from being nearly parallel to the longitudinal axis of the cell to being perpendicular[68] (Fig. 5a, b). Interestingly, visualization of actin dynamics in Chlamydomonas using Lifeact-NeonGreen showed a similar response to osmotic stress with an increase in skewness (Supplementary Fig. 11). Together these data suggest that actin reorganization upon osmotic stress is conserved across the green lineage.

Treatment of prf5 mutants with 140 mM NaCl led to cell death in the elongation zone (Fig. 4c, e). Treatments with 50 mM NaCl, on the other hand, did not cause such tissue damage and promoted actin reorganization in the prf5 mutant (Fig. 5c, d). No changes were observed in wild-type roots grown under these conditions suggesting

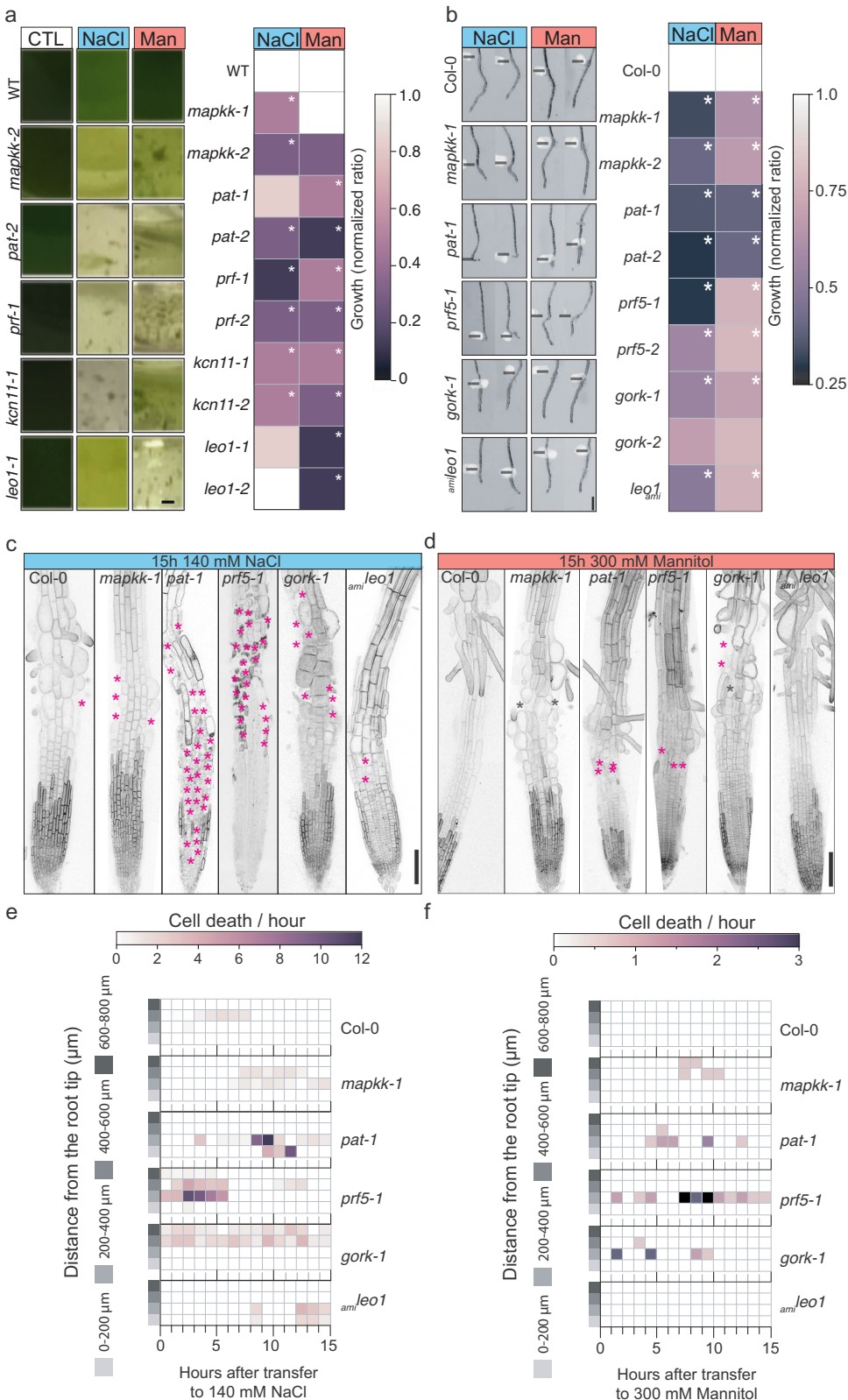

that the *prf5* actin network is hypersensitive to osmotic stress (Fig. 5a, b). Localization of a *ProPRF5:PRF5:GFP* translational reporter revealed fluorescence in root hair cells of the elongation zone and differentiation zone as well as the most mature two layers of the lateral root cap. This specific pattern of expression contrasts with the much broader pattern of cell death observed in the mutant and suggests that the loss of integrity in one cell type may cause defects in tissue

integrity across the entire epidermis (Supplementary Fig. 12, Fig. 4c, e). Interestingly, CrPRF has low affinity for phosphatidylinositol[69], and thus low affinity for cell membranes[70], which may explain the cytoplasmic localization of the protein (Fig. 3c). Time-course imaging of the expression of P*roPRF5:PRF5:GFP* upon osmotic stress revealed the downregulation of its expression during the initial hours after treatment (Supplementary Fig. 12). Since our mutant analysis suggests that

**Fig. 4 | Conserved osmotic stress pathways across the green lineage show different spatiotemporal cell death patterns in Arabidopsis roots.**
**a** Representative growth vessels of 4 days post inoculation Chlamydomonas cells growing in control media (CTL), 100 mM NaCl (NaCl) and 300 mM Mannitol (Mannitol). (right) Growth quantification of mutant strains upon stress were normalized to growth of wild-type upon stress. Duplicate cell counts were performed in three independent replicates.* represents $p < 0.05$ (two-way ANOVA). Scale bar = 500 mm. **b** Arabidopsis primary roots 4 days post-transfer to 140 mM NaCl or 300 mM mannitol, black lines indicate transfer side, scale bar = 1 mm. (right) Quantification of root growth after transfer to 140 mM NaCl or 300 mM mannitol, root growth ratio upon transfer was normalized to the growth of wild-type upon

stress treatment. * Indicates significance, two-way ANOVA test interactions between genotype and treatment, $p < 0.05$. See also Supplementary Fig. 7 and Supplementary Data 9 and Source Data for raw data and statistical information. **c** Confocal images of 4 days post germination root tips, 15 h after transfer to 140 mM NaCl. * Indicates cell death. Scale bar = 100 μm. See Supplementary Fig. 9 for controls. **d** Confocal images of 4 days post germination root tips, 15 h after transfer to 300 mM mannitol. * Indicates cell death. Scale bar = 100 μm. See Supplementary Fig. 9 for controls. **e** Quantification of root cell death after transfer to 140 mM NaCl. $n = 10$ roots for each genotype in three independent replicates. **f** Quantification of root cell death after transfer to 300 mM mannitol. $n = 10$ roots for each genotype in three independent replicates.

PRF5 inhibits NaCl-induced bundling, we hypothesize that PFR5's normal function may be to delay the induction of osmotic-stress mediated actin bundling.

Calcium signaling is an important downstream mediator of abiotic stress[14,71]. To test whether the defects observed in *prf5* are due to a lack of a stress response or a hyper-induction of a stress response, we monitored calcium dynamics in roots transferred to a mild NaCl concentration in wild-type and *prf5* backgrounds (Fig. 5e, f). Wild-type plants transferred to 50 mM NaCl showed calcium spikes in the early elongation zone with ~40 spikes per hour within the first 5 h, compared to the lack of such dynamics when transferred to control media. In contrast, *prf5* mutants showed an increase in spiking frequency immediately after transfer to stress. Thus, osmotic-stress mediated calcium dynamics are hyperactivated in *prf5* mutants, supporting the hypothesis that this mutant exhibits a heightened stress response due to the inability to acclimate appropriately through the reorganization of the actin cytoskeleton.

Osmotic-stress induces plasmolysis in walled organisms when enough water leaves the cell to cause turgor pressure to completely subside and the plasma membrane to partially detach from the wall[72]. Typically plasmolysis occurs only after a severe osmotic shock and is typically measured in Arabidopsis roots after treatment with 500 mM mannitol treatment[72]. Examination of the plasma-membrane in *prf5* mutants after treatment with 250 mM mannitol demonstrated that this occurred even at this lower-level of stress, which did not significantly affect wild-type roots (Fig. 5g, h). Pretreatment with LatB, an actin depolymerizing drug, followed by an osmotic shock, increased the extent of plasmolysis compared to no LatB treatment (Supplementary Fig. 13, Fig. 5g, h). Treatment of seedlings with a more severe osmotic shock was lethal to *prf5* roots, while wild-type was able to recover after transfer back to control conditions (Supplementary Fig. 14)

## Discussion

Across Chlamydomonas, Saccharomyces, and Arabidopsis we have observed common transcriptional signatures to reduce investment in growth and to redirect resources towards osmotic homeostasis. Importantly, canonical ABA-mediated osmotic response is not conserved between land plants and algae. Not surprisingly our systems biology approach yielded complementary insight across approaches but with low overlap between candidates identified through transcriptomic, phosphoproteomic, and functional genomic, as was previously described for other model systems and biological processes[61]. The transcriptomic response uncovered the dominant non-hormonal nature of the Chlamydomonas osmotic response compared to land plants, thus providing unique insight into these alternative stress responses. Our phosphoproteomic approach suggested a substantial role for phosphatases in the regulation of immediate osmotic-stress induced postranslational modifications, similar to land plants[73], while our functional genomic approach led to the identification of dozens of new genes with confirmed functions in osmotic stress, however, clear limitations were also revealed. The lack of multiple alleles for all genes and the limited representation of small genes in the mutant library used[11] makes it difficult to assess the genetic role of all genes in the

genome. The generation of future functional genomic tools in suitable photosynthetic organisms holds the potential to significantly accelerate our understanding of fundamental biological mechanisms, as shown in other organisms[74]. Nevertheless, the scale at which this study characterizes the osmotic stress pathway provides an extensive resource for hypothesis testing and refining the search space for candidate genes and proteins when combined with other large-scale methods.

Land plants use cation/H+ antiporters to detoxify intracellular sodium[75] and lack P2C Na+/K+ type ATPases present in Chlamydomonas and Ostreococcus, species of fresh and marine algae, respectively[49]. The presence of both detoxifying mechanisms allows Chlamydomonas to rapidly adapt to seawater[76], while many vascular plants, show high sensitivity to elevated Na+ in the soil[77]. Halophytes, which are plants tolerant to high-sodium soils, represent an exception as they have evolved distinct mechanisms to handle intracellular osmolytes[50]. Recent identification of marine Chlamydomonas support the strong sodium detoxifying mechanism present in this algae[78], and suggest that algae P2C Na+/K+-ATPase can be used to engineer salt tolerant land plants as previously hypothesized[79]. Chlamydomonas uses the contractile vacuole, an ancient organelle present in freshwater protists without cell walls, to remove excess intracellular water under hypoosmotic stress[13]. We found a conserved role for an outward rectifying K+ channel that underwent an evolutionary relocation from the contractile vacuole in Chlamydomonas (KCN11)[80] to the plasma membrane in Arabidopsis (GORK)[81].

Actin is a highly conserved protein found in eukaryotic cells, and its organization is crucial for various cellular processes including cellulose deposition[82], vacuole morphology[83], calcium signaling[84,85], plant programmed cell death[86,87], and ion channel activity[88,89]. The actin cytoskeleton is reorganized upon osmotic stress in various organisms including yeast, plant and animal cells to maintain cell integrity and function, and the observed dynamic reorganization led to the hypothesis that actin may function as a bona fide osmosensor[1].

Following sensing, the reorganization of actin in response to osmotic stress affects the cell through both direct and indirect regulatory and mechanical pathways. On one hand, elevated osmolarity can directly influence actin filament formation and affect the activity and binding of actin-interacting proteins due to increased intracellular ion concentrations[90]. The regulation of ion channel activity by actin interaction is exemplified by the tethering of the mechanosensitive calcium channel, Piezo, which regulates its activity[89], and where disruption of the actin cytoskeleton impairs Piezo-mediated responses[91]. On the other hand, osmotic stress can alter turgor pressure, subsequently modifying the mechanical properties of the cell wall and cell membrane, which could in turn indirectly affect actin dynamics. Osmotic-dependent actin reorganization could also impact organelle dynamics such as vesicle trafficking[92] and affect cellulose biosynthesis[82], resulting in impaired cell wall rigidity.

We propose a model where plant cells use the actin cytoskeleton as a readout of their osmotic status and use the reorganization of the actin network to orchestrate changes in cellular organization that facilitate acclimation to osmotic stress. However, whether actin and/or

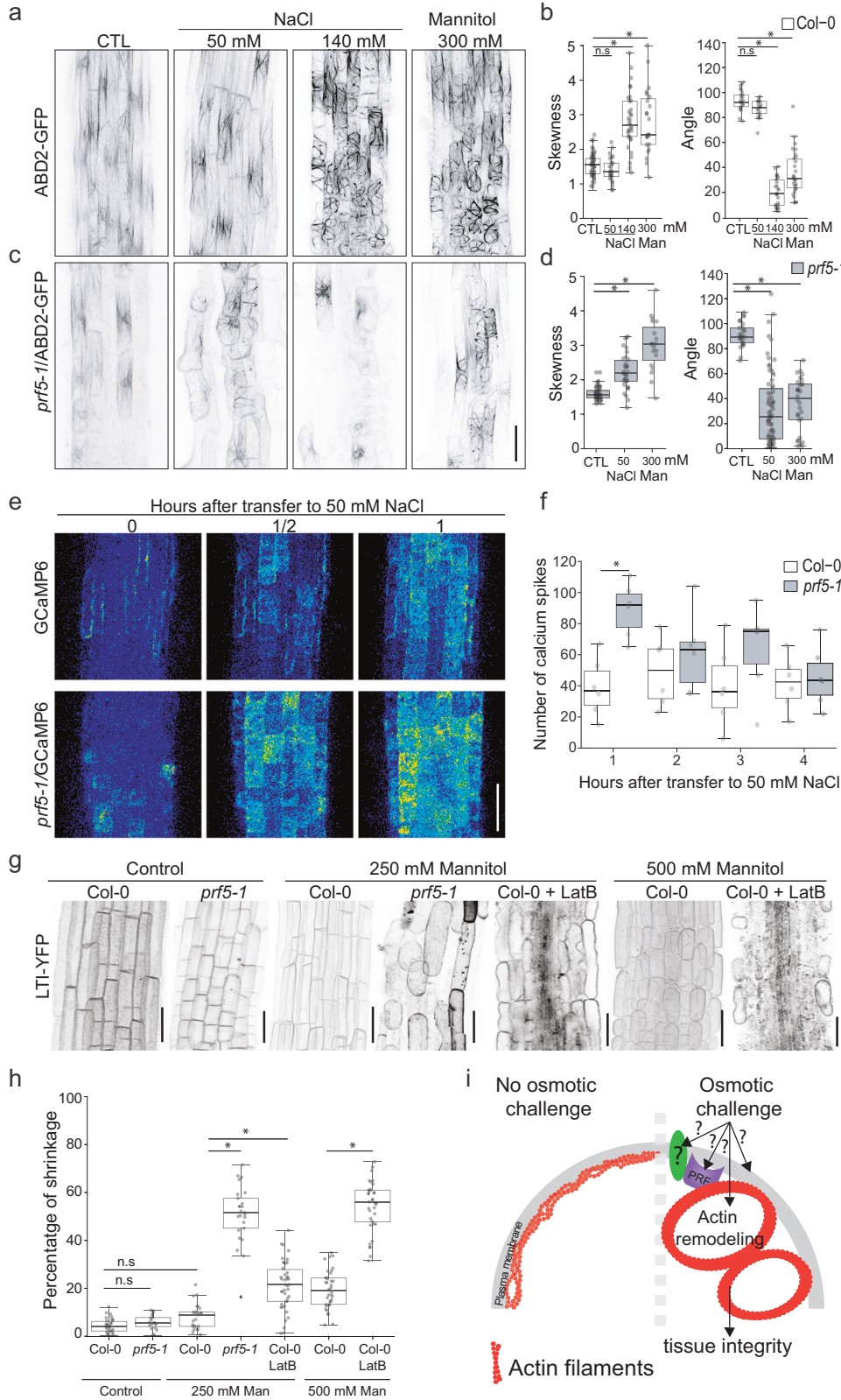

actin interacting proteins directly react to the cellular biomechanical alterations caused by water loss, or indirectly through the secondary signals generated by water availability, is still unclear. Further investigation into mechanical and signaling functions of the actin cytoskeleton under osmotic stress will enhance our understanding of the function that actin reorganization has on the acclimation of cells to environmental stress.

## Methods

### General maintenance of *Chlamydomonas reinhardtii* strains and sample preparation

The background *Chlamydomonas reinhardtii* strain for all experiments was wild-type (WT) cMJ030 (CC-4533). All *C. reinhardtii* strains were maintained in Tris-Acetate-Phosphate (TAP) solid media, 1.5% agar, with modified trace elements[93] at 22 °C in low continuous light

**Fig. 5 | Osmotic stress promotes PROFILIN mediated actin-reorganization, maintaining cell and tissue integrity upon osmotic stress. a** Actin localization in the Arabidopsis primary root elongation zone of wild-type roots on control conditions (CTL), transferred to 50 mM NaCl, 140 mM NaCl or 300 mM mannitol for 16 h (left to right). Images from treatments correspond to cells actively elongating at the time of transfer. Actin filaments were visualized with the actin binding domain of the FIMBRIN protein tagged with green fluorescent protein (ABD2:GFP). Scale bar = 50 μm. **b** Quantification of skewness and angle of actin filaments relative to the cell radial axis. Each point represents a cell that was actively elongating when roots were transferred. $n > 25$ cells per condition, taken from three independent experiments. Center lines show medians and box limits indicate the 25th and 75th percentiles. Whiskers represent minima and maxima, (*) $P$ values are from a one-way ANOVA test between control condition and treatments, $p < 0.05$. See Source Data. **c** Actin localization in the Arabidopsis root elongation zone of *prf5-1* roots on control conditions (CTL), transferred to 50 mM NaCl, 140 mM NaCl, or 300 mM mannitol for 16 h (left to right). Images from treatments correspond to cells actively elongating at the time of transfer. Actin filaments are visualized with ABD2:GFP. Due to silencing of the reporter and a loss of cell viability, some cells within the field of view are not marked by GFP expression. Scale bar = 50 μm. See Source Data. **d** Quantification of skewness and angle of actin filaments relative to the cell radial axis. Each point represents a cell that was actively elongating when roots were transferred. $n = 25$ cells per condition, taken from three independent experiments. Center lines show medians and box limits indicate the 25th and 75th percentiles.

Whiskers represent minima and maxima, (*) $P$ values are from a one-way ANOVA test between control condition and treatments, $p < 0.05$. See Source Data. **e** 16 colors LUT confocal images of primary root elongation zone expressing GCamp6, and *prf5-1, GCamp6* in control conditions (0) or upon transfer to 50 mM NaCl for ½ h and 1 h. Images represent z-projection in each timepoint. Scale bar = 50 μm. **f** Hourly average number of $[Ca^{2+}]$ spikes in wild-type (GCamp6) cells and *prf5-1* with introgressed GCamp6 (*prf5-1;GCamp6*) treated 50 mM NaCl. $n = 6$ roots, taken from three independent experiments. Center lines show medians and box limits indicate the 25th and 75th percentiles. Whiskers represent minima and maxima, * indicates significance, one-way ANOVA test between genotypes 1 h after transfer to 50 mM NaCl, $p = 0.0007179$. See Source Data. **g** Confocal images of 5-day old primary root expressing membrane marker LTI-YFP in wild-type plants and *prf5-1* mutants. 250 and 500 mM Mannitol represent the plasmolysis treatments performed for 60 min. Control roots were incubated in liquid MS media lacking Mannitol. Scale bar = 50 μm. **h** Quantification of protoplast shrinkage in wild-type (Col-0) and *prf5-1* mutants. Percentage of shrinkage was calculated based on the protoplast area (LTI-YFP signal) and the cell wall area (bright field channel) (see Material and Methods). $n = 37$ cells, taken from three independent experiments. Center lines show medians and box limits indicate the 25th and 75th percentiles. Whiskers represent minima and maxima, * indicates significance, one-way ANOVA test between different genotypes or treatments within the same plasmolysis regime, $p < 0.001$. See Source Data. **i** Cartoon illustrating actin filaments remodeling upon osmotic stress to promote tissue integrity.

(-15–10 μmol photons m$^{-2}$ s$^{-1}$). Lines harboring VENUS-3xFlagged tagged genes were maintained in the same conditions with media supplemented with 20 μg/ml hygromycin. Routine maintenance intervals were every 4 weeks. Complete list of mutant strains used can be found in Supplementary Data 11, and primers used in Supplementary Data 10.

## RNAseq samples and analysis

RNA-seq samples: prior to sample collection wild-type strains (CMJ030/CC-4533) were refreshed in solid media and inoculated in liquid media, Tris-Acetate-Phosphate (TAP) with modified trace elements[93] at 22 °C in low continuous light (100 μmol photons m$^{-2}$ s$^{-1}$). After 2 passages in liquid media (-6 doublings/passage), cells growing in early-mid exponential phase (-2E + 06 cells/ml) were used for inoculation. Once cultures reached early-mid exponential phase (-2E + 06 cells/ml) different osmolyte concentrations were inoculated, 100 mM NaCl (Sigma) and 300 mM mannitol using 1 liter Erlenmeyer flasks and grown with mild agitation (-150 rpm), 22 °C and low continuous light. Samples were collected at the designed timepoints from the same flask. Sample collection of -1E07 cells was performed using a funnel with a POLYVINYLIDENE FLUORIDE membrane (GCWP4700, millipore) coupled to a vacuum station and subsequently cells were snap-frozen in liquid nitrogen and saved at −80 °C until processing.

For each treatment we performed 3 biological replicates with 2 control conditions, one at the beginning of the time course and another 24 h after inoculation. RNA extraction was performed using Direct-zol RNA miniprep kit (R2050) following manufacturer's instructions including in-column DNase I treatment prior RNA washing and elution steps. Each RNA sample was run on an Agilent 2100 Bioanalyzer RNA 6000 Nano chip for quantification and quality control. RNA samples were submitted to the Stanford Functional Genomics facility for subsequent library preparation and sequencing. RNAseq libraries were made using the Kapa mRNA HyperPrep kit (Roche, KK8540) following manufacturer's protocols. Libraries were pooled based on fragment analyzer concentrations. Sequencing was performed on Nextseq high-output flow cell, 1 × 75 bp run (Ilumina).

RNA-seq reads were aligned using BWA to the Chlamydomonas reference genome 5.6[94]. DESeq2 package was used to call transcripts as differentially expressed at a false-discovery-rate (Benjamini-Hochberg method) FC > 2, (FDR < 0.01). GO-term enrichment analysis was performed with the BINGO plugin for Cytoscape[95]. To perform GO-term-enrichment analysis of differentially regulated transcripts, we used several approaches. First, we used GO terms from the Joint Genome

Institute (JGI) for Chlamydomonas genome 5.6 version, and used these as a reference set. Second we used the annotations of the homologous genes in *Arabidopsis thaliana*. To identify these homologs, we identified gene pairs by reciprocal best BLAST hit (RBH) and selected "Best hit" for each pair having the lowest e-value, setting an e-value threshold of 1E-10.

## Proline quantification

Proline quantification in Chlamydomonas cells was adapted from[96]. Briefly, 1E05 cells from control or treated samples in early-mid exponential growing phase were collected and transferred to 96 wells plates together with L-proline standards. Subsequently the same volume of glacial acetic acid and acid ninhydrin was added to each well, sealed and incubated for 1 h at 100 °C. Following incubation and cool down of samples chromophore was extracted with Toluene and the absorbance read at 520 nm using toluene as blank. Determination of proline concentration was performed from a standard curve.

## Genomic comparisons upon stress

To generate the pie charts in Supplementary Fig. 3, genome wide orthology was assessed using reciprocal best BLAST hit (RBH) requiring a E-values 1E-10. FASTA protein sequences were downloaded from the JGI, Phytozone v 12 (phytozone.jgi.doe.gov), (Saccharomyces Genome Database (www.yeastgenome.org) and The Arabidopsis Information Resource (www.arabidopsis.org). Osmotic pie chart was generated with all differentially expressed genes from our transcriptomic experiment (1456 genes (FC > 2, FDR < 0.01) and assesing orthology with RBH requiring an $E < $ 1E-10.

Transcriptome datasets from Arabidopsis and yeast were selected that mimic our experiments in Chlamydomonas. The transcriptomic data generated in Arabidopsis was performed with osmolyte concentrations that are not lethal to plant cells, promote growth arrest in short term treatments allowing growth recovery after a short period of acclimation(4 h)[7,97], similarly to what we report for our treatments (Supplementary Fig. 1 a, b). Additionally, the high spatial resolution of the transcriptomic experiments in Arabidopsis provide a greater number of deregulated genes than are zone specific[7], not identified in previous studies.

Transcriptome responses to osmotic stress is highly dynamic in yeast and varies with concentrations, yeast strain and time. We used transcriptome dataset performed with yeast strain BY4741[98], with deregulated genes previously reported in other yeast strains and

osmotic treatments[99,100]. The yeast iron dataset used[101] uncover a complete list of genes regulated upon iron deficiency and gives a global perspective of cellular processes involved in iron metabolism[101]. Chlamydomonas iron deficiency transcriptomes from ref. 102 were grown in the same media used for our osmotic treatments (TAP), and growing conditions (light regime), thus being the most comparable conditions to our experiments. The bar size represents the number of Chlamydomonas osmotically or iron regulated genes with an orthologous gene either in yeast or Arabidopsis. The color filled within the bars represent the fraction of genes regulated in yeast (orange) or Arabidopsis (blue). As an example, from the total osmotically regulated genes in Chlamydomonas 6% are osmotically regulated in yeast.

## Phosphoproteomics samples and analysis

Phosphoproteomics samples: prior to sample collection wild-type strains (CMJ030/CC-4533) were refreshed in solid media and inoculated in liquid media, Tris-Acetate-Phosphate (TAP) with modified trace elements[93] at 22 °C in low continuous light (100 μmol photons $m^{-2} s^{-1}$). After 2 passages of cells growing in early-mid exponential phase (~6 doublings/passage), with either $14NH_4Cl$ or $15 NH_4Cl$ as a sole source of nitrogen, cells growing in early-mid exponential phase (~2E + 06 cells/ml) were used to inoculate 2 liter bottles with bubbled air and constant stirring at an initial concentration of 2E04 cells/ml. This ensured 15N incorporation, cells were grown for at least eight generations. Cells at early-mid exponential phase were spun out (2000 g, 5 min, 4 °C) and resuspended in a 1:1 (v/w) ratio of ice cold homogenization buffer (290 mM sucrose, 250 mM Tris-HCL (pH = 8), 25 mM EDTA, 25 mM sodium fluoride, 50 mM sodium pyrophosphate, 1 mM ammonium molybdate, 0.5% polyvinyl pyrovinyl pyrrolidone in water and 1 complete EDTA-free protease inhibitor (Sigma-Aldrich) for 50 ml of buffer. The cell slurry was then added drop wise to liquid nitrogen to form small Chlamydomonas pellets ~5 mm in diameter. Samples were stored at −80 °C.

Samples for phosphopeptide enrichment were prepared as previously described (Minkoff 2014) with minor modifications. Samples were combined in an experimental pair consisting of one treated sample grown in $14NH_4Cl$ and one control sample grown in $15NH_4Cl$. For the second reciprocal experimental pair, the samples were combined in the inverse fashion ($15NH_4Cl$ treated and $14NH_4Cl$ control). Frozen cells were combined at 1:1 weight ratio prior homogenization, with a total weight of 4 grams (2 grams of cells grown in $14NH_4Cl$ and 2 grams of $15NH_4CL$). Three biological replicates were processed for each treatment condition. For all experiments, samples were processed in homogenization buffer supplemented with phosphatase inhibitors (Minkoff 2014) using a sonicator (1 cm probe, (12 × 5 s) × 2, and 50% duty cycle (Bransom 45 Digital Sonifier (Marshall Scientific)) while kept on ice. The resulting homogenate was filtered through two layers of Miracloth (Calbiochem) and spun out 15 min at 1500 g and 4 °C. Subsequent steps were performed as described previously[103].

Phosphopeptide-enriched samples were analyzed on an LTQ-Orbitrap XL mass spectrometer (Thermo Scientific). Acquired data files containing MS/MS spectra were searched against the JGI *Chlamydomonas reinhardt* annotation v5.6 protein database using MASCOT software (Matrix Science). Searches were performed using settings for both 14N and 15N protein masses. MASCOT search results were filtered to maintain a 1% false discovery rate at the peptide level using a reverse-protein sequence database. Quantitative ratio measurements from peak areas were performed using Census software[104]. Only phosphopeptides showing reciprocal changes of 2-fold or greater in two of the three replicates were selected for data shown in Fig. 1. Phosphopeptides showing reciprocal changes of 1.5-fold or greater were used to identify homologous phosphorylated proteins in Arabidopsis (Supplementary Data 3).

## Chlamydomonas mutant screen

Library maintenance, pooling, competitive growth, DNA extraction, barcode amplification, and library preparation were performed as described[41]. To identify mutants with growth defects or enhancements due to a specific treatment, we compared the abundance of each mutant after growth under the treatment condition to its abundance after growth under a control condition. A minimum number of 50 reads was required for the phenotype to be calculated. PRO and MAPKK genes are small genes that did not reach the threshold required for the high confidence list, which required at least 3 mutant alleles with a growth defect. The mutant phenotypes of PRF and MAPKK were identified during the initial screens of the mutant library only based on mutant growth rate of single mutants (see Supplementary Fig. 6 for growth data). For each mutant allele contingency table of the phenotypes [Φ < 0.0625, 0.0625 ≤ Φ < 0.125, 0.125 ≤ Φ < 0.25, 0.25 ≤ Φ < 0.5, 0.5 ≤ Φ < 2.0, 2.0 ≤ Φ < 4.0, 4.0 ≤ Φ < 8.0, 8.0 ≤ Φ < 16.0]. High confidence hits were identified based on the phenotype of multiple mutant alleles of each gene. For high confidence genes, we generated a contingency table of the phenotypes, Φ, by counting the number of alleles that met the following thresholds: [Φ < 0.0625, 0.0625 ≤ Φ < 0.125, 0.125 ≤ Φ < 0.25, 0.25 ≤ Φ < 0.5, 0.5 ≤ Φ < 2.0, 2.0 ≤ Φ < 4.0, 4.0 ≤ Φ < 8.0, 8.0 ≤ Φ < 16.0]. A p value was generated for each gene by using Fisher's exact test to compare a gene's phenotype contingency table to a phenotype contingency table for all insertions in the screen.

A false discovery rate was performed on the p-values of genes with more than 2 alleles using the Benjamini-Hochberg method. High-confidence gene-phenotype relationships (Fig. 1, Supplementary Data 6) were based on false discovery rate (FDR < 0.3) using the Benjamini-Hochberg method[105].

Phylogenetic relationships between Chlamydomonas genes and Arabidopsis orthologs revealed that (i) PRF, PAT gene families expanded during evolution (PRF; single member in Chlamydomonas [Cr], five members in Arabidopsis [At][106]; PAT from 9 to 24 members[107] (ii) MAPKKK gene family have a greater number of members in Chlamydomonas 108 compared to 89 in Arabidopsis[108] (iii) Voltage gated K+ channels structural diversity in Chlorophyta collapsed with the transition from aquatic environments to terrestrial environments[80,109]. Further characterization of the responses to osmotic stress of complete gene family members will unveil the degree of functional conservation between ortholog genes. Despite the similarity between CreLEO1 and its ortholog AtLEO1 (RBH, E-value 5E-61), further annotation and phylogenetic analysis of this gene family is required to uncover the evolutionary functional conservation.

## Secondary screens

Chlamydomonas mutants selected for secondary screens were picked from the Chlamydomonas mutant library and inoculated in 96-well plates containing liquid TAP and 10 μg/ml paromomycin. Plates were re-inoculated 3 times (ensuring ~10 doublings), and early-mid exponential cells were used to inoculate control plates (TAP liquid), plates supplemented with 100 mM NaCl or plates containing 300 mM mannitol. Plates were grown under low light conditions. 5 days post-inoculation plates were scanned using a CanonScan 9000F flatbed scanner (Canon). Images were quantified using Fiji. 8-bit RGB images were converted to gray scale and average pixel intensity was measured with a sampling area including the entire well. Wells with bubbles and/or irregular clamping were avoided and not quantified. Comparisons between average pixel intensity were used to estimate the mutant growth compared to wild-type. Each plate contained 2 biological replicates of each mutant and multiple wild-type colonies to avoid any positional effect. The resulting Z-scores can be found in Supplementary Data 7.

## Chlamydomonas and Arabidopsis protein localization

To generate fluorescently tagged proteins in Chlamydomonas, we used pRAM118 (a generous gift from Silvia Ramundo). Open reading frames were PCR amplified (Phusion Hotstart II polymerase, Thermo-Fisher Scientific or KOD DNA polymerase, Millipore) from genomic DNA or BAC, gel purified (MinElute Gel Extraction Kit, QIAGEN) and cloned in frame with the C-terminal VENUS-3xFLAG tag by Gibson assembly. Primers were designed to amplify target genes from their predicted start codon up to, but not including, the stop codon (Supplementary Data 10). All resulting constructs were verified by Sanger sequencing. Constructs were linearized by EcoRV prior to transformation. Electroporation was performed with cells grown to ~8E06 cells/ml, resuspended at ~8E08 cells/ml in CHES buffer at room temperature, and electroporated in a volume of 125 μl in a 2-mm-gap electro cuvette using a NEPA21 square-pulse electroporator (Bulldog Bio)[110], using two poring pulses of 250 and 150 V for 8 ms each, and five transfer pulses of 50 ms each starting at 20 V with a "decay rate" of 40%[111]. Cells were transferred to a 15-ml centrifugation tube containing 8 ml TAP plus 40 mM sucrose. After overnight incubation at 24 °C under low light, cells were collected by centrifugation and spread on TAP agar (1.5% w/v) plates containing 25 μg/ml hygromycin.

Colonies for plate reader analysis resulting from transformation plates were picked and transferred to a 96-well microplate plate containing 150 μl liquid of TAP supplemented with 25 μg/ml hygromycin, without shaking. Fluorescence readings were acquired with excitation and emission wavelengths of 515 and 550 nm, TECAN infinite 200 Pro microplate reader. 5 independent colonies with values greater than background were further selected for imaging. Expression of Cre12.g524950:VENUS (LEO) was further confirmed with Western blot.

Arabidopsis microRNA silencing lines for *leo-1* were generated using pAmiR containing an amiRNA hairpin targeting At2g25660. The pAmiR containing the hairpin vector was obtained from the ABRC, stock number CSHL_03309. Col-0 plants were used as a background and transformed. T2 homozygous lines containing single insertions were selected using BASTA to generate homozygous T3 plants. The Profilin 5 expression under the native promoter construct was generated using golden gate reactions to assemble level 1 cassettes flanked by Bbs1 cut sites. The three resulting cassettes were combined using Bbs1-based golden gate cloning into a binary vector JAB2158 (a generous gift from Jennifer Brophy).

## Arabidopsis growth and treatments

Arabidopsis thaliana Col-0 ecotype was used in this study. A complete list of the mutant lines used in this study can be found in Supplementary Data 11 and primers used to genotype the mutant lines in Supplementary Data 10. Seeds were surface sterilized by washing with 20% bleach for 5 min and rinsed with sterile deionized water four times. After stratification seeds were grown in 10 × 10 cm petri dish plates containing sterile full MS media (MSP01-50LT; Caisson), 1% Sucrose (Sigma-Aldrich), 0.7% Gelzan, 0.05% MES (Sigma-Aldrich) adjusted to pH = 5.7 using 1 M KOH. Screening of Arabidopsis mutants was performed with seedlings growing under standard media and transferred to plates containing 140 mM NaCl (Sigma-Aldrich) or 300 mM mannitol (Sigma-Aldrich) after 4 days post germination.

Growth of seedlings was performed in a Percival CU41L4 incubator at constant temperature 24 °C with 14 h light and 10 h dark cycles at 130 μmol m$^2$ s$^{-1}$ light intensity. Plates were sealed using micropore tape (3 M). Plates were placed vertically to allow vertical growth of the roots. Images of seedlings were captured using Epson perfection V800 Photo color scanner, root length was quantified using Fiji[112].

## Microcopy

Confocal laser microscopy was performed on a Leica TCS SP8 inverted confocal scanning microscope in resonant scanning mode with LASX software. Chlamydomonas cells were mounted in chambered coverglass (Ibidi, 80826), covered with 2% low melting agar, and imaged with a 93x/1.3 NA glycerin immersion objective. Arabidopsis roots were mounted in chambered coverglass (ThermoFisher, 155360) with a pad of gelzan MS media or gelzan MS media supplemented with corresponding treatment. Root growth time lapse movies were taken using the Navigator mode, with the root tip position in the middle of the second tile at the beginning of the time course, to ensure capture of the root growth upon the 16 h period. Fluorescence of LTi6b:YFP, ABD2:GFP and GCaMP6 lines was captured with a white light laser, excitation set at 488 nm, and detection from 500–550 nm with a HyD SMD hybrid detector (Leica).

All image quantifications were performed with Fiji[112]. Quantification of root cell death was performed using Z-stack projections and manual counting of cell death. Actin quantification was performed with stack images taken from the first visual cortical actin in the epidermal elongating cells, collecting 20 steps of 1 μm each. Imaging parameters gain and pinhole were selected such that individual actin filaments could be observed, but actin filament bundles were not saturated. All cropped images used for quantification were taken from original 8-bit files. Actin images were cropped along the entire length of every cell in the elongation zone. Skewness was analyzed according to ref. 113. No image processing was applied to maximum-intensity projections that were analyzed for skewness. The size of each cell crop was maintained constant across all images and was smaller than the entire length of a cell (10 × 10 μm). Relative angle of actin filaments was computed using Fiji and aligning cells horizontally. Calcium spike quantification was performed at the transition zone in root epidermal cells (~200 μm from the stem cell niche) during a period of 4 h. Roots were mounted in chambered coverglass (ThermoFisher, 155360) with media supplemented with 50 mM NaCl, 0 h timepoints corresponds to the first image stack taken, ~10 min after mounting the sample. Registration was applied to time lapse image stacks using the Hyper-StackReg plugin. A MaxEntropy Threshold macro was applied and 8 ROIs (24 × 24 μm) were selected across a 150 μm area to measure the signal corresponding to calcium spikes. A minimum of 6 pixels was used to exclude sporadic signals. Calcium spikes and their duration were confirmed manually.

Plasmolysis experiments were performed by treating 5 day post germination Arabidopsis roots with 250 mM or 500 mM Mannitol for 60 min in liquid solution. Seedlings treated with LatB were incubated prior to the plasmolysis induction with MS liquid media containing LatB or DMSO as control. Roots were mounted with the corresponding plasmolysis solution and imaged following the treatment. The percentage of shrinkage was calculated from the cell protoplast area and total cell area using the bright field channel.

## Expression data

We used 2 mg of 6 day old seedling to perform expression analysis of the PROFILIN5 gene in both wild-type plants (Col-0) and 2 independent mutant alleles of PROFILIN. RNA was extracted using Quick-RNA Plant kit (Zymo Research, R2024) following the manufacturer's instructions. cDNA synthesis was performed using iScript Reverse transcription Supermix for RT-qPCR (Biorad, Cat#1708841). Detection of cDNA levels was performed using SensiFAST SYBR No-ROX Mix (Bioline, BIO-9820) with Biorad CFX384 Real-time system.

## Statistical Information

Statistical analysis was performed in Microsoft Excel. *P* values and sample number are included in figures and figure legends. See additional Supplementary tables for raw measurements and *p* values.

## Reporting summary

Further information on research design is available in the Nature Portfolio Reporting Summary linked to this article.

## Data availability

All data are available in the manuscript, in the Supplementary material or in the following databases; high-throughput sequencing data sets generated in this study are available through the National Center for Biotechnology Information Sequence Read Archive (SRA), GSE260814. Phosphoproteomic data sets generated in this study are available throught the Center for Computational Mass Spectometry, Mass Spectrometry Interactive Virtual Environment (MassIVE) (https://massive.ucsd.edu/ProteoSAFe/static/massive.jsp), dataset ID; MSV000094492. [https://massive.ucsd.edu/ProteoSAFe/dataset.jsp?task=3e301feb87904983b4ccd5d9e949350b]. Source data are provided with this paper.

## Materials availability

Chlamydomonas strains and plasmids are deposited at the Chlamydomonas Resource Center. Arabidopsis mutant lines and markers are deposited at the ABRC.

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

## Acknowledgements

We thank Xiobo Li for sharing early versions of the mutant library; Silvia Ramundo for generous gift of pRAM118 plasmid; Masayuki Onishi for generously sharing Lifeact-NeonGreen Chlamydomonas strain. Heather Cartwright and the Carnegie imaging facility for microscopy support. Members of the Dinneny lab for helpful discussions. We thank Christopher J. Staiger for providing *ABD2:GFP* reporter, David Ehrhardt for providing LTI6:YFP reporter, and Wolf Frommer for providing *GCaMP6* reporter. This project was supported by grants awarded to J.R.D. from the NIH NIGMS (R01 GM123259-01) and a Faculty Scholars grant from the Simons Foundation and Howard Hughes Medical Institute (55108515); grants awarded to M.C.J. from NIH (DP2-GM-119137), NSF (MCB- 1146621 and MCB-1914989) and the Simons Foundation and HHMI (55108535); grants awarded to M.R.S. IOS PGRP No. 2010789 and MCB No. 9143816; Simons Foundation fellowships of the Life Sciences Research Foundation awarded to R.E.J. and J.V.-B; and an EMBO long term fellowship (ALTF 1450-2014) awarded to J.V.-B.

## Author contributions

R.E.J., T.X., and J.V.-B. analyzed RNAseq data: B.B.M., G.A.B.-W., and M.R.S. performed phosphoproteomic experiments; R.E.J., F.F., and J.V.-B. prepared mutant pools, performed treatments, processed samples, and analyzed data with guidance from M.C.J.; T.V. generated

Chlamydomonas fluorescently tagged lines and validated mutant insertion sites; T.V., K.K., and J.V.-B. performed and analyzed Chlamydomonas secondary screenings; L.W. generated Cre10.g45568 reporter line; K.K. provided technical assistance and help with the analysis of the phosphoproteomic dataset; M.G. and J.O. performed initial screen of Arabidopsis orthologous phenotypes; J.V.-B. performed all the other experiments; J.V.B. and J.D. designed the experiments and analyze the data; J.V.-B. wrote the initial draft of the manuscript; J.R.D. and J.V.-B. wrote the final version of the manuscript with input from all authors.

## Competing interests

The authors declare no competing interests.
