## [Peer Review File · Nature Communications]

Reviewers' comments:

Reviewer #1 (Remarks to the Author):

The manuscript by Vilarrasa-Blasi et al. reports the characterization of cellular components involved in responses to osmotic stress in the green alga *Chlamydomonas reinhardtii* and the land plant *Arabidopsis thaliana*. Given the limited knowledge of these processes in photosynthetic organisms, the results provide evidence for evolutionary conservation of some pathways within the green lineage.

The authors first examined the response of *Chlamydomonas* to osmotic stress (100 mM NaCl or 300 mM mannitol) by transcriptomic and phosphoproteomic analyses, contrasting responsive genes/proteins with the behavior of orthologs in *Arabidopsis* and *Saccharomyces cerevisiae* (previously characterized under similar stress conditions) (Fig. 1).

The authors also carried out forward genetic screens to identify genes with roles in osmotic stress, by using a barcoded genome-wide mutant library in *Chlamydomonas reinhardtii*. They identified 76 genes, which when mutated resulted in a growth defect under NaCl, mannitol, polyethylene glycol, or hypo-osmotic stress (Fig. 1). A secondary screen was performed with 140 mutants, resulting in the validation of 55% of the hits (although the criteria used to select these candidate mutant genes were not clearly outlined in the manuscript). The validated list includes several genes previously implicated in osmotic stress responses in *Arabidopsis* (such as CreOSCA1 and CreMSL) as well as 24 genes lacking functional annotation and therefore renamed as Osmotic growth defective in *Chlamydomonas* (OSMO).

Vilarrasa-Blasi et al. then examined in more detail the role(s) of five genes with mutants exhibiting osmotic phenotypes in both *Chlamydomonas* and *Arabidopsis*, including MAPKK (Cre08.g384900), KCN11/GORK (Cre06.g278111), PAT (Cre06.g277000), PRF (Cre10.g427250) and LEO1 (Cre12.g524950) (Fig. 2). The meticulous characterization of PRF (PROFILIN) mutants, particularly in *Arabidopsis*, led the authors to conclude that the actin cytoskeleton reorganizes upon osmotic stress and defects in this response may result in hypersensitivity to osmotic perturbation (Fig. 3).

In summary, the authors demonstrated that a number of conserved genes are involved in osmotic homeostasis in *Chlamydomonas* and *Arabidopsis*. Moreover, the data suggests that osmotic stress induces a pause in general cellular functions and activates stress response genes, a pattern already observed in several other organisms. Overall, the manuscript is well-written and easy to follow.

Main comments:

1. The authors are reporting a substantial amount of work in this manuscript. However, for many of the identified genes, it is not certain whether they are specifically involved in osmotic stress responses or they participate in general stress responses (which commonly cause a pause in growth). For instance, the authors state that 34 out of 76 high confidence hits in their forward genetic screens in *Chlamydomonas* also showed a growth phenotype in other stress conditions (lines 148-151).

2. Of the five mutants/genes characterized in more detail, two [namely MAPKK (Cre08.g384900) and PRF (Cre10.g427250)] were not reported in the high confidence *Chlamydomonas* hits from the forward genetic screens (Supplementary Table 6) or in the secondary (validation) screens (Supplementary Table 7). Data for PROFILIN is not even shown in the raw data of osmotic mutant screens (Supplementary Table 5). The manuscript implies that these genes were first identified in genetic screens in *Chlamydomonas* and then *Arabidopsis* seedlings with mutants in homologous genes were characterized. Since one of the main points of the paper is that PROFILIN-dependent actin remodeling plays a conserved role in osmotic stress tolerance, it seems important to describe how the *Chlamydomonas* PRF mutant was identified. It would also be informative to know if this mutant is also hypersensitive to other stresses (in addition to osmotic stress).

3. Osmotic stress induces reorganization of the actin cytoskeleton and PRF mutants are hypersensitive to osmotic stress. However, is the PRF phenotype caused by a defect in the dynamics of the actin cytoskeleton? Alternatively, could PRF have another role in tolerance to osmotic stress? What happens to osmotic stress tolerance when cells/roots are treated with drugs that stabilize or depolymerize actin filaments (or in mutants of other genes that affect actin filament dynamics)?

Minor comments:

Fig 1c. In the bottom diagram, several of the gene symbols are not labeled, even though they have orthologs in *Arabidopsis*.

Fig. 2a. The representative growth vessels are not clear. The controls panels appear to be black.

Fig. 3i. This cartoon appears to be premature since the mechanistic role of PRF/actin filaments in osmotic stress responses remains to be characterized.

Supplementary Fig. 2c. The accumulation of proline under osmotic stress in *Chlamydomonas* does not appear to be mentioned in the text.

Supplementary Fig. 3. The pie chart for Fe-deficient orthology does not add up to 100%. It is not clear what is represented by the open bars at the bottom. Are these representing the percentage of orthologous genes?

Supplementary Fig 8a. PRF-Venus appears to have a diffuse localization within the cytosol of *Chlamydomonas* under standard laboratory conditions. Is this expected for an actin binding protein? What happens with PRF localization when the cells are subject to osmotic stress?

Reviewer #2 (Remarks to the Author):

The manuscript describes a large scale genetic screen to identify genes involved in osmotic stress in the alga *Chlamydomonas* that can in turn be used to identify candidate genes for land plant osmotic stress responses. The manuscript represents a very large amount of work, and includes some novel and very interesting findings. The demonstration of the potential of this screening approach to identify novel genes involved with plant stress responses is particularly important. However, the manuscript as written does not feel like a coherent piece of work. I found it difficult to follow and felt that many important aspects of the research were not adequately described or did not refer to previous work in this area. The major claim of the title – uncovering a role for actin in stress acclimation – seems to be well established in the plant community for many years.

The manuscript is broken into four major sections.

In section 1, the transcriptional response of *Chlamydomonas* to osmotic stress is described. A huge amount data is presented in Figure 1A-B, but this is dealt with very briefly, with very little detail provided in the text.

Section 2 is the genetic screen in which mutants sensitive to osmotic stress are detected through a large scale sequencing approach to find under-represented strains within a population of cells. This is a fantastic approach, but the integration of this dataset with the wealth of transcriptomic and

phosphoproteomic data in section 1 seems very poor. It makes it difficult to determine what the major findings are in each of these sections.

Section 3 uses candidate genes that were identified in *Chlamydomonas* to identify 5 genes that also have an osmotic stress phenotype in *Arabidopsis*.

Section 4 characterises the phenotype of the *Arabidopsis* PRF5 profilin mutant on more detail describing how osmotic stress causes a major rearrangement of the actin cytoskeleton. Whilst this is presented as a novel hypothesis (line 68, 265), it is well known that osmotic stress causes a rearrangement of actin within root hairs.

The major emphasis of the manuscript is on how *Chlamydomonas* screens can be used to successfully identify plant genes. This requires the scope of the manuscript to be large, but for me there is just far too much information included for this to work as a single manuscript. The large breadth of the manuscript means that some basic aspects of physiology are not well discussed and there are several instances where important findings are missed or not described, or insufficient details are included to allow interpretation of the data.

Major points

1) Comparison of algal/plant physiology. The manuscript lacks basic information on the nature of osmoregulation in *Chlamydomonas* and land plants that is required to make sense of the findings. The Introduction and Discussion are very short, so no overview or context is provided for the reader. *Chlamydomonas* and *Arabidopsis* exhibit fundamental differences in their osmoregulatory strategies through use of a contractile vacuole to remove excess water or a rigid cell wall to invoke turgor. Moreover, they also utilise Na⁺ in fundamentally different ways. *Chlamydomonas* contains a homologue of the Na⁺/K⁺ pump found in animals, whereas this has been lost in vascular plants. Explanation of these differences in cell physiology is essential for a comparative study of the osmoregulatory strategies between land plants and chlorophytes.

2) Conservation of signalling pathways. The manuscript identifies a number of genes that are proposed to represent conserved signalling components between *Chlamydomonas* and *Arabidopsis*. There is a distinction between identifying a similar protein (e.g. a MAPkinase) in both species and finding two proteins that perform an evolutionary conserved role. In a large scale screen it makes sense to initially identify potential homologues through best hits in similarity searches, but once key candidate proteins have been identified, they need to be examined in more detail (e.g. through phylogenetic approaches) before they can be described as conserved. Similarity searches can often reveal erroneous relationships, so this aspect of further validation is important.

There are several examples in the manuscript where mechanisms are proposed to be conserved when this is clearly not the case (e.g. CDPK, P-type ATPase, see below).

3) Relationship to other studies. Whilst it is not always possible to fully cite other work, there were large sections of this manuscript where no indication was given of previous research in a particular area. For example, the transcriptomic results were not compared to other transcriptomic studies of salt stress in *Chlamydomonas*. The observation of flagella shortening was not compared to previous studies and previous work on the role of actin rearrangement in the response of plants to salt stress was not presented clearly. These omissions make it difficult to judge how the current manuscript contributes to an advance in these fields.

Specific points

L91 The likelihood of differential expression of osmotic stress genes is greater in *Arabidopsis* homologues than yeast. How was this assessed, I cannot find the details in the methods? Is the stress applied to yeast and *Arabidopsis* directly comparable?

L104 This finding does not necessarily suggest an absence of ABA in osmotic stress response in *Chlamydomonas*, but it does demonstrate that aspects of the plant response to osmotic stress that are mediated by ABA are not conserved in *Chlamydomonas*. Previous researchers have shown that *Chlamydomonas* does produce ABA (ref) and contains elements of the ABA signalling pathways (e.g. Wang 2015 <https://doi.org/10.1104/pp.114.247403>)

L106 It needs to be made clear which plants do not have cilia (i.e. angiosperms and some gymnosperms)

L108 the regulation of flagella length by osmolarity has been reported previously (e.g. Solter 1978 *Nature*, Lefebvre 1978 *JCB*)

Supplementary Fig 2C. It's not discussed in the manuscript at all, but why was proline measured as an osmolyte? *Chlamydomonas* primarily accumulates trehalose and glycerol as osmoprotective solutes (e.g. Colina 2020 <https://doi.org/10.1016/j.envexpbot.2020.104261>).

L136 how do these results compare with other transcriptomic studies in *Chlamydomonas* (e.g. Colina 2020; Zhang 2022 doi: 10.3389/fpls.2022.828321; Wang 2018 <https://doi.org/10.3390/ijms19113359>)? These studies identify a number of signalling proteins (MAPK, PP2Cs, PKL1, GUN4) that are activated and suggest that elements of the ABA-mediated signalling pathway are conserved.

L169 The annotation of CDPKs is often incorrect, based on similarity to the kinase domain. The gene described as CALCIUM DEPENDENT PROTEIN KINASE (CPDK) Cre03.g153150 is a protein kinase, but does not have any calcium binding EF-hands and so is not a CDPK.

L179 The authors need to look carefully at the annotation of the P-type ATPases in *Chlamydomonas*, as automatic annotation based on BLAST is often incorrect. Pederson et al *Frontiers Plant Science* (doi: 10.3389/fpls.2012.00031) categorised P-type ATPases from

Chlamydomonas – the authors should cite this and use this annotation. The gene described here as ENDOPLASMIC RETICULUM-TYPE CALCIUM TRANSPORTING ATPASE (ECA, Cre06.g263950), is actually CrNAK1 – a P2C-type ATPase that is the major Na⁺/K⁺ ATPase in Chlamydomonas. It is likely that this protein is responsible for Na⁺ efflux across the plasma membrane, so it makes sense that it is important for NaCl stress!

The really important point here is that chlorophytes have P2C-type Na⁺ pump, whereas these have been lost in land plants. This is a really important mechanistic difference between green algae and land plants in their response to Na⁺ stress.

L191 The authors that it is was not surprising that only five of the mutant strains exhibiting an osmotic stress phenotype are found in their extensive transcriptomic analysis. Can the authors explain why this is unsurprising? I would have thought that many genes playing a fundamental role in the osmotic stress response would be regulated transcriptionally.

L215 How were the homologues of these genes identified in Arabidopsis? Is it based on best hits in sequence similarity searches? This needs to be explained.

L265 The authors state that ‘We therefore hypothesized that osmotic stress may alter actin cytoskeleton dynamics’. It is well known that osmotic stress causes a major rearrangement of the actin cytoskeleton in plant roots (e.g. Wang 2020 doi: 10.1104/pp.20.00480).

P292 Plant roots exhibit a conserved transient increase in cytosolic Ca²⁺ on the addition of NaCl that lasts approx 100s. The calcium imaging performed in figure 3 does not examine this response, but measures calcium spiking over a period of 4 hours. The nature of these spikes is not clear (e.g. amplitude, duration) as they are not shown or defined in the manuscript. It is also not shown whether calcium spikes are present in control roots, so it is not clear whether they are associated with NaCl treatment. The authors previously demonstrated that the FERONIA receptor kinase is responsible for generating calcium spikes in Arabidopsis roots in response to salt stress. Although they cite this paper in the manuscript, it is not explained that this is the calcium response being examined here. I suggest they present the data in the current manuscript in a similar format to the FERONIA paper, because it is currently not possible to interpret their data on the information provided.

The images in Fig 3E appear to show a substantial increase in basal (resting) cytosolic Ca²⁺ concentrations over a period of 1 hour in both wild type and prf5 roots, that it is more pronounced in prf5. This does not make sense and is not described in the text. It is likely that the images in Fig 3E are actually maximum intensity projections representing accumulated spiking activity over time but this is not stated. Overall, the calcium imaging data is poorly presented and requires more information for it to be included in the manuscript.

Reviewer #3 (Remarks to the Author):

Vilarrasa-Blasi et al. Identification of green lineage osmotic stress pathways 1 uncovers a role for actin during stress acclimation

Authors use a time-course -omics study in *Chlamydomonas* to identify genes that are responsive and essential for the response to osmotic stress and salinity. They are then testing the homologues of these genes in *Arabidopsis* as landplant model and can confirm some of the cellular phenotypes, making them conclude that sensing of osmotic stress signalling is conserved from the Chlorophyta to the land plants.

While the amount of work is impressive and the data sets are of good quality, the gain of knowledge on function remains a bit limited. It is rather a collection of candidates, less a real attempt to understand mechanisms, which may be tribute to the very generalistic approach that does not appreciate evolutionary differences. For instance, it would have been interesting to learn more about the role of profilin 5 for the function of contractile vacuole in *Chlamydomonas*, since actin plays an essential role here which is quite certainly impacted during osmotic stress doi: 10.1007/s00709-017-1123-y. Likewise, it would have been interesting to expand a bit on the differences between salinity and osmotic stress alone, since salinity is a composite stress, where an osmotic component is accompanied by a ionic component. The choice of the conditions (concentrations) could have been more deliberate, such that the osmotic challenge would be comparable. The reorganisation of actin in response to osmotic stress has been addressed in the past, it would have been fair to quote the literature, the findings are not that novel as they are presented.

Overall, I support publication, but would suggest that the authors rewrite the text by 1) giving justice to the literature on the cellular responses to osmotic stress, especially of the actin cytoskeleton, 2) be more careful in negating essential differences between *Chlamydomonas* and multicellular land plants, 3) coming up with a working hypothesis what the actin remodelling actually means.

REVIEWERS NATURE COMM (12_2022)

Reviewers' comments:

Reviewer #1 (Remarks to the Author):

The manuscript by Vilarrasa-Blasi et al. reports the characterization of cellular components involved in responses to osmotic stress in the green alga *Chlamydomonas reinhardtii* and the land plant *Arabidopsis thaliana*. Given the limited knowledge of these processes in photosynthetic organisms, the results provide evidence for evolutionary conservation of some pathways within the green lineage.

The authors first examined the response of *Chlamydomonas* to osmotic stress (100 mM NaCl or 300 mM mannitol) by transcriptomic and phosphoproteomic analyses, contrasting responsive genes/proteins with the behavior of orthologs in *Arabidopsis* and *Saccharomyces cerevisiae* (previously characterized under similar stress conditions) (Fig. 1).

The authors also carried out forward genetic screens to identify genes with roles in osmotic stress, by using a barcoded genome-wide mutant library in *Chlamydomonas reinhardtii*. They identified 76 genes, which when mutated resulted in a growth defect under NaCl, mannitol, polyethylene glycol, or hypo-osmotic stress (Fig. 1). A secondary screen was performed with 140 mutants, resulting in the validation of 55% of the hits (although the criteria used to select these candidate mutant genes were not clearly outlined in the manuscript). The validated list includes several genes previously implicated in osmotic stress responses in *Arabidopsis* (such as CreOSCA1 and CreMSL) as well as 24 genes lacking functional annotation and therefore renamed as Osmotic growth defective in *Chlamydomonas* (OSMO).

Vilarrasa-Blasi et al. then examined in more detail the role(s) of five genes with mutants exhibiting osmotic phenotypes in both *Chlamydomonas* and *Arabidopsis*, including MAPKK (Cre08.g384900), KCN11/GORK (Cre06.g278111), PAT (Cre06.g277000), PRF (Cre10.g427250) and LEO1 (Cre12.g524950) (Fig. 2). The meticulous characterization of PRF (PROFILIN) mutants, particularly in *Arabidopsis*, led the authors to conclude that the actin cytoskeleton reorganizes upon osmotic stress and defects in this response may result in hypersensitivity to osmotic perturbation (Fig. 3).

In summary, the authors demonstrated that a number of conserved genes are involved in osmotic homeostasis in *Chlamydomonas* and *Arabidopsis*. Moreover, the data suggests that osmotic stress induces a pause in general cellular functions and activates stress response genes, a pattern already observed in several other organisms. Overall, the manuscript is well-written and easy to follow.

We thank the reviewer for acknowledging the impact of the present manuscript. We would like to point out that supplementary table S7 contains the criteria used to select candidates for the secondary screen, see column “Statistical criteria used to select candidate”. See material and methods for statistical analysis used to select candidates.

Main comments:

Comment 1.1: The authors are reporting a substantial amount of work in this manuscript. However, for many of the identified genes, it is not certain whether they are specifically involved in osmotic stress responses, or they participate in general stress responses (which commonly cause a pause in growth). For instance, the authors state that 34 out of 76 high confidence hits in their forward genetic screens in *Chlamydomonas* also showed a growth phenotype in other stress conditions (lines 148-151).

Response 1.1: The reviewer highlights an important strength of our dataset and experimental system, which is that we can directly compare the candidates identified in our study with those identified in a previous study of ours that tested many additional conditions. Last year we released the largest gene-phenotype database in a photosynthetic organism (Fauser et al. 2022) with 121 different environmental conditions examined using *Chlamydomonas reinhardtii* and the same mutant library used in this manuscript. We used the database to query the putative phenotype of our osmotic stress mutants under other environmental conditions (supplementary table 6). Comparing our high confidence hits with the high confidence hit list generated previously (Fauser et al. 2022) found that 55% of our osmotic-associated genes show a specific growth defect only under osmotic stress while 34 out of our 76 hits, 45%, have a growth phenotype in at least one of the 121 conditions previously examined (Supplementary Table 6, phenotypes in other conditions). We reanalyzed these data to get further insights into the molecular crosstalk between osmotic stress and other pathways and we found; (i) 21% (7 out of 34) of the high confidence hits are also sensitive to DNA damage-related conditions or chemical stress; (ii) 15% (5 out of 34) of the osmosensitive hits are sensitive to paromomycin, a drug that inhibits protein synthesis by interacting with ribosomal subunits; (iii) 12% (4 out of 34) of the 36 high confidence hits have a growth defective phenotype or photosynthesis related phenotype. These results identified potential molecular components mediating crosstalk between osmotic stress and other biological processes such as DNA damage, chemical stress, protein biosynthesis, growth, and photosynthesis.

We added this analysis to Supplementary Table 6 (Phenotypes in other conditions).

In addition, we added a brief description of these results to the main text, lines 214-220.

Comment 1.2: Of the five mutants/genes characterized in more detail, two [namely MAPKK (Cre08.g384900) and PRF (Cre10.g427250)] were not reported in the high confidence

Chlamydomonas hits from the forward genetic screens (Supplementary Table 6) or in the secondary (validation) screens (Supplementary Table 7). Data for PROFILIN is not even shown in the raw data of osmotic mutant screens (Supplementary Table 5). The manuscript implies that these genes were first identified in genetic screens in Chlamydomonas and then Arabidopsis seedlings with mutants in homologous genes were characterized. Since one of the main points of the paper is that PROFILIN-dependent actin remodeling plays a conserved role in osmotic stress tolerance, it seems important to describe how the Chlamydomonas PRF mutant was identified. It would also be informative to know if this mutant is also hypersensitive to other stresses (in addition to osmotic stress).

Response 1.2: We apologize for oversimplifying the process for screening the mutant library and for not specifying the process by which each mutant was identified. The Chlamydomonas gene PRF is a very small gene in Chlamydomonas and is represented by a single allele in the mutant library. Because of this, the gene did not reach the threshold required to make the high confidence list, which requires that at least 3 mutant alleles show a growth defect. PRF was identified during preliminary screening of the mutant library using criteria based solely on growth rate. We picked several mutants and validated them independently by measuring growth in culture (final growth experiments figure 2). The MAPKK gene was also identified through a similar analysis.

We have now added these data in the Supplementary figure 6 and have included text in the results section and the materials and methods section describing this initial screening procedure.

We added the following text in the material and methods, lines 1058-1062.

“PRF and MAPKK genes are small genes that did not reach the threshold required for the high confidence list, which required at least 3 mutant alleles with a growth defect. The mutant phenotypes of PRF and MAPKK were identified during the initial screens of the mutant library based only on mutant growth rate of single mutant alleles (see supplementary figure 6 for growth data).”

The Chlamydomonas profilin mutants shows hypersensitivity to LatB, an actin depolymerizing drug. We added this data in Supplementary table 6 with growth sensitivity of profilin mutants upon treatment with LatB (Fauser et al. 2022). Similarly, Arabidopsis profilin mutants are sensitive to LatB treatment (Supplementary figure 10).

We update the main text accordingly, line 347-348.

“In addition, *prf5* mutant root growth is hypersensitive to the actin depolymerization drug Latrunculin B (Supplementary Fig. 10). Similarly, Chlamydomonas profilin mutants are hypersensitive to LatB treatment (Supplementary table S6).”

Comment 1.3: Osmotic stress induces reorganization of the actin cytoskeleton and PRF mutants are hypersensitive to osmotic stress. However, is the PRF phenotype caused by a defect in the dynamics of the actin cytoskeleton? Alternatively, could PRF have another role in tolerance to osmotic stress? What happens to osmotic stress tolerance when cells/roots are treated with drugs that stabilize or depolymerize actin filaments (or in mutants of other genes that affect actin filament dynamics)?

We agree with the reviewer that this is an important connection and have provided several lines of evidence supporting the role of actin dynamics as being the primary target for the osmotic stress functions of PRF. We find that short-term depolymerization of actin filaments via treatment with LatB increases sensitivity to plasmolysis, similar to the *prf* mutant (Fig 3 g, h). In supplementary figure 13, we showed that 60 min treatment with 5 uM LatB removes actin filaments from the Arabidopsis root tip. Furthermore, we show that the *prf5* mutant only shows a defect in actin organization upon osmotic stress treatment, suggesting that the stress context is necessary to reveal the defect. Further investigation of the relationship between actin organization dynamics and PRF5 function is definitely warranted, and we have now added a brief section to the discussion highlighting outstanding questions that our study raises, but has not yet answered.

We added in the discussion section, line 443-451.

“We propose that the regulatory and mechanical impact of actin in response to osmotic stress can be either direct or indirect. On one hand, elevated osmolarity can directly influence actin filament formation or affect the activity and binding of actin-interacting proteins due to increased intracellular ion concentrations. On the other hand, osmotic stress can alter turgor pressure, subsequently modifying the mechanical properties of the cell wall and cell membrane, which in turn indirectly affects actin dynamics. Further investigation into cellular dynamics under osmotic stress, employing high temporal and spatial microscopy, will enhance our understanding of the role played by the actin cytoskeleton in response to osmotic stress.”

Line 396-398 describes the effect of actin depolymerization drugs to an osmotic stress shock.

“Pretreatment with LatB, an actin depolymerizing drug, followed by an osmotic shock increased the cell shrinkage compared to untreated roots (Supplementary Fig. 13, Fig. 4g, h).”

Minor comments:

Comment 1.4: Fig 1c. In the bottom diagram, several of the gene symbols are not labeled, even though they have orthologs in Arabidopsis.

Response 1.4: We have now updated the table and added the missing gene symbols for Arabidopsis orthologues.

We update Supplementary figure 6.

We reorganize the figures and now figure 1c corresponds to figure 3a, which we updated with the missing gene symbols.

Comment 1.5: Fig. 2a. The representative growth vessels are not clear. The controls panels appear to be black.

Response 1.5: In the images from figure 2a the control panels should appear dark green which corresponds to the growth of the algae without stress. We will be sure that during the conversion to pdf the green color is retained. With the new figure rearrangement now it corresponds figure 3a.

Comment 1.6: Fig. 3i. This cartoon appears to be premature since the mechanistic role of PRF/actin filaments in osmotic stress responses remains to be characterized.

Response 1.6: The goal of figure 3i is to summarize the observation made in figure 3. We want to pinpoint that the actin remodeling under osmotic challenge is mediated by profilin. We added question marks to the cartoon to highlight the direct or indirect effect of osmotic stress to promote actin remodeling. In addition, we extended the figure legend to match the new figure.

We added the following text in the discussion, lines 435-451.

“We show here that osmotic stress causes a PROFILIN-mediated actin reorganization into bundles and cables, which can have both a regulatory and/or mechanical role for cell survival. Disruption of the actin cytoskeleton directly impacts cellulose deposition(Sampathkumar et al. 2013), vacuole morphology(Scheuring et al. 2016), calcium signaling(Bascom, Winship, and Bezanilla 2018; Cárdenas et al. 2008), and ion channel activity (Sasaki, Yui, and Noda 2014; J. Wang et al. 2022). Moreover, the actin cytoskeleton establishes barriers that influence the diffusion of membrane proteins and lipids to regulate signal transduction (Ienne 2006, Kusumi 2005). Defects in actin organization, as revealed in our *prf5* mutant and treatment with an actin depolymerization drug LatB, led to the loss of cellular integrity, more severe plasmolysis and roots showed penetration defects upon germination on hard media, suggesting that impaired mechanical properties (Supplementary Fig. 14). We propose that the regulatory and mechanical impact of actin in response to osmotic stress can be either direct or indirect. On one hand, elevated osmolarity can

directly influence actin filament formation or affect the activity and binding of actin-interacting proteins due to increased intracellular ion concentrations. On the other hand, osmotic stress can alter turgor pressure, subsequently modifying the mechanical properties of the cell wall and cell membrane, which in turn indirectly affects actin dynamics. Further investigation into cellular dynamics under osmotic stress, employing high temporal and spatial microscopy, will enhance our understanding of the role played by the actin cytoskeleton in response to osmotic stress.”

Comment 1.7: Supplementary Fig. 2c. The accumulation of proline under osmotic stress in *Chlamydomonas* does not appear to be mentioned in the text.

Response 1.7: Thanks for the observation in the revised version we now mention the experiments where we quantify proline accumulation. Lines 131-137.

“In agreement with previous transcriptomic analysis upon osmotic stress using different *Chlamydomonas* strains (GY-D55 and CC-503) and solute concentrations (Zhang et al. 2022; N. Wang et al. 2018), we found that *Chlamydomonas* uses Glycerol and Proline as main osmolytes. Interestingly, our time-course data set distinguished between an earlier role of glycerol 15 to 60 minutes after stress, while proline may have a later role, since induction is not observed until 6 hours after stress is applied (Fig 1 a, Supplementary Figure 2 b).”

Comment 1.8: Supplementary Fig. 3. The pie chart for Fe-deficient orthology does not add up to 100%. It is not clear what is represented by the open bars at the bottom. Are these representing the percentage of orthologous genes?

Response 1.8: We apologize for the error. We realized that we did not add the percentage representing orthologs for *S. cerevisiae* and the exact number for all parts of the pie chart. We now added all the percentages and update supplementary figure 3.

“The percentages from the orthology pie charts represent; (left pie chart) genome-wide orthology comparison of the *Chlamydomonas* genome with *S. cerevisiae* and *A. thaliana*. Orange, represent genes present in the *Chlamydomonas* genome with orthologous in *S. cerevisiae*. Brown, represents genes in the *Chlamydomonas* genome with orthologs in *S. cerevisiae* and *A. thaliana*. Blue represents *Chlamydomonas* genes with orthologs in *A. thaliana*. Green represents *Chlamydomonas* genes without orthologs neither in *S. cerevisiae* nor *A. thaliana*. The same colour code was used in all pie charts; (central pie) orthology of *Chlamydomonas* osmotically regulated genes (from our transcriptomes upon osmotic stress) with *S. cerevisiae* and *A. thaliana*; (right pie) orthology of *Chlamydomonas* iron deficiency regulated genes (Urzica et al. 2012) with *S. cerevisiae* and *A. thaliana*.”

“The open bars represent the percentage of genes transcriptionally regulated across different organisms. From our list of Chlamydomonas osmotically regulated genes, 6% of the *S. cerevisiae* orthologous genes are also osmotically regulated under osmotic stress in *S. cerevisiae*. In contrast, 24% of the *A. thaliana* orthologous genes are also osmotically regulated under comparable conditions in *A. thaliana* (Dinney et al. 2008). Using the Same rationale applies for Fe-deficiency.”

To clarify the figure content, we update the figure legend lines 705-716.

“Pie charts representing the percentatge of orthology for the Chlamydomonas genome (left), osmotically regulated genes (central) or Fe regulated genes (right), with *Saccharomyces* and *Arabidopsis*. (left) genome-wide orthology comparison all Chlamydomonas genes with *S. Cerevisiae* and *A. thaliana*. (central) Orthology of differentially expressed genes in the Chlamydomonas osmotic time course; (right) Orthology of differentially expressed genes of Chlamydomonas iron-deficiency response; all othologies are based on reciprocal BLAST ($e\text{-val} > 1E-10$) (see Material and Methods and supplementary table S2). Orange, represent genes present in the Chlamydomonas genome with orthologous in *S.cerevisiae*. Brown, represents genes in the Chlamydomonas genome with orthologs in *S. cerevisiae* and *A. thaliana*. Blue represents Chlamydomonas genes with orthologs in *A. thaliana*. Green represents Chlamydomonas genes without orthologs neither in *S. cerevisiae* nor *A. thaliana*. Bars represent the percentage of genes differentially expressed upon osmotic stress or iron deficiency in Chlamydomonas with orthologous genes in *Saccharomyces* (orange bars) or *Arabidopsis* (blue) responding to the same stress (Dinney et al. 2008; Urzica et al. 2012; Jo et al. 2009). (See Material and Methods and supplementary table 2).”

In addition, we added a section in the Material and Methods, line 979-1010.

“To generate the pie charts in Supplementary Figure 3 genome wide orthology was assessed using reciprocal best BLAST hit (RBH) requiring a E-values $1E-10$. FASTA protein sequences were downloaded from the JGI, Phytozone v 12 (phytozone.jgi.doe.gov), (*Saccharomyces* Genome Database (www.yeastgenome.org) and The Arabidopsis Information Resource (www.arabidopsis.org). Osmotic pie chart was generated with all differentially expressed genes from our transcriptomic experiment (1,456 genes ($FC > 2$, $FDR < 0.01$) and assesing orthology with RBH requiring and E-value $< 1E-10$.

Selected transcriptome datasets from Arabidopsis and yeast were selected to mimic our experiments in Chlamydomonas. The transcriptomic data generated in Arabidopsis was performed with osmolyte concentrations that are not lethal to plant cells, promote growth arrest in short term treatments allowing growth recovery after a short period of acclimation (4 h) (Dinney et al. 2008; Geng et al. 2013), similarly to what we report for our treatments (Supplementary figure 1 a-b). Additionally, the high spatial resolution of the transcriptomic experiments in Arabidopsis

provide a greater number of deregulated genes than are zone specific(Dinnyeny et al. 2008), not identified in previous studies. The transcriptomic data generated in Arabidopsis was performed with osmolyte concentrations that are not lethal to plant cells, promote growth arrest in short term treatments allowing growth recovery after a short period of acclimation(4 h)(Dinnyeny et al. 2008; Geng et al. 2013), similarly to what we report for our treatments (Supplementary figure 1 a-b). Additionally, the high spatial resolution of the transcriptomic experiments in Arabidopsis provide a greater number of deregulated genes than are zone specific(Dinnyeny et al. 2008), not identified in previous studies.

Transcriptome responses to osmotic stress is highly dynamic in yeast and varies with concentrations, yeast strain and time. We used transcriptome dataset performed with yeast strain BY4741(Melamed, Pnueli, and Arava 2008), with deregulated genes previously reported in other yeast strains and osmotic treatments(Posas et al. 2000; Hirasawa et al. 2006). The yeast iron dataset used(Jo et al. 2009) uncover a complete list of genes regulated upon iron deficiency and gives a global perspective of cellular processes involved in iron metabolism(Jo et al. 2009). Chlamydomonas iron deficiency transcriptomes from(Urzica et al. 2012) were grown in the same media used for our osmotic treatments (TAP), and growing conditions (light regime), thus being the most comparable conditions to our experiments. The bar size represents the number of Chlamydomonas osmotically or iron regulated genes with an orthologous gene either in yeast or Arabidopsis. The color filled within the bars represent the fraction of genes regulated in yeast (orange) or Arabidopsis (blue). As an example, from the total osmotically regulated genes in Chlamydomonas 6% are osmotically regulated in yeast.”

Comment 1.9: Supplementary Fig 8a. PRF-Venus appears to have a diffuse localization within the cytosol of Chlamydomonas under standard laboratory conditions. Is this expected for an actin binding protein? What happens with PRF localization when the cells are subject to osmotic stress?

Response 1.9: Profilin is localized to the cell membrane and cytosol in yeast cells (Ostrander, Gorman, and Carman 1995). The membrane localization is dependent on the affinity of profilin for phosphatidylinositol (Ostrander, Gorman, and Carman 1995). PRF:YFP in Arabidopsis is localized at the plasma membrane and cytosol (Supplementary figure 12). Previously, profilin was localized to the cytoplasm using Immunofluorescence localization in Chlamydomonas cells (Kovar et al. 2001). (Kovar et al. 2001) described that Chlamydomonas profilin has low affinity for poly-L-proline and for phosphatidylinositol which suggest its low affinity for the cell membranes. Treatment of osmotic stress downregulates the expression of PRF (supplementary figure 12).

We update the text accordingly lines 373-377.

“To localize PRF5 we generated a PRF5 fluorescent tag reporter driven by its own promoter. *ProPRF5:PRF5:GFP* is expressed in root hair cells of the elongation zone and differentiation zone

as well as the last two layers of the lateral root cap, suggesting that the loss of integrity in one cell type causes defects in tissue integrity across the epidermis (Supplementary Fig. 12, Fig. 2c, e). Interestingly, CrPRF has low affinity for phosphatidylinositol (Kovar et al. 2001) having low affinity for cell membranes and localizes in the cytoplasm (Supplementary figure 8a). Time-course imaging of the expression of ProPRF5:PRF5:GFP upon osmotic stress revealed the downregulation of its expression during the initial hours after treatment (Supplementary Fig. 12).”

Reviewer #2 (Remarks to the Author):

The manuscript describes a large scale genetic screen to identify genes involved in osmotic stress in the alga *Chlamydomonas* that can in turn be used to identify candidate genes for land plant osmotic stress responses. The manuscript represents a very large amount of work, and includes some novel and very interesting findings. The demonstration of the potential of this screening approach to identify novel genes involved with plant stress responses is particularly important. However, the manuscript as written does not feel like a coherent piece of work.

Comment 2.1: I found it difficult to follow and felt that many important aspects of the research were not adequately described or did not refer to previous work in this area. The major claim of the title – uncovering a role for actin in stress acclimation – seems to be well established in the plant community for many years.

Response 2.1: We rewrote several sections of the manuscript add the missing references for the previous studies showing that osmotic stress promotes changes in actin dynamics (see below). Furthermore, we modified the title to “Multi-omics analysis of green lineage osmotic stress pathways unveils crucial roles of different cellular compartments”, which more accurately conveys the main message of the manuscript.

Comment 2.2: The manuscript is broken into four major sections.

In section 1, the transcriptional response of *Chlamydomonas* to osmotic stress is described. A huge amount data is presented in Figure 1A-B, but this is dealt with very briefly, with very little detail provided in the text.

Response 2.2: We updated the section and provided more details both in the main manuscript and in Material and methods.

We added detail on the osmolytes concentrations used, Lines 127-128.

“We used treatments that substantially reduce cell growth but are non-lethal (100 mM NaCl and 300 mM Mannitol, Supplementary Fig. 1a-b).”

We added comparisons with recently published transcriptomes upon osmotic stress in *Chlamydomonas*. Lines 131-137.

“In agreement with previous transcriptomic analysis upon osmotic stress using different *Chlamydomonas* strains (GY-D55 and CC-503) and solute concentrations (Zhang et al. 2022; N. Wang et al. 2018), we found that *Chlamydomonas* uses Glycerol and Proline as main osmolytes. Interestingly our time-course distinguished between an earlier role of glycerol 15 to 60 minutes after stress, while proline may have a later role, since induction is not observed until 6 hours after stress is applied (Fig 1 A, Supplementary Figure 2 C).”

We elaborate on the lack of canonical hormonal regulation in the *Chlamydomonas* response to osmotic stress. Lines 146-157.

“While the abscisic acid (ABA) hormone signaling pathway mediates much of the transcriptional response downstream of osmotic stress perception in *Arabidopsis* (Fujita et al. 2011), and exogenous application of ABA has been shown to increase tolerance to salinity in *Chlamydomonas* (Abu-Ghosh et al. 2021; K. Yoshida et al. 2004), we did not find enrichment of gene annotations associated with ABA-regulated genes nor related cis-regulatory elements (e.g. ABA RESPONSE ELEMENT or G-box) in the promoters of genes regulated by stress (Supplementary Fig 1e-f, Supplementary Table 2) (Gonzalez-Guzman et al. 2012). We hypothesize that the use of high concentrations of ABA in previous studies (Abu-Ghosh et al. 2021; K. Yoshida et al. 2004), ranging from 50 to 500 micromolar ABA, which are significantly higher than the nanomolar range of receptor affinity (Cutler et al. 2010), may induce non-physiological responses in *Chlamydomonas*. This, in concordance with the lack of orthologous genes to the PYL/PYR/RCAR ABA receptors identified in land plants (Merchant et al. 2007; Chunyang Wang et al. 2015) suggests that canonical ABA-mediated osmotic response is not a conserved feature between *Chlamydomonas* and *Arabidopsis*.”

In addition, we describe the unique response of *Chlamydomonas* flagella upon osmotic stress. Lines 158-166.

“A cellular compartment of *Chlamydomonas* cells that is shared with animals and certain plant lineages is the formation of motile cilia. Within land plants, cilia are exclusively produced in sperm cells of all nonseed plants, however, the vast majority of seed plants are non-ciliated (Hodges et al. 2012). We found an enrichment of genes encoding ciliary proteins (Pazour et al. 2005) being differentially expressed in *Chlamydomonas* under osmotic stress (Fisher’s Exact Test $<1E-05$), as well as a substantial reduction in flagellar length upon osmotic stress (Fig 1b, Supplementary Table 2), in agreement with previous reports (Solter and Gibor 1978; Lefebvre et al. 1978). Together, these data demonstrate that osmotic stress induces rapid changes in the *Chlamydomonas*

transcriptional landscape that are likely under the control of signaling pathways distinct from the dominant ABA-dependent pathway of land plants.”

We update the material and methods section, lines 979-1010.

“To generate the pie charts in Supplementary Figure 3 genome wide orthology was assessed using reciprocal best BLAST hit (RBH) requiring a E-values $1E-10$. FASTA protein sequences were downloaded from the JGI, Phytozone v 12 (phytozone.jgi.doe.gov), (Saccharomyces Genome Database (www.yeastgenome.org) and The Arabidopsis Information Resource (www.arabidopsis.org). Osmotic pie chart was generated with all differentially expressed genes from our transcriptomic experiment (1,456 genes ($FC>2$, $FDR<0.01$) and assessing orthology with RBH requiring and E-value $<1E-10$.

Selected transcriptome datasets from Arabidopsis and yeast were selected to mimic our experiments in Chlamydomonas. The transcriptomic data generated in Arabidopsis was performed with osmolyte concentrations that are not lethal to plant cells, promote growth arrest in short term treatments allowing growth recovery after a short period of acclimation(4 h)(Dinny et al. 2008; Geng et al. 2013), similarly to what we report for our treatments (Supplementary figure 1 a-b). Additionally, the high spatial resolution of the transcriptomic experiments in Arabidopsis provide a greater number of deregulated genes than are zone specific(Dinny et al. 2008), not identified in previous studies. The transcriptomic data generated in Arabidopsis was performed with osmolyte concentrations that are not lethal to plant cells, promote growth arrest in short term treatments allowing growth recovery after a short period of acclimation(4 h)(Dinny et al. 2008; Geng et al. 2013), similarly to what we report for our treatments (Supplementary figure 1 a-b). Additionally, the high spatial resolution of the transcriptomic experiments in Arabidopsis provide a greater number of deregulated genes than are zone specific(Dinny et al. 2008), not identified in previous studies.

Transcriptome responses to osmotic stress is highly dynamic in yeast and varies with concentrations, yeast strain and time. We used transcriptome dataset performed with yeast strain BY4741(Melamed, Pnueli, and Arava 2008), with deregulated genes previously reported in other yeast strains and osmotic treatments(Posas et al. 2000; Hirasawa et al. 2006). The yeast iron dataset used(Jo et al. 2009) uncover a complete list of genes regulated upon iron deficiency and gives a global perspective of cellular processes involved in iron metabolism(Jo et al. 2009). Chlamydomonas iron deficiency transcriptomes from(Urzica et al. 2012) were grown in the same media used for our osmotic treatments (TAP), and growing conditions (light regime), thus being the most comparable conditions to our experiments. The bar size represents the number of Chlamydomonas osmotically or iron regulated genes with an orthologous gene either in yeast or Arabidopsis. The color filled within the bars represent the fraction of genes regulated in yeast (orange) or Arabidopsis (blue). As an example, from the total osmotically regulated genes in Chlamydomonas 6% are osmotically regulated in yeast.”

Comment 2.3: Section 2 is the genetic screen in which mutants sensitive to osmotic stress are detected through a large scale sequencing approach to find under-represented strains within a population of cells. This is a fantastic approach, but the integration of this dataset with the wealth of transcriptomic and phosphoproteomic data in section 1 seems very poor. It makes it difficult to determine what the major findings are in each of these sections.

Response 2.3: Due to the limited overlap in candidates across approaches, we view the different sections and methodologies used as revealing different pathways associated with osmotic stress. This limited overlap is entirely expected and in line with similar approaches performed in other model systems.

We now added comparisons of our datasets with published datasets.

Lines 131-137, we compared our transcriptomic time-course with other transcriptomes of *Chlamydomonas* subjected to osmotic stress.

“In agreement with previous transcriptomic analyses using different *Chlamydomonas* strains (GY-D55 and CC-503) and solute concentrations (Zhang et al. 2022; N. Wang et al. 2018), we found that *Chlamydomonas* accumulates glycerol and proline as the main compatible osmolytes during stress. Interestingly, our time-course data suggests an earlier role for glycerol 15 to 60 minutes after stress, while proline may have a later role, since induction of proline metabolic gene transcription is not observed until 6 hours after stress is applied (Fig 1a, Supplementary Figure 2 b).”

Lines 193-199, we compared the overlap of our transcriptomic and phosphoproteomics compared with overlaps in similar approaches in other model systems.

“Overall, we found limited overlap between the candidates identified in our transcriptomic and phosphoproteomic analysis with only 5 genes/proteins shared between the two data sets. This list included a MYB transcription factor, IP 5-P, NAP57, and 2 *Chlamydomonas*-specific proteins (Fig. 1b, Supplementary Table 4). Similarly limited overlap between proteomic and transcriptomic data sets examining the same process has been previously shown in *Arabidopsis* (Stecker, Minkoff, and Sussman 2014) and *Saccharomyces* (Kanshin et al. 2015) and is not entirely unexpected considering the different stress durations used with each assay.”

Lines 275-281 we highlight overlap between our transcriptomics and functional genomic approach.

“Not surprisingly, we also found limited overlap in the gene lists identified through transcriptomics and functional genomics, with just five genes (0.34%, 5 of our functional genomics overlap with

1456 genes differentially regulated upon osmotic stress) exhibiting a significant growth defect and transcriptional response upon osmotic stress: an AP2 transcription factor, SBE, a motor microtubule protein KINESIN 7 (KIN7, Cre13.g568450), a proprotein convertase subtilisin (Cre05.g242100) and Cre17.g735650, a gene without any known domain.”

Comment 2.4: Section 3 uses candidate genes that were identified in *Chlamydomonas* to identify 5 genes that also have an osmotic stress phenotype in *Arabidopsis*.

Comment 2.5: Section 4 characterises the phenotype of the *Arabidopsis* PRF5 profilin mutant on more detail describing how osmotic stress causes a major rearrangement of the actin cytoskeleton. Whilst this is presented as a novel hypothesis (line 68, 265), it is well known that osmotic stress causes a rearrangement of actin within root hairs.

Response 2.5: We introduce background on the effects of osmotic stress in the cytoskeleton reorganization to acknowledge the previous observations.

Lines 350-352

“Osmotic stress promotes actin reorganization in leaf and differentiated root tissues (C. Wang et al. 2010; X. Wang et al. 2020), but the cellular consequences and molecular components controlling this process remain unknown.”

The major emphasis of the manuscript is on how *Chlamydomonas* screens can be used to successfully identify plant genes. This requires the scope of the manuscript to be large, but for me there is just far too much information included for this to work as a single manuscript. The large breadth of the manuscript means that some basic aspects of physiology are not well discussed and there are several instances where important findings are missed or not described, or insufficient details are included to allow interpretation of the data.

Major points

Comment 2.6: Comparison of algal/plant physiology. The manuscript lacks basic information on the nature of osmoregulation in *Chlamydomonas* and land plants that is required to make sense of the findings. The Introduction and Discussion are very short, so no overview or context is provided for the reader. *Chlamydomonas* and *Arabidopsis* exhibit fundamental differences in their osmoregulatory strategies through use of a contractile vacuole to remove excess water or a rigid cell wall to invoke turgor. Moreover, they also utilise Na⁺ in fundamentally different ways. *Chlamydomonas* contains a homologue of the Na⁺/K⁺ pump found in animals, whereas this has been lost in vascular plants. Explanation of these differences in cell physiology is essential for a comparative study of the osmoregulatory strategies between land plants and chlorophytes.

Response 2.6: We modified the introduction and increase the text in the different sections to give more background and highlight the differences between Chlamydomonas and Arabidopsis.

Introduction, lines 79-92

“We sought to advance our understanding of osmotic signaling pathways in photosynthetic organisms by establishing the unicellular alga Chlamydomonas (*Chlamydomonas reinhardtii*) as a system that allows for rapid genetic and molecular investigation of the process. During the dominant phase of the lifecycle, Chlamydomonas exists as a haploid single cell, which facilitates the characterization of genetic mutants (Merchant et al. 2007). Recent large-scale characterization of mutant collections for growth defects under diverse stresses have demonstrated the importance of genes with homologues in animal and land-plant lineages (Fauser et al. 2022). Being a freshwater alga, structural and physiological differences with land plants also exist, such as the lack of a rigid cell wall to counteract cellular turgor generated by intracellular solutes (Wolf 2022), or the presence of an ancient contractile vacuole used to expel excess water out of the cell (Komsic-Buchmann, Wöstehoff, and Becker 2014). Furthermore, while initial responses to osmotic stress in land plants involve Ca²⁺-mediated signaling (Feng et al. 2018), Chlamydomonas lacks several Ca²⁺ channels present in algae and animal cells such as Voltage-dependent Ca²⁺ channels (VDCCs), the transient receptor potential (TRP) channels and the inositol triphosphate receptor (Ip3R), which may be important in mediating physiological responses to the environment (Wheeler and Brownlee 2008; Verret et al. 2010; Edel and Kudla 2015).”

Results, lines 252-262

“Eukaryotic cells use two systems to actively extrude intracellular toxic sodium: Cation/H⁺ antiporters (Masrati et al. 2018) (CPAs) and Na⁺-P-ATPases (Kumari and Rathore 2020). Our screens uncovered sodium chloride sensitivity to; (i) a conserved CPA transporter SODIUM/HYDROGEN EXCHANGER (NHX, Cre01.g034150) previously describe in plants (Yokoi et al. 2002) and fungi (Nass and Rao 1998) to play a role sodium toxicity; and (ii) SODIUM/POTASSIUM-EXCHANGING ATPase (CrNAK1) (Cre06.g263950) (C. N. S. Pedersen et al. 2012), a P2C Na⁺/K⁺-ATPase lost at the base of the land plant lineage. Only two chlorophytes retain NAK1 transporters, a marine algae (*Ostreococcus tauri*) and Chlamydomonas (fresh water), suggesting that the high sensitivity of land plants to Na⁺ arose from the loss of an effective system to remove intracellular Na⁺ (C. N. S. Pedersen et al. 2012). Our screens have osmolyte specificity as shown by the number of membrane transporters necessary for Chlamydomonas growth under NaCl, since these transporters were not identified in the other conditions screened.”

We rewrote the entire discussion section, Lines 402-451.

“Across *Chlamydomonas*, *Saccharomyces* and *Arabidopsis* we have observed common transcriptional signatures to reduce investment in growth and to redirect resources towards osmotic homeostasis. Importantly, canonical ABA-mediated osmotic response is not conserved between land plant and algae. Not surprisingly our systems biology approach yielded complementary insight across approaches but with low overlap between candidates identified through transcriptomic, phosphoproteomic and functional genomic, as was previously described for other model systems and biological processes (Giaever et al. 2002). The transcriptomic response uncovered the dominant non-hormonal nature of the *Chlamydomonas* osmotic response compared to land plants, thus providing unique insight into these alternative stress responses. Our phosphoproteomic approach suggested a substantial role for phosphatases in the regulation of immediate osmotic-stress induced postranslational modifications, similar to land plants (T. Yoshida, Mogami, and Yamaguchi-Shinozaki 2014), while our functional genomic approach led to the identification of dozens of new genes with confirmed functions in osmotic stress, clear limitations were also revealed. The lack of multiple alleles for all genes and the limited representation of small genes in the mutant library used (Fauser et al. 2022) makes it difficult to assess the genetic role of all genes in the genome. The generation of future functional genomic tools in suitable photosynthetic organisms holds the potential to significantly accelerate our understanding of fundamental biological mechanisms, as shown in other organisms (Scherens and Goffeau 2004). Nevertheless, the scale at which this study characterizes the osmotic stress pathway provides an extensive resource for hypothesis testing and refining the search space for candidate genes and proteins when combined with other large-scale methods.

Land plants use cation/H⁺ antiporters to detoxify intracellular sodium (Isayenkov et al. 2020) and lack P2C Na⁺/K⁺ type ATPases present in *Chlamydomonas* and *Ostreococcus*, species of fresh and marine algae respectively (C. N. S. Pedersen et al. 2012). The presence of both detoxifying mechanisms allows *Chlamydomonas* to rapidly adapt to seawater (Lachapelle, Bell, and Colegrave 2015), while many vascular plants, show high sensitivity to elevated Na⁺ in the soil (Maathuis 2014). Halophytes, which are plants tolerant to high-sodium soils, represent an exception as they have evolved distinct mechanisms to handle intracellular osmolytes (van Zelm, Zhang, and Testerink 2020). Recent identification of marine *Chlamydomonas* support the strong sodium detoxifying mechanism present in this algae (Carrasco Flores et al. 2021), and suggest that algae P2C Na⁺/K⁺-ATPase can be used to engineer salt tolerant land plants as previously hypothesized (J. T. Pedersen and Palmgren 2017). *Chlamydomonas* uses the contractile vacuole, an ancient organelle present in freshwater protists without cell walls, to remove excess intracellular water under hyposmotic stress (Komsic-Buchmann, Wöstehoff, and Becker 2014). We found a conserved role for an outward rectifying K⁺ channel that underwent an evolutionary relocation from the contractile vacuole in *Chlamydomonas* (KCN11) (Xu et al. 2016) to the plasma membrane in *Arabidopsis* (GORK) (Eisenach et al. 2014).”

Comment 2.7: Conservation of signaling pathways. The manuscript identifies a number of genes that are proposed to represent conserved signaling components between *Chlamydomonas* and

Arabidopsis. There is a distinction between identifying a similar protein (e.g. a MAPkinase) in both species and finding two proteins that perform an evolutionary conserved role. In a large scale screen it makes sense to initially identify potential homologues through best hits in similarity searches, but once key candidate proteins have been identified, they need to be examined in more detail (e.g. through phylogenetic approaches) before they can be described as conserved. Similarity searches can often reveal erroneous relationships, so this aspect of further validation is important. There are several examples in the manuscript where mechanisms are proposed to be conserved when this is clearly not the case (e.g. CDPK, P-type ATPase, see below).

Response 2.7: We further examined the phylogenetic relationships of the five genes we describe with a conserved phenotype between *Chlamydomonas* and *Arabidopsis*.

Chlamydomonas contains a single Profilin, and the gene family expanded to five members in *Arabidopsis* as shown in previous phylogenies (Pandey and Chaudhary 2017) performed across the green lineage. *Chlamydomonas* PAT gene family contains 9 PAT genes and *Arabidopsis* has 24 members (X. Yuan et al. 2013). The phylogenetic relationship between Cre06.g277000 (CrePAT5) and At4g24630 (AtPAT8) was previously described (X. Yuan et al. 2013). *Chlamydomonas* contains 14 potassium channels (Xu et al. 2016), having 10 voltage-gated K⁺ channels (Gomez-Porras et al. 2012). Cre06.g278111 (KCN11) has been phylogenetically associated with SKOR/GORK gene family (Xu et al. 2016). GORK/SKOR gene family arose from recent Brassica-specific gene duplication (Harris et al. 2020; Dreyer et al. 2021).

Cre03.g194100 annotated as a MAPKKK and its ortholog AT2g31010 is annotated as a protein kinase belonging to the MAPKKK family, Raf subfamily (Raf13) (Jonak et al. 2002). Phylogenetic relationships have been established between them previously (González-Coronel, Rodríguez-Alonso, and Guevara-García 2021). Cre12.g524950 encodes for a 50S ribosome-binding GTPase activity located in the chloroplast (Mackinder et al. 2017) and our data. Best hit reciprocal blast identified as the closest ortholog with E-value 5E-61 At2g22870 encoding for P-loop containing nucleoside triphosphate hydrolase superfamily protein. Despite the high sequence similarity, we can not conclude that both genes are phylogenetically related.

We update the text to provide an overview of the evolutionary trajectory of these gene families. Material and methods lines 1074-1083.

“Phylogenetic relationships between *Chlamydomonas* genes and *Arabidopsis* orthologs revealed that (i) PRF, PAT gene families expanded during evolution (PRF; single member in *Chlamydomonas* [Cr], five members in *Arabidopsis* [At] (Pandey and Chaudhary 2017); PAT from 9 to 24 members (X. Yuan et al. 2013) (ii) MAPKKK gene family have a greater number of members in *Chlamydomonas* 108 compared to 89 in *Arabidopsis* (Gomez-Osuna et al. 2020) (iii) Voltage gated K⁺ channels structural diversity in Chlorophyta collapsed with the transition from

aquatic environments to terrestrial environments(Dreyer et al. 2021; Xu et al. 2016). Further characterization of the responses to osmotic stress of complete gene family members will unveil the degree of functional conservation between ortholog genes. Despite the similarity between CreLEO1 and its ortholog AtLEO1 (RBH, E-value 5E-61), further annotation and phylogenetic analysis of this gene family is required to uncover the evolutionary functional conservation.”

Comment 2.8: Relationship to other studies. Whilst it is not always possible to fully cite other work, there were large sections of this manuscript where no indication was given of previous research in a particular area. For example, the transcriptomic results were not compared to other transcriptomic studies of salt stress in *Chlamydomonas*. The observation of flagella shortening was not compared to previous studies and previous work on the role of actin rearrangement in the response of plants to salt stress was not presented clearly. These omissions make it difficult to judge how the current manuscript contributes to an advance in these fields.

Response 2.8: We now add comparison with other transcriptomic manuscripts. Lines 131-147

“Consistent with these data, we found that an osmotic challenge induced an increase in the number of vesicles and starch puncta in *Chlamydomonas* cells (Supplementary Fig. 2a), using fluorescent reporters that mark these compartments (Mackinder et al. 2017). In agreement with previous transcriptomic analysis upon osmotic stress using different *Chlamydomonas* strains (GY-D55 and CC-503) and solute concentrations (Zhang et al. 2022; N. Wang et al. 2018), we found that *Chlamydomonas* uses Glycerol and Proline as main osmolytes. Interestingly our timecourse distinguished between an earlier role of glycerol 15 to 60 minutes after stress, while proline had a role later after stress application 6 hours after stress (Fig 1 A, Supplementary Figure 2 C).”

We reference the flagella shortening upon osmotic stress previously described in *Chlamydomonas* upon osmotic stress. Line 163-164

“We found an enrichment of genes encoding ciliary proteins being differentially expressed in *Chlamydomonas* under osmotic stress (Pazour et al. 2005) (Fisher’s Exact Test<1E-05), as well as a substantial reduction in flagellar length upon osmotic stress (Supplementary Fig. 2a, d, Supplementary Table 2), in agreement with previous reports (Solter and Gibor 1978; Lefebvre et al. 1978)”

Specific points

Comment 2.9: L91 The likelihood of differential expression of osmotic stress genes is greater in *Arabidopsis* homologues than yeast. How was this assessed, I cannot find the details in the methods? Is the stress applied to yeast and *Arabidopsis* directly comparable?

Response 2.9: The description of the pie chart and bar charts in Supplementary Figure 3 was embedded in the figure caption and further supported by Supplementary Table 2. The percentages from the orthology pie charts represent; (left pie chart) genome-wide orthology comparison of *Chlamydomonas* genes with *S. cerevisiae* and *A. thaliana*; (central pie) orthology of *Chlamydomonas* osmotically regulated genes (from our transcriptomes upon osmotic stress) with *S. cerevisiae* and *A. thaliana*; (right pie) orthology of *Chlamydomonas* iron deficiency regulated genes with *S. cerevisiae* and *A. thaliana*. The open bars represent the percentage of genes transcriptionally regulated across different organisms. From our list of *Chlamydomonas* osmotically regulated genes, 6% of the *S. cerevisiae* orthologous genes are also osmotically regulated under osmotic stress in *S. cerevisiae*. In contrast, 24% of the *A. thaliana* orthologous genes are also osmotically regulated under comparable conditions in *A. thaliana*. Same rationale applies for Fe-deficiency.

The transcriptomic datasets selected from yeast and *Arabidopsis* have comparable physiological stresses between the different organisms. The transcriptomic data generated in *Arabidopsis* was performed with osmolyte concentrations that are not lethal to plant cells, promote growth arrest in short term treatments allowing growth recovery after a short period of acclimation(4 h)(Dinny et al. 2008; Geng et al. 2013), similarly to what we report for our treatments (Supplementary figure 1 a-b). Additionally, the high spatial resolution of the transcriptomic experiments in *Arabidopsis* provide a greater number of deregulated genes than are zone specific (Dinny et al. 2008), not identified in previous studies.

Transcriptome responses to osmotic stress is highly dynamic in yeast and varies with concentrations, yeast strain and time. We used transcriptome dataset performed with yeast strain BY4741(Melamed, Pnueli, and Arava 2008), with differentially regulated genes previously reported in other yeast strains and osmotic treatments (Posas et al. 2000; Hirasawa et al. 2006). The yeast iron dataset used (Jo et al. 2009) uncover a complete list of genes regulated upon iron deficiency and gives a global perspective of cellular processes involved in iron metabolism (Jo et al. 2009).

In this updated version we add a section in the material and methods. We detailed the datasets used and how comparable are the experiments performed in different species. Lines 979-1010.

“To generate the pie charts in Supplementary Figure 3 genome wide orthology was assessed using reciprocal best BLAST hit (RBH) requiring a E-values $1E-10$. FASTA protein sequences were downloaded from the JGI, Phytozone v 12 (phytozone.jgi.doe.gov), (*Saccharomyces* Genome Database (www.yeastgenome.org) and The *Arabidopsis* Information Resource (www.arabidopsis.org). Osmotic pie chart was generated with all differentially expressed genes from our transcriptomic experiment (1,456 genes ($FC>2$, $FDR<0.01$) and assessing orthology with RBH requiring and E-value $<1E-10$.

Selected transcriptome datasets from *Arabidopsis* and yeast were selected to mimic our experiments in *Chlamydomonas*. The transcriptomic data generated in *Arabidopsis* was performed

with osmolyte concentrations that are not lethal to plant cells, promote growth arrest in short term treatments allowing growth recovery after a short period of acclimation(4 h)(Dinneny et al. 2008; Geng et al. 2013), similarly to what we report for our treatments (Supplementary figure 1 a-b). Additionally, the high spatial resolution of the transcriptomic experiments in Arabidopsis provide a greater number of deregulated genes than are zone specific(Dinneny et al. 2008), not identified in previous studies. The transcriptomic data generated in Arabidopsis was performed with osmolyte concentrations that are not lethal to plant cells, promote growth arrest in short term treatments allowing growth recovery after a short period of acclimation(4 h)(Dinneny et al. 2008; Geng et al. 2013), similarly to what we report for our treatments (Supplementary figure 1 a-b). Additionally, the high spatial resolution of the transcriptomic experiments in Arabidopsis provide a greater number of deregulated genes than are zone specific(Dinneny et al. 2008), not identified in previous studies.

Transcriptome responses to osmotic stress is highly dynamic in yeast and varies with concentrations, yeast strain and time. We used transcriptome dataset performed with yeast strain BY4741(Melamed et al. 2008), with deregulated genes previously reported in other yeast strains and osmotic treatments (Posas et al. 2000; Hirasawa et al. 2006). The yeast iron dataset used(Jo et al. 2009) uncover a complete list of genes regulated upon iron deficiency and gives a global perspective of cellular processes involved in iron metabolism(Jo et al. 2009). Chlamydomonas iron deficiency transcriptomes from (Urzica et al. 2012) were grown in the same media used for our osmotic treatments (TAP), and growing conditions (light regime), thus being the most comparable conditions to our experiments. The bar size represents the number of Chlamydomonas osmotically or iron regulated genes with an orthologous gene either in yeast or Arabidopsis. The color filled within the bars represent the fraction of genes regulated in yeast (orange) or Arabidopsis (blue). As an example, from the total osmotically regulated genes in Chlamydomonas 6% are osmotically regulated in yeast.”

We further extend the figure legend in Supplementary figure 3. Lines 705-716.

“Pie charts representing the orthology of the Chlamydomonas genome with Saccharomyces and Arabidopsis genomes. (left) genome-wide orthology comparison of all Chlamydomonas genes with *S. cerevisiae* and *A. thaliana*. (central) Orthology of differentially expressed genes in the Chlamydomonas osmotic time course; (right) Orthology of differentially expressed genes of Chlamydomonas iron-deficiency response; all orthologies are based on reciprocal BLAST ($e\text{-val} > 1E-10$) (see Material and Methods and supplementary table S2). Bars represent the percentage of genes differentially expressed upon osmotic stress or iron deficiency in Chlamydomonas with orthologous genes in Saccharomyces (orange bars) or Arabidopsis (blue) responding to the same stress(Dinneny et al. 2008; Urzica et al. 2012; Jo et al. 2009). (See Material and Methods and supplementary table 2).”

Comment 2.10: L104 This finding does not necessarily suggest an absence of ABA in osmotic stress response in *Chlamydomonas*, but it does demonstrate that aspects of the plant response to osmotic stress that are mediated by ABA are not conserved in *Chlamydomonas*. Previous researchers have shown that *Chlamydomonas* does produce ABA and contains elements of the ABA signaling pathways (Wang 2015).

Response 2.10: We change the text and cite the work that shows that despite containing elements of the ABA signaling pathway there are no ABA receptors in *Chlamydomonas*. Lines 146-157.

“While the abscisic acid (ABA) hormone signaling pathway mediates much of the transcriptional response downstream of osmotic stress perception in *Arabidopsis*(Fujita et al. 2011), and exogenous application of ABA has been shown to increase tolerance to salinity in *Chlamydomonas*(Abu-Ghosh et al. 2021; K. Yoshida et al. 2004), we did not find enrichment of gene annotations associated with ABA-regulated genes nor related cis-regulatory elements (e.g. ABA RESPONSE ELEMENT or G-box) in the promoters of genes regulated by stress (Supplementary Fig 1e-f, Supplementary Table 2)(Gonzalez-Guzman et al. 2012). We hypothesize that the use of high concentrations of ABA in previous studies(Abu-Ghosh et al. 2021; K. Yoshida et al. 2004), ranging from 50 to 500 micromolar ABA, which are significantly higher than the nanomolar range of receptor affinity (Cutler et al. 2010), may induce non-physiological responses in *Chlamydomonas*. This, in concordance with the lack of orthologous genes to the PYL/PYR/RCAR ABA receptors identified in land plants(Merchant et al. 2007; Chunyang Wang et al. 2015) suggests that canonical ABA-mediated osmotic response is not a conserved feature between *Chlamydomonas* and *Arabidopsis*.”

Comment 2.11: L106 It needs to be made clear which plants do not have cilia (i.e. angiosperms and some gymnosperms)

Response 2.11: We rewrote the section, Lines 170-171.

“Within land plants, cilia are exclusively produced in sperm cells of all nonseed plants, however, the vast majority of seed plants are non-ciliated(Hodges et al. 2012).”

Comment 2.12: L108 the regulation of flagella length by osmolarity has been reported previously (e.g. Solter 1978 Nature, Lefebvre 1978 JCB)

Response 2.12: We now introduce the citations describing the regulation of flagella length upon osmotic stress, Lines 190-194.

“We found an enrichment of genes encoding ciliary proteins³¹ being differentially expressed in *Chlamydomonas* under osmotic stress (Fisher’s Exact Test<1E-05), as well as a substantial

reduction in flagellar length upon osmotic stress (Fig 1b, Supplementary Table 2), in agreement with previous reports^{32,33}”

Comment 2.13: Supplementary Fig 2C. It’s not discussed in the manuscript at all, but why was proline measured as an osmolyte? Chlamydomonas primarily accumulates trehalose and glycerol as osmoprotective solutes (e.g. Colina 2020 <https://doi.org/10.1016/j.envexpbot.2020.104261>).

Response 2.13: We thank the reviewer for pointing out the missing reference to the proline quantification. (Colina et al. 2020) examined the proteome and metabolome changes upon UV stress suggesting a role for glycerol and trehalose. We were not able to find any protein related to glycerol or trehalose in our phosphoproteomic experiments (Figure 1b, Supplementary table S4). Interestingly, a comparison of our transcriptomic experiments with others performed in Chlamydomonas upon osmotic stress (N. Wang et al. 2018; Zhang et al. 2022) yield glycerol and proline as the main osmoprotectants upon osmotic stress in Chlamydomonas. Importantly our timecourse is able to distinguish between an earlier role of glycerol (15 min to 1 hour) compared to a later role for proline (6 hours) (Figure 1 A).

We updated the text accordingly, Lines 131-137.

“In agreement with previous transcriptomic analyses using different Chlamydomonas strains (GY-D55 and CC-503) and solute concentrations(Zhang et al. 2022; N. Wang et al. 2018), we found that Chlamydomonas accumulates glycerol and proline as the main compatible osmolytes during stress. Interestingly, our time-course data suggests an earlier role for glycerol 15 to 60 minutes after stress, while proline may have a later role, since induction of proline metabolic gene transcription is not observed until 6 hours after stress is applied (Fig 1a, Supplementary Figure 2 b).”

Comment 2.14: L136 how do these results compare with other transcriptomic studies in Chlamydomonas (e.g Colina 2020; Zhang 2022 doi: 10.3389/fpls.2022.828321; Wang 2018). These studies identify a number of signaling proteins (MAPK, PP2Cs, PKL1, GUN4) that are activated and suggest that elements of the ABA-mediated signaling pathway are conserved.

Response 2.14: There are currently a number of different transcriptomic datasets available for Chlamydomonas. Unfortunately, comparisons are hard to make within the resulting datasets because the experiments and analysis were performed using different strains, growth conditions (including media and stress) and methodologies.

Colina et al, 2020 (Colina et al. 2020) use CC-503 cw92 strain cultured in HEPES acetate phosphate medium (HAP) and induce UV stress at 0.1 W m⁻² during 15 min. Subsequently, they

performed GelC-LTQ-Orbitrap MS experiments and analyzed the data using the SEQUEST algorithm. The authors identify a PP2A protein, in control conditions, with homology to an Arabidopsis protein.

Zhang et al 2022(Zhang et al. 2022) use GY-D55 strain cultured in BG-11 medium and induce salt stress with 200 mM NaCl. The authors showed that growth was impaired during the treatment and the cells form palmelloids (aggregations of cells). We performed physiological experiments using our strain CC-4533 to determine the stress that will impair growth without being lethal for Chlamydomonas cells. We determined (i) growth under different concentrations of NaCl, and (ii) the cell viability under different NaCl treatments (Supplementary figure 1 a, b). Using our strain (CC-4533), and growing conditions (Tris-Acetate-Phosphate (TAP) media, with modified trace elements (Kropat et al. 2011) at 22 °C in low continuous light (~15-10 $\mu\text{mol photons m}^{-2} \text{s}^{-1}$, see methods for further details) we found that concentrations above 100 mM NaCl severely impaired cell growth and promote cell death in Chlamydomonas cells.

The reviewer may mistakenly left out the key reference supporting we did not find any reference to ABA, MAPK, PP2Cs, PKL1 nor GUN 4 in the text of the mentioned references.

We now examined the previously published transcriptome data sets upon osmotic stress. In both cases (Zhang et al. 2022; N. Wang et al. 2018) the authors identified glycerol and proline as major osmoprotectants. Our timecourse transcriptomic experiments allow us to distinguish between an early role of glycerol upon osmotic stress and a later role for trehalose.

We updated the text accordingly, Lines 131-137.

“In agreement with previous transcriptomic analyses using different Chlamydomonas strains (GY-D55 and CC-503) and solute concentrations(Zhang et al. 2022; N. Wang et al. 2018), we found that Chlamydomonas accumulates glycerol and proline as the main compatible osmolytes during stress. Interestingly, our time-course data suggests an earlier role for glycerol 15 to 60 minutes after stress, while proline may have a later role, since induction of proline metabolic gene transcription is not observed until 6 hours after stress is applied (Fig 1a, Supplementary Figure 2 b).”

Comment 2.15: L169 The annotation of CDPKs is often incorrect, based on similarity to the kinase domain. The gene described as CALCIUM DEPENDENT PROTEIN KINASE (CPDK) Cre03.g153150 is a protein kinase, but does not have any calcium binding EF-hands and so is not a CDPK.

Response 2.15: For simplicity we used the standard annotation provided by phytozone 13. The current version for the Chlamydomonas genome v5.6 annotates Cre03.g153150 as a Calcium-

dependent protein kinase/Microtubule associated protein 2 kinase with EF hand domains (<https://phytozone-next.jgi.doe.gov/phytomine/report.do?id=143254321&trail=%7c143254321>). We change the text to the correct annotation.

We include the current annotation of phytozone and the text now reads, lines 247-251.

“We identified sodium chloride sensitive phenotypes for signaling components such as kinases: CALCIUM-DEPENDENT PROTEIN KINASE/ MICROTUBULE ASSOCIATE PROTEIN 2 KINASE (CDPK, Cre03.g153150) and MITOGEN-ACTIVATED PROTEIN KINASE KINASE KINASE (MAPKKK, Cre13.g576600), and phosphatase: PROTEIN PHOSPHATASE 2C (PP2C, Cre03.g211073) (Fig. 2a, Supplementary Table 6).”

Comment 2.16: L179 The authors need to look carefully at the annotation of the P-type ATPases in *Chlamydomonas*, as automatic annotation based on BLAST is often incorrect. Pederson et al *Frontiers Plant Science* (doi: 10.3389/fpls.2012.00031) categorised P-type ATPases from *Chlamydomonas* – the authors should cite this and use this annotation. The gene described here as ENDOPLASMIC RETICULUM-TYPE CALCIUM TRANSPORTING ATPASE (ECA, Cre06.g263950), is actually CrNAK1 – a P2C-type ATPase that is the major Na⁺/K⁺ ATPase in *Chlamydomonas*. It is likely that this protein is responsible for Na⁺ efflux across the plasma membrane, so it makes sense that it is important for NaCl stress!

The really important point here is that chlorophytes have P2C-type Na⁺ pump, whereas these have been lost in land plants. This is a really important mechanistic difference between green algae and land plants in their response to Na⁺ stress.

Response 2.16: For simplicity, we used the Phytozone genome annotation that describes Cre06.g263950 as a Sodium/potassium-exchanging ATPase and named it as the closest ortholog gene family in *Arabidopsis*, ECA, which turns out to be misleading. We acknowledge the reviewer's observation and we believe it raises a very important point in our story.

To keep the text coherent with annotation we changed the ECA annotation to SODIUM/POTASSIUM-EXCHANGING ATPase and use acronym CrNAK1. We include in the text the importance of the finding, lines 235-245.

“Eukaryotic cells use two systems to actively extrude intracellular toxic sodium: Cation/H⁺ antiporters(Masrati et al. 2018) (CPAs) and Na⁺-P-ATPases(Kumari and Rathore 2020). Our screens uncovered sodium chloride sensitivity to; (i) a conserved CPA transporter SODIUM/HYDROGEN EXCHANGER (NHX, Cre01.g034150) previously describe in plants (Yokoi et al. 2002) and fungi(Nass and Rao 1998) to play a role sodium toxicity; and (ii) SODIUM/POTASSIUM-EXCHANGING ATPase (CrNAK1) (Cre06.g263950)(C. N. S. Pedersen et al. 2012), a P2C Na⁺/K⁺-ATPase lost at the base of the land plant lineage. Only two chlorophytes retain NAK1

transporters, a marine algae (*Ostreococcus tauri*) and Chlamydomonas (fresh water), suggesting that the high sensitivity of land plants to Na⁺ arose from the loss of an effective system to remove intracellular Na⁺ (C. N. S. Pedersen et al. 2012). Our screens have osmolyte specificity as shown by the number of membrane transporters necessary for Chlamydomonas growth under NaCl, since these transporters were not identified in the other conditions screened.”

We will notify phytozone about the missannotation to change it.

Comment 2.17: L191 The authors that it is was not surprising that only five of the mutant strains exhibiting an osmotic stress phenotype are found in their extensive transcriptomic analysis. Can the authors explain why this is unsurprising? I would have thought that many genes playing a fundamental role in the osmotic stress response would be regulated transcriptionally.

Response 2.17: Genome-wide functional genomic approaches are commonly used in yeast. Genome-wide analysis of a yeast deletion library subjected to osmotic stress was compared to expression profiling against the same osmotic stress (Giaever et al. 2002). The authors found that less than 7% of the genes that exhibited a significant increase in mRNA expression also exhibited a significant decrease in fitness in galactose treatment (Giaever et al. 2002). Additional analysis showed lower correlations for other stresses; pH 8(3%), 1M NaCl (0.88%), 1.5 M Sorbitol (0.34%)(Giaever et al. 2002).

An extreme case is the absence of correlation between transcriptional responses and phenotypic identification upon DNA-damage agents in yeast (Birrell et al. 2002).

Recently genome-wide mutant analysis in Chlamydomonas subjected to a variety of treatment was performed. The correlation of phenotypes was largely unrelated to transcriptional expression correlations (Fauser et al. 2022).

Number genes differentially regulated upon osmotic stress= 1456

Overlap functional genomic and transcriptomics= 5

Percentage of genes with correlation transcriptomic and phenomics= $5/1456 = 0.34\%$

We updated the text accordingly, line 272-285.

“Not surprisingly, we also found limited overlap in the gene lists identified through transcriptomics and functional genomics, with just five genes (0.34%, 5 of our functional genomics overlap with 1456 genes differentially regulated upon osmotic stress) exhibiting a significant growth defect and transcriptional response upon osmotic stress”

Lines 286-289

“Overall, our functional genomics identified new high-confidence genes involved in osmotic stress signaling and corroborates the low overlap between transcriptomics and functional genomics (0.34%) previously reported in yeast for other stresses; pH (3%), 1M NaCl (0.88%), 1.5M Sorbitol (0.34%)(Evans 2015; Giaever et al. 2002; Fauser et al. 2022).”

Comment 2.18: L215 How were the homologues of these genes identified in Arabidopsis? Is it based on best hits in sequence similarity searches? This needs to be explained.

Response 2.18: We update the main text and mention the method used for homology searches. Lines 298-299

“Thus, to identify new osmotic cellular pathways conserved across the green lineage, we grew Arabidopsis seedlings with mutations in genes homologous to Chlamydomonas osmo-sensitive genes identified in our screens (homology based on Reciprocal Best BLAST hit RBH, see material and methods) and analyzed the ability of their roots to acclimate to osmotic stress induced by NaCl or mannitol containing media.”

We update the Material and Methods section 962-995

“To generate the pie charts in Supplementary Figure 3 genome wide orthology was assessed using reciprocal best BLAST hit (RBH) requiring a E-values $1E-10$. FASTA protein sequences were downloaded from the JGI, Phytozone v 12 (phytozone.jgi.doe.gov), (Saccharomyces Genome Database (www.yeastgenome.org) and The Arabidopsis Information Resource (www.arabidopsis.org). Osmotic pie chart was generated with all differentially expressed genes from our transcriptomic experiment (1,456 genes ($FC>2$, $FDR<0.01$) and assessing orthology with RBH requiring and E-value $<1E-10$. Selected transcriptome datasets from Arabidopsis and yeast were selected to mimic our experiments in Chlamydomonas. The transcriptomic data generated in Arabidopsis was performed with osmolyte concentrations that are not lethal to plant cells, promote growth arrest in short term treatments allowing growth recovery after a short period of acclimation(4 h)(Dinneny et al. 2008; Geng et al. 2013), similarly to what we report for our treatments (Supplementary figure 1 a-b).Additionally, the high spatial resolution of the transcriptomic experiments in Arabidopsis provide a greater number of deregulated genes than are zone specific(Dinneny et al. 2008), not identified in previous studies. The transcriptomic data generated in Arabidopsis was performed with osmolyte concentrations that are not lethal to plant cells, promote growth arrest in short term treatments allowing growth recovery after a short period of acclimation(4 h)(Dinneny et al. 2008; Geng et al. 2013), similarly to what we report for our treatments (Supplementary figure 1 a-b). Additionally, the high spatial resolution of the transcriptomic experiments in Arabidopsis provide a greater number of deregulated genes than are zone specific(Dinneny et al. 2008), not identified in previous studies.

Transcriptome responses to osmotic stress is highly dynamic in yeast and varies with concentrations, yeast strain and time. We used transcriptome dataset performed with yeast strain BY4741(Melamed et al. 2008), with deregulated genes previously reported in other yeast strains and osmotic treatments (Posas et al. 2000; Hirasawa et al. 2006). The yeast iron dataset used(Jo et al. 2009) uncover a complete list of genes regulated upon iron deficiency and gives a global perspective of cellular processes involved in iron metabolism(Jo et al. 2009). Chlamydomonas iron deficiency transcriptomes from (Urzica et al. 2012 were grown in the same media used for our osmotic treatments (TAP), and growing conditions (light regime), thus being the most comparable conditions to our experiments. The bar size represents the number of Chlamydomonas osmotically or iron regulated genes with an orthologous gene either in yeast or Arabidopsis. The color filled within the bars represent the fraction of genes regulated in yeast (orange) or Arabidopsis (blue). As an example, from the total osmotically regulated genes in Chlamydomonas 6% are osmotically regulated in yeast.”

Comment 2.19: L265 The authors state that ‘We therefore hypothesized that osmotic stress may alter actin cytoskeleton dynamics’. It is well known that osmotic stress causes a major rearrangement of the actin cytoskeleton in plant roots (e.g. Wang 2020 doi: 10.1104/pp.20.00480).

Response 2.19: We acknowledge the reviewer’s point in the lack of background and citation of the previous literature linking osmotic stress and actin dynamics. Wang et al 2020 describe the role of GL2-VLN1 pathway regulating root hair growth in Arabidopsis primary roots(X. Wang et al. 2020), where both genes are transcriptionally regulated by osmotic stress and subsequently, root hair growth is regulated by osmotic stress. In our study we describe the actin bundling in elongating cells (undifferentiated) compared to root hair cells (totally differentiated). The first observation of osmotic stress mediating actin reorganization in leaf and root tissues was performed by Wang 2010 (C. Wang et al. 2010).

We update the text accordingly and cite previous work related, lines 350-352.

“Profilin is an actin interacting protein that determines the dynamics of the actin cytoskeleton by controlling the rate of polymerization, bundling, and cable formation (Porta and Borgstahl 2012; Funk et al. 2019). Osmotic stress promotes actin reorganization in leaf and differentiated root tissues (C. Wang et al. 2010; X. Wang et al. 2020), but the cellular consequences and molecular components controlling this process remain unknown.”

Comment 2.20: P292 Plant roots exhibit a conserved transient increase in cytosolic Ca²⁺ on the addition of NaCl that lasts approx 100s. The calcium imaging performed in figure 3 does not examine this response, but measures calcium spiking over a period of 4 hours. The nature of these spikes is not clear (e.g. amplitude, duration) as they are not shown or defined in the manuscript. It is also not shown whether calcium spikes are present in control roots, so it is not clear whether

they are associated with NaCl treatment. The authors previously demonstrated that the FERONIA receptor kinase is responsible for generating calcium spikes in Arabidopsis roots in response to salt stress. Although they cite this paper in the manuscript, it is not explained that this is the calcium response being examined here. I suggest they present the data in the current manuscript in a similar format to the FERONIA paper, because it is currently not possible to interpret their data on the information provided.

Response 2.20: Figure 3 provides the number of calcium spikes within the first 4 hours after treatment with 50 mM NaCl. The imaging was performed in the elongation zone of roots transferred to 50 mM NaCl and control experiments transferring roots to MS control media. For each timepoint we took a stack of images and use z-stacks to quantify the number of spikes every hour. For each timepoint and genotype we quantified 4 different roots. The profilin mutant roots show a massive loss of cell viability with 50 mM NaCl treatment similar to the FERONIA manuscript with treatments of 140mM NaCl (Feng et al. 2018). Cell viability is seriously compromised after the first hour in profilin mutants (see figure 2 c-f) for quantification, therefore we focus on the initial calcium spikes.

We update the text to clarify the spiking that we quantify is a result of NaCl treatment and not present in control conditions, line 385.

“Wild-type plants transferred to 50 mM NaCl showed calcium spikes in the early elongation zone with ~40 spikes per hour within the first 5 hours, compared to the lack of such dynamics when transferred to control media.”

We additionally updated the Material and methods description of the quantification, lines 1171-1179.

“Calcium spike quantification was performed at the transition zone in root epidermal cells (~200 μm from the stem cell niche) during a period of 4 hours. Roots were mounted in chambered coverglass (ThermoFisher, 155360) with media supplemented with 50 mM NaCl, 0 hours timepoints corresponds to the first image stack taken, approximately 10 minutes after mounting the sample. Registration was applied to time lapse image stacks using the HyperStackReg plugin. A MaxEntropy Threshold macro was applied, and 8 ROIs (24x24 μm) were selected across a 150 μm area to measure the signal corresponding to calcium spikes. A minimum of 6 pixels was used to exclude sporadic signals. Calcium spikes and their duration were confirmed manually.”

Comment 2.21: The images in Fig 3E appear to show a substantial increase in basal (resting) cytosolic Ca^{2+} concentrations over a period of 1 hour in both wild type and *prf5* roots, that it is more pronounced in *prf5*. This does not make sense and is not described in the text. It is likely that the images in Fig 3E are actually maximum intensity projections representing accumulated spiking

activity over time but this is not stated. Overall, the calcium imaging data is poorly presented and requires more information for it to be included in the manuscript.

Response 2.22: Figure 3E represents z-stack projections of roots expressing GCamp6 for both wild-type plants and *prf-5* mutants, and therefore as the reviewer point out an increase in background is displayed in the figure. The images display in the figure correspond to samples mounted in media containing NaCl at the starting of the imaging session (time 0) (aprox 10 min after mounting) and 1 hour after initial timepoint.

In light of our results, we conclude that the *prf-5* mutants have an increase in the Calcium spiking compared to wild-type plants carrying the same calcium sensor GCamp6.

We update the figure legend in figure 3 E to clarify that the images are z-stacks without subtracting the 6 pixel background images used in the final quantifications. Lines 611-612.

“16 colors LUT confocal images of primary root elongation zone expressing GCamp6, and *prf5-1*, *GCamp6* in control conditions (0) or upon transfer to 50 mM NaCl for ½ hour and 1 hour. Images represent z-projection in each timepoint. Scale bar = 50 µm”

We update the material and methods section accordingly, lines 1173-1175.

“Roots were mounted in chambered coverglass (ThermoFisher, 155360) with media supplemented with 50 mM NaCl, 0 hours timepoints corresponds to the first image stack taken, approximately 10 minutes after mounting the sample. Registration was applied to time lapses using the HyperStackReg plugin. A MaxEntropy Threshold was applied and 8 ROIs (24x24 µm) were selected across a 150 µm area to measure the signal corresponding to calcium spikes. A minimum of 6 pixels was used to exclude sporadic signals. Calcium spikes and their duration were confirmed manually.”

Reviewer #3 (Remarks to the Author):

Vilarrasa-Blasi et al. Identification of green lineage osmotic stress pathways 1 uncovers a role for actin during stress acclimation

Authors use a time-course -omics study in *Chlamydomonas* to identify genes that are responsive and essential for the response to osmotic stress and salinity. They are then testing the homologues of these genes in *Arabidopsis* as landplant model and can confirm some of the cellular phenotypes, making them conclude that sensing of osmotic stress signalling is conserved from the Chlorophyta to the land plants.

While the amount of work is impressive and the data sets are of good quality, the gain of knowledge on function remains a bit limited. It is rather a collection of candidates, less a real

attempt to understand mechanisms, which may be tribute to the very generalistic approach that does not appreciate evolutionary differences. For instance, it would have been interesting to learn more about the role of profilin 5 for the function of contractile vacuole in *Chlamydomonas*, since actin plays an essential role here which is quite certainly impacted during osmotic stress doi: 10.1007/s00709-017-1123-y. Likewise, it would have been interesting to expand a bit on the differences between salinity and osmotic stress alone, since salinity is a composite stress, where an osmotic component is accompanied by a ionic component. The choice of the conditions (concentrations) could have been more deliberate, such that the osmotic challenge would be comparable. The reorganisation of actin in response to osmotic stress has been addressed in the past, it would have been fair to quote the literature, the findings are not that novel as they are presented.

Overall, I support publication, but would suggest that the authors rewrite the text by

Comment 3.1: giving justice to the literature on the cellular responses to osmotic stress, especially of the actin cytoskeleton,

Response 3.1: We appreciate the author's comments and suggestions for where to place greater emphasis in the text. We also apologize for the inadvertent absence of appropriate citations to past work on the connection between actin organization and osmotic stress. While we acknowledge that important past work in this area has been performed, we believe the interrogation of this process using the *prf5* mutant provides important genetic insight into the factors that affect the timing and concentration dependency of the process, as the physiological impact of such reorganization as it relates to plasmolysis. We updated the text: Line 350-352.

“Osmotic stress promotes actin reorganization in leaf and differentiated root tissues (C. Wang et al. 2010; X. Wang et al. 2020), but the cellular consequences and molecular components controlling this process remain unknown.”

Comment 3.2: be more careful in negating essential differences between *Chlamydomonas* and multicellular land plants,

Response 3.2: We updated the text to introduce the differences between *Chlamydomonas* and land plants;

Introduction

Lines 79-92

“We sought to advance our understanding of osmotic signaling pathways in photosynthetic organisms by establishing the unicellular alga *Chlamydomonas* (*Chlamydomonas reinhardtii*) as

a system that allows for rapid genetic and molecular investigation of the process. During the dominant phase of the lifecycle, *Chlamydomonas* exists as a haploid single cells, which facilitates the characterization of genetic mutants (Merchant et al. 2007). Recent large-scale characterization of mutant collections for growth defects under diverse stresses have demonstrated the importance of genes with homologues in animal and land-plant lineages (Fauser et al. 2022). Being a freshwater alga, structural and physiological differences with land plants also exist, such as the lack of a rigid cell wall to counteract cellular turgor generated by intracellular solutes (Wolf 2022), or the presence of an ancient contractile vacuole used to expel excess water out of the cell (Komsic-Buchmann, Wöstehoff, and Becker 2014). Furthermore, while initial responses to osmotic stress in land plants involve Ca^{2+} -mediated signaling (Feng et al. 2018), *Chlamydomonas* lacks several Ca^{2+} channels present in algae and animal cells such as Voltage-dependent Ca^{2+} channels (VDCCs), the transient receptor potential (TRP) channels and the inositol triphosphate receptor (Ip3R), which may be important in mediating physiological responses to the environment (Wheeler and Brownlee 2008; Verret et al. 2010; Edel and Kudla 2015).”

Results

Lines 252-262

“Eukaryotic cells use two systems to actively extrude intracellular toxic sodium: Cation/ H^+ antiporters (Masrati et al. 2018) (CPAs) and Na^+ -P-ATPases (Kumari and Rathore 2020). Our screens uncovered sodium chloride sensitivity to; (i) a conserved CPA transporter SODIUM/HYDROGEN EXCHANGER (NHX, Cre01.g034150) previously describe in plants (Yokoi et al. 2002) and fungi (Nass and Rao 1998) to play a role sodium toxicity; and (ii) SODIUM/POTASSIUM-EXCHANGING ATPase (CrNAK1) (Cre06.g263950) (C. N. S. Pedersen et al. 2012), a P2C Na^+/K^+ -ATPase lost at the base of the land plant lineage. Only two chlorophytes retain NAK1 transporters, a marine algae (*Ostreococcus tauri*) and *Chlamydomonas* (fresh water), suggesting that the high sensitivity of land plants to Na^+ arose from the loss of an effective system to remove intracellular Na^+ (C. N. S. Pedersen et al. 2012). Our screens have osmolyte specificity as shown by the number of membrane transporters necessary for *Chlamydomonas* growth under NaCl, since these transporters were not identified in the other conditions screened.”

Comment 3.3: coming up with a working hypothesis what the actin remodelling actually means.

Response 3.3: In our study, we showed the conservation of actin remodeling upon osmotic stress across the green lineage. Our phenotypic characterization of the *prf* mutants together with the rapid remodeling of actin upon stress suggests that this remodeling is regulated by profilin. In addition, early responses to osmotic stress such as calcium signaling are impaired in the initial stages of the stress application (Figure 3). Altogether suggesting an early role for the actin cytoskeleton for proper acclimation to osmotic stress. Acclimation to osmotic stress required the coordination of a

myriad of molecular components, from plasma membrane-localized proteins ion transporters (F. Yuan et al. 2014; Hamilton et al. 2015),) to intracellular activities (Crowell et al. 2009), vacuole homeostasis (Radin et al. 2021; Kim and Bassham 2011), and subsequent changes in gene expression (Dinnyeny et al. 2008), among others. Actin is an intracellular protein present in all eukaryotic cells. The association of actin with the cell membranes and cell walls (Yu et al. 2018) provides an interface to rapidly sense extracellular osmolarity changes and coordinate intracellular events. The association with rigid cell walls and flexible membranes provides a “sensing platform” where cell wall anchored actin filaments are constantly monitoring the cell-wall cell membrane interface (Liu, Persson, and Zhang 2015). Actin interacts with membrane channels such as aquaporin, sodium, chloride or potassium channels (Sasaki, Yui, and Noda 2014). Interestingly PIEZO channels gating is mediated by actin tethering, suggesting the existence of a force-from-filaments mechanism enabling tunable mechanoresponses (J. Wang et al. 2022).

We update the manuscript to summarize our working model, lines 503-510.

“We show here that osmotic stress causes a PROFILIN-mediated actin reorganization into bundles and cables, which can have both a regulatory and/or mechanical role for cell survival. Disruption of the actin cytoskeleton directly impacts cellulose deposition (Sampathkumar et al. 2013), vacuole morphology (Scheuring et al. 2016), calcium signaling (Bascom, Winship, and Bezanilla 2018; Cárdenas et al. 2008), and ion channel activity (Sasaki, Yui, and Noda 2014; J. Wang et al. 2022). Moreover, the actin cytoskeleton establishes barriers that influence the diffusion of membrane proteins and lipids to regulate signal transduction (Ienne 2006, Kusumi 2005). Defects in actin organization, as revealed in our *prf5* mutant and treatment with an actin depolymerization drug LatB, led to the loss of cellular integrity, more severe plasmolysis and roots showed penetration defects upon germination on hard media, suggesting that impaired mechanical properties (Supplementary Fig. 14). We propose that the regulatory and mechanical impact of actin in response to osmotic stress can be either direct or indirect. On one hand, elevated osmolarity can directly influence actin filament formation or affect the activity and binding of actin-interacting proteins due to increased intracellular ion concentrations. On the other hand, osmotic stress can alter turgor pressure, subsequently modifying the mechanical properties of the cell wall and cell membrane, which in turn indirectly affects actin dynamics. Further investigation into cellular dynamics under osmotic stress, employing high temporal and spatial microscopy, will enhance our understanding of the role played by the actin cytoskeleton in response to osmotic stress.”

REFERENCES

- Abu-Ghosh, Said, David Iluz, Zvy Dubinsky, and Gad Miller. 2021. “Exogenous Abscisic Acid Confers Salinity Tolerance in *Chlamydomonas Reinhardtii* During Its Life Cycle.” *Journal of Phycology* 57 (4): 1323–34.
- Bascom, Carlisle S., Jr, Lawrence J. Winship, and Magdalena Bezanilla. 2018. “Simultaneous

- Imaging and Functional Studies Reveal a Tight Correlation between Calcium and Actin Networks.” *Proceedings of the National Academy of Sciences of the United States of America* 115 (12): E2869–78.
- Birrell, Geoff W., James A. Brown, H. Irene Wu, Guri Giaever, Angela M. Chu, Ronald W. Davis, and J. Martin Brown. 2002. “Transcriptional Response of *Saccharomyces Cerevisiae* to DNA-Damaging Agents Does Not Identify the Genes That Protect against These Agents.” *Proceedings of the National Academy of Sciences of the United States of America* 99 (13): 8778–83.
- Cárdenas, Luis, Alenka Lovy-Wheeler, Joseph G. Kunkel, and Peter K. Hepler. 2008. “Pollen Tube Growth Oscillations and Intracellular Calcium Levels Are Reversibly Modulated by Actin Polymerization.” *Plant Physiology* 146 (4): 1611–21.
- Carrasco Flores, David, Markus Fricke, Valentin Wesp, Daniel Desirò, Anja Kniewasser, Martin Hölzer, Manja Marz, and Maria Mittag. 2021. “A Marine Chlamydomonas Sp. Emerging as an Algal Model.” *Journal of Phycology* 57 (1): 54–69.
- Colina, Francisco, María Carbó, Mónica Meijón, María Jesús Cañal, and Luis Valledor. 2020. “Low UV-C Stress Modulates Chlamydomonas Reinhardtii Biomass Composition and Oxidative Stress Response through Proteomic and Metabolomic Changes Involving Novel Signalers and Effectors.” *Biotechnology for Biofuels* 13 (1): 110.
- Crowell, Elizabeth Faris, Volker Bischoff, Thierry Desprez, Aurélie Rolland, York-Dieter Stierhof, Karin Schumacher, Martine Gonneau, Herman Höfte, and Samantha Vernhettes. 2009. “Pausing of Golgi Bodies on Microtubules Regulates Secretion of Cellulose Synthase Complexes in Arabidopsis.” *The Plant Cell* 21 (4): 1141–54.
- Cutler, Sean R., Pedro L. Rodriguez, Ruth R. Finkelstein, and Suzanne R. Abrams. 2010. “Abscisic Acid: Emergence of a Core Signaling Network.” *Annual Review of Plant Biology* 61: 651–79.
- Dinnyeny, José R., Terri A. Long, Jean Y. Wang, Jee W. Jung, Daniel Mace, Solomon Pointer, Christa Barron, Siobhan M. Brady, John Schiefelbein, and Philip N. Benfey. 2008. “Cell Identity Mediates the Response of Arabidopsis Roots to Abiotic Stress.” *Science* 320 (5878): 942–45.
- Dreyer, Ingo, Frances C. Sussmilch, Kenji Fukushima, Gonzalo Riadi, Dirk Becker, Jörg Schultz, and Rainer Hedrich. 2021. “How to Grow a Tree: Plant Voltage-Dependent Cation Channels in the Spotlight of Evolution.” *Trends in Plant Science* 26 (1): 41–52.
- Duan, Lina, Daniela Dietrich, Chong Han Ng, Penny Mei Yeen Chan, Rishikesh Bhalerao, Malcolm J. Bennett, and José R. Dinnyeny. 2013. “Endodermal ABA Signaling Promotes Lateral Root Quiescence during Salt Stress in Arabidopsis Seedlings.” *The Plant Cell* 25 (1): 324–41.
- Edel, Kai H., and Jörg Kudla. 2015. “Increasing Complexity and Versatility: How the Calcium Signaling Toolkit Was Shaped during Plant Land Colonization.” *Cell Calcium* 57 (3): 231–46.
- Eisenach, Cornelia, Maria Papanatsiou, Ellin-Kristina Hillert, and Michael R. Blatt. 2014. “Clustering of the K⁺ Channel GORK of Arabidopsis Parallels Its Gating by Extracellular K⁺.” *The Plant Journal: For Cell and Molecular Biology* 78 (2): 203–14.
- Evans, Tyler G. 2015. “Considerations for the Use of Transcriptomics in Identifying the ‘Genes That Matter’ for Environmental Adaptation.” *The Journal of Experimental Biology* 218 (Pt 12): 1925–35.
- Fausser, Friedrich, Josep Vilarrasa-Blasi, Masayuki Onishi, Silvia Ramundo, Weronika Patena,

- Matthew Millican, Jacqueline Osaki, et al. 2022. "Systematic Characterization of Gene Function in the Photosynthetic Alga *Chlamydomonas Reinhardtii*." *Nature Genetics*, May. <https://doi.org/10.1038/s41588-022-01052-9>.
- Feng, Wei, Daniel Kita, Alexis Peaucelle, Heather N. Cartwright, Vinh Doan, Qiaohong Duan, Ming-Che Liu, et al. 2018. "The FERONIA Receptor Kinase Maintains Cell-Wall Integrity during Salt Stress through Ca²⁺ Signaling." *Current Biology: CB* 28 (5): 666–75.e5.
- Fujita, Yasunari, Miki Fujita, Kazuo Shinozaki, and Kazuko Yamaguchi-Shinozaki. 2011. "ABA-Mediated Transcriptional Regulation in Response to Osmotic Stress in Plants." *Journal of Plant Research* 124 (4): 509–25.
- Funk, Johanna, Felipe Merino, Larisa Venkova, Lina Heydenreich, Jan Kierfeld, Pablo Vargas, Stefan Raunser, Matthieu Piel, and Peter Bieling. 2019. "Profilin and Formin Constitute a Pacemaker System for Robust Actin Filament Growth." *eLife* 8 (October). <https://doi.org/10.7554/eLife.50963>.
- Geng, Yu, Rui Wu, Choon Wei Wee, Fei Xie, Xueliang Wei, Penny Mei Yeen Chan, Cliff Tham, Lina Duan, and José R. Dinneny. 2013. "A Spatio-Temporal Understanding of Growth Regulation during the Salt Stress Response in *Arabidopsis*." *The Plant Cell* 25 (6): 2132–54.
- Giaever, Guri, Angela M. Chu, Li Ni, Carla Connelly, Linda Riles, Steeve Véronneau, Sally Dow, et al. 2002. "Functional Profiling of the *Saccharomyces Cerevisiae* Genome." *Nature* 418 (6896): 387–91.
- Gomez-Osuna, Aitor, Victoria Calatrava, Aurora Galvan, Emilio Fernandez, and Angel Llamas. 2020. "Identification of the MAPK Cascade and Its Relationship with Nitrogen Metabolism in the Green Alga *Chlamydomonas Reinhardtii*." *International Journal of Molecular Sciences* 21 (10). <https://doi.org/10.3390/ijms21103417>.
- Gomez-Porras, Judith Lucia, Diego Mauricio Riaño-Pachón, Begoña Benito, Rosario Haro, Kamil Sklodowski, Alonso Rodríguez-Navarro, and Ingo Dreyer. 2012. "Phylogenetic Analysis of K(+) Transporters in Bryophytes, Lycophytes, and Flowering Plants Indicates a Specialization of Vascular Plants." *Frontiers in Plant Science* 3 (August): 167.
- González-Coronel, José Manuel, Gustavo Rodríguez-Alonso, and Ángel Arturo Guevara-García. 2021. "A Phylogenetic Study of the Members of the MAPK and MEK Families across Viridiplantae." *PloS One* 16 (4): e0250584.
- Gonzalez-Guzman, Miguel, Gaston A. Pizzio, Regina Antoni, Francisco Vera-Sirera, Ebe Merilo, George W. Bassel, Maria A. Fernández, et al. 2012. "Arabidopsis PYR/PYL/RCAR Receptors Play a Major Role in Quantitative Regulation of Stomatal Aperture and Transcriptional Response to Abscisic Acid." *The Plant Cell* 24 (6): 2483–96.
- Hamilton, Eric S., Gregory S. Jensen, Grigory Maksaev, Andrew Katims, Ashley M. Sherp, and Elizabeth S. Haswell. 2015. "Mechanosensitive Channel MSL8 Regulates Osmotic Forces during Pollen Hydration and Germination." *Science* 350 (6259): 438–41.
- Harris, Brogan J., C. Jill Harrison, Alistair M. Hetherington, and Tom A. Williams. 2020. "Phylogenomic Evidence for the Monophyly of Bryophytes and the Reductive Evolution of Stomata." *Current Biology: CB* 30 (11): 2001–12.e2.
- Hirasawa, T., Y. Nakakura, K. Yoshikawa, K. Ashitani, K. Nagahisa, C. Furusawa, Y. Katakura, H. Shimizu, and S. Shioya. 2006. "Comparative Analysis of Transcriptional Responses to Saline Stress in the Laboratory and Brewing Strains of *Saccharomyces Cerevisiae* with DNA Microarray." *Applied Microbiology and Biotechnology* 70 (3): 346–57.
- Hodges, Matthew E., Bill Wickstead, Keith Gull, and Jane A. Langdale. 2012. "The Evolution of

- Land Plant Cilia.” *The New Phytologist* 195 (3): 526–40.
- Isayenkov, Stanislav V., Siarhei A. Dabravolski, Ting Pan, and Sergey Shabala. 2020. “Phylogenetic Diversity and Physiological Roles of Plant Monovalent Cation/H⁺ Antiporters.” *Frontiers in Plant Science* 11 (October): 573564.
- Jonak, Claudia, László Okrész, László Bögre, and Heribert Hirt. 2002. “Complexity, Cross Talk and Integration of Plant MAP Kinase Signalling.” *Current Opinion in Plant Biology* 5 (5): 415–24.
- Jo, William J., Jeung Hyoun Kim, Eric Oh, Daniel Jaramillo, Patricia Holman, Alex V. Loguinov, Adam P. Arkin, Corey Nislow, Guri Giaever, and Chris D. Vulpe. 2009. “Novel Insights into Iron Metabolism by Integrating Deletome and Transcriptome Analysis in an Iron Deficiency Model of the Yeast *Saccharomyces Cerevisiae*.” *BMC Genomics* 10 (March): 130.
- Kanshin, Evgeny, Peter Kubiniok, Yogitha Thattikota, Damien D’Amours, and Pierre Thibault. 2015. “Phosphoproteome Dynamics of *Saccharomyces Cerevisiae* under Heat Shock and Cold Stress.” *Molecular Systems Biology* 11 (6): 813.
- Kim, Sang-Jin, and Diane C. Bassham. 2011. “TNO1 Is Involved in Salt Tolerance and Vacuolar Trafficking in *Arabidopsis*.” *Plant Physiology* 156 (2): 514–26.
- Komsic-Buchmann, Karin, Luisa Wösthoff, and Burkhard Becker. 2014. “The Contractile Vacuole as a Key Regulator of Cellular Water Flow in *Chlamydomonas Reinhardtii*.” *Eukaryotic Cell* 13 (11): 1421–30.
- Kovar, D. R., P. Yang, W. S. Sale, B. K. Drobak, and C. J. Staiger. 2001. “*Chlamydomonas Reinhardtii* Produces a Profilin with Unusual Biochemical Properties.” *Journal of Cell Science* 114 (Pt 23): 4293–4305.
- Kropat, Janette, Anne Hong-Hermesdorf, David Casero, Petr Ent, Madeli Castruita, Matteo Pellegrini, Sabeeha S. Merchant, and Davin Malasarn. 2011. “A Revised Mineral Nutrient Supplement Increases Biomass and Growth Rate in *Chlamydomonas Reinhardtii*.” *The Plant Journal: For Cell and Molecular Biology* 66 (5): 770–80.
- Kumari, Jyoti, and Mangal S. Rathore. 2020. “Na⁺/K⁺-ATPase a Primary Membrane Transporter: An Overview and Recent Advances with Special Reference to Algae.” *The Journal of Membrane Biology* 253 (3): 191–204.
- Lachapelle, Josianne, Graham Bell, and Nick Colegrave. 2015. “Experimental Adaptation to Marine Conditions by a Freshwater Alga.” *Evolution; International Journal of Organic Evolution* 69 (10): 2662–75.
- Lefebvre, P. A., S. A. Nordstrom, J. E. Moulder, and J. L. Rosenbaum. 1978. “Flagellar Elongation and Shortening in *Chlamydomonas*. IV. Effects of Flagellar Detachment, Regeneration, and Resorption on the Induction of Flagellar Protein Synthesis.” *The Journal of Cell Biology* 78 (1): 8–27.
- Liu, Zengyu, Staffan Persson, and Yi Zhang. 2015. “The Connection of Cytoskeletal Network with Plasma Membrane and the Cell Wall.” *Journal of Integrative Plant Biology* 57 (4): 330–40.
- Maathuis, Frans J. M. 2014. “Sodium in Plants: Perception, Signalling, and Regulation of Sodium Fluxes.” *Journal of Experimental Botany* 65 (3): 849–58.
- Mackinder, Luke C. M., Chris Chen, Ryan D. Leib, Weronika Patena, Sean R. Blum, Matthew Rodman, Silvia Ramundo, Christopher M. Adams, and Martin C. Jonikas. 2017. “A Spatial Interactome Reveals the Protein Organization of the Algal CO₂-Concentrating Mechanism.” *Cell* 171 (1): 133–47.e14.

- Masrati, Gal, Manish Dwivedi, Abraham Rimon, Yael Gluck-Margolin, Amit Kessel, Haim Ashkenazy, Itay Mayrose, Etana Padan, and Nir Ben-Tal. 2018. "Broad Phylogenetic Analysis of Cation/proton Antiporters Reveals Transport Determinants." *Nature Communications* 9 (1): 1–14.
- Melamed, Daniel, Lilach Pnueli, and Yoav Arava. 2008. "Yeast Translational Response to High Salinity: Global Analysis Reveals Regulation at Multiple Levels." *RNA* 14 (7): 1337–51.
- Merchant, Sabeeha S., Simon E. Prochnik, Olivier Vallon, Elizabeth H. Harris, Steven J. Karpowicz, George B. Witman, Astrid Terry, et al. 2007. "The Chlamydomonas Genome Reveals the Evolution of Key Animal and Plant Functions." *Science* 318 (5848): 245–50.
- Nass, R., and R. Rao. 1998. "Novel Localization of a Na⁺/H⁺ Exchanger in a Late Endosomal Compartment of Yeast. Implications for Vacuole Biogenesis." *The Journal of Biological Chemistry* 273 (33): 21054–60.
- Ostrander, D. B., J. A. Gorman, and G. M. Carman. 1995. "Regulation of Profilin Localization in *Saccharomyces Cerevisiae* by Phosphoinositide Metabolism." *The Journal of Biological Chemistry* 270 (45): 27045–50.
- Pandey, Dhananjay K., and Bhupendra Chaudhary. 2017. "Evolutionary Expansion and Structural Functionalism of the Ancient Family of Profilin Proteins." *Gene* 626 (August): 70–86.
- Pazour, Gregory J., Nathan Agrin, John Leszyk, and George B. Witman. 2005. "Proteomic Analysis of a Eukaryotic Cilium." *The Journal of Cell Biology* 170 (1): 103–13.
- Pedersen, Christian N. S., Kristian B. Axelsen, Jeffrey F. Harper, and Michael G. Palmgren. 2012. "Evolution of Plant P-Type ATPases." *Frontiers in Plant Science* 3 (February): 31.
- Pedersen, Jesper T., and Michael Palmgren. 2017. "Why Do Plants Lack Sodium Pumps and Would They Benefit from Having One?" *Functional Plant Biology: FPB* 44 (5): 473–79.
- Porta, Jason C., and Gloria E. O. Borgstahl. 2012. "Structural Basis for Profilin-Mediated Actin Nucleotide Exchange." *Journal of Molecular Biology* 418 (1-2): 103–16.
- Posas, F., J. R. Chambers, J. A. Heyman, J. P. Hoeffler, E. de Nadal, and J. Ariño. 2000. "The Transcriptional Response of Yeast to Saline Stress." *The Journal of Biological Chemistry* 275 (23): 17249–55.
- Radin, Ivan, Ryan A. Richardson, Joshua H. Coomey, Ethan R. Weiner, Carlisle S. Bascom, Ting Li, Magdalena Bezanilla, and Elizabeth S. Haswell. 2021. "Plant PIEZO Homologs Modulate Vacuole Morphology during Tip Growth." *Science* 373 (6554): 586–90.
- Sampathkumar, Arun, Ryan Gutierrez, Heather E. McFarlane, Martin Bringmann, Jelmer Lindeboom, Anne-Mie Emons, Lacey Samuels, Tijs Ketelaar, David W. Ehrhardt, and Staffan Persson. 2013. "Patterning and Lifetime of Plasma Membrane-Localized Cellulose Synthase Is Dependent on Actin Organization in Arabidopsis Interphase Cells." *Plant Physiology* 162 (2): 675–88.
- Sasaki, Sei, Naofumi Yui, and Yumi Noda. 2014. "Actin Directly Interacts with Different Membrane Channel Proteins and Influences Channel Activities: AQP2 as a Model." *Biochimica et Biophysica Acta (BBA) - Biomembranes* 1838 (2): 514–20.
- Scherens, Bart, and Andre Goffeau. 2004. "The Uses of Genome-Wide Yeast Mutant Collections." *Genome Biology* 5 (7): 229.
- Scheuring, David, Christian Löffke, Falco Krüger, Maike Kittelmann, Ahmed Eisa, Louise Hughes, Richard S. Smith, Chris Hawes, Karin Schumacher, and Jürgen Kleine-Vehn. 2016. "Actin-Dependent Vacuolar Occupancy of the Cell Determines Auxin-Induced Growth Repression." *Proceedings of the National Academy of Sciences of the United States*

- of America* 113 (2): 452–57.
- Solter, K. M., and A. Gibor. 1978. “The Relationship between Tonicity and Flagellar Length.” *Nature* 275 (5681): 651–52.
- Stecker, Kelly E., Benjamin B. Minkoff, and Michael R. Sussman. 2014. “Phosphoproteomic Analyses Reveal Early Signaling Events in the Osmotic Stress Response.” *Plant Physiology* 165 (3): 1171–87.
- Urzica, Eugen I., David Casero, Hiroaki Yamasaki, Scott I. Hsieh, Lital N. Adler, Steven J. Karpowicz, Crysten E. Blaby-Haas, et al. 2012. “Systems and Trans-System Level Analysis Identifies Conserved Iron Deficiency Responses in the Plant Lineage.” *The Plant Cell* 24 (10): 3921–48.
- Verret, Frédéric, Glen Wheeler, Alison R. Taylor, Garry Farnham, and Colin Brownlee. 2010. “Calcium Channels in Photosynthetic Eukaryotes: Implications for Evolution of Calcium-Based Signalling.” *The New Phytologist* 187 (1): 23–43.
- Vilarrasa-Blasi, Josep, Friedrich Fauser, Masayuki Onishi, Silvia Ramundo, Weronika Patena, Matthew Millican, Jacqueline Osaki, et al. 2020. “Systematic Characterization of Gene Function in a Photosynthetic Organism.” *Cold Spring Harbor Laboratory*. <https://doi.org/10.1101/2020.12.11.420950>.
- Wang, Chunyang, Yang Liu, Si-Shen Li, and Guan-Zhu Han. 2015. “Insights into the Origin and Evolution of the Plant Hormone Signaling Machinery.” *Plant Physiology* 167 (3): 872–86.
- Wang, C., L. Zhang, M. Yuan, Y. Ge, Y. Liu, J. Fan, Y. Ruan, Z. Cui, S. Tong, and S. Zhang. 2010. “The Microfilament Cytoskeleton Plays a Vital Role in Salt and Osmotic Stress Tolerance in Arabidopsis.” *Plant Biology* 12 (1): 70–78.
- Wang, Jing, Jinghui Jiang, Xuzhong Yang, Gewei Zhou, Li Wang, and Bailong Xiao. 2022. “Tethering Piezo Channels to the Actin Cytoskeleton for Mechanogating via the Cadherin- β -Catenin Mechanotransduction Complex.” *Cell Reports* 38 (6): 110342.
- Wang, Ning, Zhixin Qian, Manwei Luo, Shoujin Fan, Xuejie Zhang, and Luoyan Zhang. 2018. “Identification of Salt Stress Responding Genes Using Transcriptome Analysis in Green Alga *Chlamydomonas Reinhardtii*.” *International Journal of Molecular Sciences* 19 (11): 3359.
- Wang, Xianling, Shuangtian Bi, Lu Wang, Hongpeng Li, Bi-Ao Gao, Shanjin Huang, Xiaolu Qu, et al. 2020. “GLABRA2 Regulates Actin Bundling Protein VILLIN1 in Root Hair Growth in Response to Osmotic Stress.” *Plant Physiology* 184 (1): 176–93.
- Wheeler, Glen L., and Colin Brownlee. 2008. “Ca²⁺ Signalling in Plants and Green Algae-- Changing Channels.” *Trends in Plant Science* 13 (9): 506–14.
- Wolf, Sebastian. 2022. “Cell Wall Signaling in Plant Development and Defense.” *Annual Review of Plant Biology*, February. <https://doi.org/10.1146/annurev-arplant-102820-095312>.
- Xu, Feifei, Xiaoan Wu, Lin-Hua Jiang, Hucheng Zhao, and Junmin Pan. 2016. “An Organelle K⁺ Channel Is Required for Osmoregulation in *Chlamydomonas Reinhardtii*.” *Journal of Cell Science* 129 (15): 3008–14.
- Yokoi, Shuji, Francisco J. Quintero, Beatriz Cubero, Maria T. Ruiz, Ray A. Bressan, Paul M. Hasegawa, and Jose M. Pardo. 2002. “Differential Expression and Function of Arabidopsis Thaliana NHX Na/H Antiporters in the Salt Stress Response.” *The Plant Journal: For Cell and Molecular Biology* 30 (5): 529–39.
- Yoshida, Kenji, Eiko Igarashi, Eiko Wakatsuki, Kazuhisa Miyamoto, and Kazumasa Hirata. 2004. “Mitigation of Osmotic and Salt Stresses by Abscisic Acid through Reduction of Stress-Derived Oxidative Damage in *Chlamydomonas Reinhardtii*.” *Plant Science: An*

- International Journal of Experimental Plant Biology* 167 (6): 1335–41.
- Yoshida, Takuya, Junro Mogami, and Kazuko Yamaguchi-Shinozaki. 2014. “ABA-Dependent and ABA-Independent Signaling in Response to Osmotic Stress in Plants.” *Current Opinion in Plant Biology* 21 (October): 133–39.
- Yuan, Fang, Huimin Yang, Yan Xue, Dongdong Kong, Rui Ye, Chijun Li, Jingyuan Zhang, et al. 2014. “OSCA1 Mediates Osmotic-Stress-Evoked Ca²⁺ Increases Vital for Osmosensing in Arabidopsis.” *Nature* 514 (7522): 367–71.
- Yuan, Xiaowei, Shizhong Zhang, Meihong Sun, Shiyang Liu, Baoxiu Qi, and Xinzheng Li. 2013. “Putative DHHC-Cysteine-Rich Domain S-Acyltransferase in Plants.” *PLoS One* 8 (10): e75985.
- Yu, Qin, Jing-Jing Ren, Lan-Jing Kong, and Xiu-Ling Wang. 2018. “Actin Filaments Regulate the Adhesion between the Plasma Membrane and the Cell Wall of Tobacco Guard Cells.” *Protoplasma* 255 (1): 235–45.
- Zelm, Eva van, Yanxia Zhang, and Christa Testerink. 2020. “Salt Tolerance Mechanisms of Plants.” *Annual Review of Plant Biology* 71 (April): 403–33.
- Zhang, Luo-Yan, Zhao-Tian Xing, Li-Qian Chen, Xue-Jie Zhang, and Shou-Jin Fan. 2022. “Comprehensive Time-Course Transcriptome and Co-Expression Network Analyses Identify Salt Stress Responding Mechanisms in *Chlamydomonas Reinhardtii* Strain GY-D55.” *Frontiers in Plant Science* 13 (February): 828321.

REVIEWER COMMENTS

Reviewer #3 (Remarks to the Author):

this work is addressing a central ability of land plants - to cope with water-related stresses. The topic is becoming progressively accentuated by the need to adapt agriculture to climate change. The move, to use *Chlamydomonas* as model to screen for resilience factors and then to validate the findings in *Arabidopsis*, is well done. The central finding of profilin-dependent actin remodelling as crucial event deciding about adaptation or cell death, is interesting. The work has been conducted in a professional manner. While most studies of this type are restricted to the transcriptome (missing the numerous regulatory processes conveyed by phosphorylation), these authors also address the phosphoproteome. They arrive at an interesting player.

I have only one critical point - the explanation, why actin remodelling should decide on resilience remains too vague. There is a lot of knowledge on the signalling function of actin, for instance in programmed cell death or the interaction of organelles. Authors should spend more effort or come up with a more specific working model worth of their impressive effort in filtering out profilin as a player.

Reviewer #4 (Remarks to the Author):

The manuscript "Multi-omics analysis of green lineage osmotic stress pathways unveils crucial roles of different cellular compartments" by Vilarrasa-Blasi al. uses the model system *Chlamydomonas reinhardtii* to compare the osmotic stress response in single cell algae and land plants. The authors combine three different omics analyses: a time-course analysis of changes in the transcriptome upon NaCl and mannitol treatment, a study for changes in the phosphoproteome upon 5-minute treatment with NaCl and mannitol and a genome-wide *Chlamydomonas* mutant screen for phenotypic responses to NaCl, mannitol, PEG or hypo-osmotic stress treatment. With these datasets and a follow-up experiment using *Arabidopsis* mutants, the authors address the interesting question of conserved osmotic stress response in divergent evolutionary lineages. This is an impressive piece of work with a large amount of information contained here, which has arisen from a collection of well-performed experiments.

Major comments:

1) Could you work out more clearly what is your rationale for combining these three omics analyses? In the manuscript you write that the little overlap in potential hits from the transcriptome/phosphoproteome or transcriptome/functional genomics approach is expected and has also been observed in other studies. It was not entirely clear to me if you use the information from the omics analysis for your follow up experiments. How many of the respective hits from the

transcriptome, phosphoproteome and functional genomics approach were tested in the Arabidopsis mutant experiment?

2) For the phosphoproteome data it would be useful to add the phosphorylation Site information in your data analysis. In Figure 2, Supplemental Figure 4 and Supplemental table S4 you refer to phosphorylated proteins without any site information. However the localization of the protein phosphorylation is very important since different phosphorylation sites can have different functionalities within the same protein.

The homology comparison to Arabidopsis phosphorylated proteins should also take the site localization into account. Depending on the sequence conservation between the Chlamydomonas and Arabidopsis orthologue the phosphorylation site should at least be located in corresponding protein domains to propose conserved functionality.

3) The transcriptomic and proteomic raw data and database search files should be made available in a public repository.

Minor comments:

4) Line 135. You refer here to Fig 1.b, Supplementary Fig. 1 a. Should this not be Supplementary Fig. 2 a?

5) Supplemental Figure 3. What do the two bar plot columns represent? Please clearly label the bar plot or add a more comprehensive description in the figure legend text.

6) Line 176 &193. Reference to Fig. 1c. This should be Fig. 2.

7) Line 201. Reference to Fig. 1 B. Labeling in Supplemental table S4 –tab Figure 1b. The Reference should be Fig. 2.

8) Line 1019, Line 1027. The reference for proteomic sample preparation #103 and Census software #104 seem to be shifted. Reference #103: Posas et al., The transcriptional response of yeast to saline stress (2000). #104: Hirasawa et al. Comparative analysis of transcriptional responses in saline stress in the laboratory and brewing strains of *Saccharomyces cerevisiae* with DNA microarray (2006)

I assume this should be a) Minkoff et al., A pipeline for ¹⁵N metabolic labeling and phosphoproteome analysis in *Arabidopsis thaliana* (2014) – currently #105, and b) Parke et al., A quantitative analysis software tool for mass spectrometry-based proteomics (2008) – currently #106

9) Supplemental Figure 4. The color coding shows reciprocal ratios for ¹⁴N or ¹⁵N samples for quite some proteins. Could you double check that ratio calculation is always treatment vs control for both label-swap conditions? If this is the case, can you explain why you see such consistent different trends within the label replicates?

10) Supplemental table S2. Tab “Pie chart osmotic orthology”. Values for “Pie chart Fe-deficiency orthology” missing.

11) Supplemental table S3. Do you have an explanation for the high $\log(2)$ ratio (“Sam_Int”/”Ref_Int”; supplemental table S3) of CAS1_BOVIN and CAS2_BOVIN peptides in the phosphoproteome samples?

12) Supplemental table S3. “Sequence” column. Could you add a legend or explanation in the table description for symbol meanings (#,@,*)?

13) Supplemental table S6. Tab “Highconfidencehits_FDR<0.3_FD”. Remove internal comment in line 87/88. Tab “Highconfidence_geneannotation”. Is Column A meant as internal comment?

REVIEWER COMMENTS

Reviewer #3 (Remarks to the Author):

this work is addressing a central ability of land plants - to cope with water-related stresses. The topic is becoming progressively accentuated by the need to adapt agriculture to climate change. The move, to use *Chlamydomonas* as model to screen for resilience factors and then to validate the findings in *Arabidopsis*, is well done. The central finding of profilin-dependent actin remodelling as crucial event deciding about adaptation or cell death, is interesting. The work has been conducted in a professional manner. While most studies of this type are restricted to the transcriptome (missing the numerous regulatory processes conveyed by phosphorylation), these authors also address the phosphoproteome. They arrive at an interesting player.

I have only one critical point - the explanation, why actin remodelling should decide on resilience remains too vague. There is a lot of knowledge on the signalling function of actin, for instance in programmed cell death or the interaction of organelles. Authors should spend more effort to come up with a more specific working model worth of their impressive effort in filtering out profilin as a player.

Authors answer: We thank the reviewers for the supportive comments. We now include in the discussion section an extended version of the working model for how actin remodeling may relate to osmotic stress resilience, Lines 442-466.

Actin is a highly conserved protein found in eukaryotic cells, and its organization is crucial for various cellular processes including cellulose deposition⁸², vacuole morphology⁸³, calcium signaling^{84,85}, plant programmed cell death^{86,87} and ion channel activity^{88,89}. The actin cytoskeleton is reorganized upon osmotic stress in various organisms including yeast, plant and animal cells to maintain cell integrity and function and the observed dynamic reorganization led to the hypothesis that actin could function as a bona fide osmosensor⁹⁰.

Following sensing, the reorganization of actin in response to osmotic stress affects the cell through both direct and indirect regulatory and mechanical pathways. On one hand, elevated osmolarity can directly influence actin filament formation and affect the activity and binding of actin-interacting proteins due to increased intracellular ion concentrations⁹¹. The regulation of ion channel activity by actin interaction is exemplified by the tethering of the mechanosensitive calcium channel, Piezo, regulates its activity⁸⁹, and where disruption of the actin cytoskeleton impairs Piezo-mediated responses⁹². On the other hand, osmotic stress can alter turgor pressure, subsequently modifying the mechanical properties of the cell wall and cell membrane, which could

in turn indirectly affect actin dynamics. Osmotic-dependent actin reorganization could also impact organelle dynamics such as vesicle trafficking⁹³ and affect cellulose biosynthesis⁸², resulting in impaired cell wall rigidity.

We propose a model where plant cells use the actin cytoskeleton as a direct or indirect readout of their osmotic status and use the reorganization of the actin network to orchestrate changes in cellular organization that facilitate acclimation to osmotic stress. However, whether actin and/or actin interacting proteins directly react to the cellular biomechanical alterations caused by water loss, or indirectly through the secondary signals generated by water availability, is still unclear. Further investigation into mechanical and signaling functions of the actin cytoskeleton under osmotic stress will enhance our understanding of the function that actin reorganization has on the acclimation of cells to environmental stress.

Reviewer #4 (Remarks to the Author):

The manuscript “Multi-omics analysis of green lineage osmotic stress pathways unveils crucial roles of different cellular compartments” by Vilarrasa-Blasi al. uses the model system *Chlamydomonas reinhardtii* to compare the osmotic stress response in single cell algae and land plants. The authors combine three different omics analyses: a time-course analysis of changes in the transcriptome upon NaCl and mannitol treatment, a study for changes in the phosphoproteome upon 5-minute treatment with NaCl and mannitol and a genome-wide *Chlamydomonas* mutant screen for phenotypic responses to NaCl, mannitol, PEG or hypo-osmotic stress treatment. With these datasets and a follow-up experiment using *Arabidopsis* mutants, the authors address the interesting question of conserved osmotic stress response in divergent evolutionary lineages. This is an impressive piece of work with a large amount of information contained here, which has arisen from a collection of well-performed experiments.

Major comments:

1) Could you work out more clearly what is your rationale for combining these three omics analyses? In the manuscript you write that the little overlap in potential hits from the transcriptome/phosphoproteome or transcriptome/functional genomics approach is expected and has also been observed in other studies. It was not entirely clear to me if you use the information from the omics analysis for your follow up experiments. How many of the respective hits from the transcriptome, phosphoproteome and functional genomics approach were tested in the *Arabidopsis* mutant experiment?

Authors answer:

Each approach used in this study provided us with different layers of knowledge regarding the osmotic stress responses in *Chlamydomonas*.

- The transcriptome analysis allowed us to rule out the hormonal regulation that dominates the osmotic stress response in land plants. In addition, it allowed us to identify the timing of *Chlamydomonas* responses to osmotic stress, such as the regulation of flagella and the conserved responses between *Chlamydomonas* and higher plants or yeast.
- The phosphoproteomics analysis allowed us to identify the immediate targets of osmotic stress signaling in *Chlamydomonas* and discover the conservation of several important signaling components across the green lineage.
- The functional genomics screening set the basis for further analysis of mutants in *Arabidopsis*. We selected the *Arabidopsis* candidates based on the phenotype that they had in *Chlamydomonas*. The primary criteria to select candidates was the phenotype observed through the functional genomics analysis. When we compared our top hits in the functional genomics with hits in the transcriptomics and phosphoproteomics with hits in functional genomics we found a low overlap (Figure 3) lines 280-286.

To clarify that the hits tested in *Arabidopsis* come exclusively from the *Chlamydomonas* functional genomics analysis, we rephrase the results (line 303) as follows:

To understand how the candidate genes identified in *Chlamydomonas* may function in a multicellular context, we grew *Arabidopsis* seedlings with mutations in genes homologous to our hits from the functional genomics in *Chlamydomonas* (homology based on Reciprocal Best BLAST hit [RBH], see material and methods) and analyzed the ability of their roots to acclimate to osmotic stress induced by NaCl or mannitol containing media

2) For the phosphoproteome data it would be useful to add the phosphorylation Site information in your data analysis. In Figure 2, Supplemental Figure 4 and Supplemental table S4 you refer to phosphorylated proteins without any site information. However the localization of the protein phosphorylation is very important since different phosphorylation sites can have different functionalities within the same protein.

The homology comparison to *Arabidopsis* phosphorylated proteins should also take the site localization into account. Depending on the sequence conservation between the *Chlamydomonas* and *Arabidopsis* orthologue the phosphorylation site should at least be located in corresponding protein domains to propose conserved functionality.

We agree with the reviewer that the localization of the phosphorylation sites are important for interpreting the potential regulatory impact on the target protein. All the phosphoproteomic raw data can be found in Supplementary table S3, detailing all the phosphosites detected. For convenience, we now add an extra tab in the supplementary table 4 named “phosphorylation sites” where the phosphosites are displayed.

The divergence time between *Chlamydomonas* and *Arabidopsis* is estimated to be over 1 billion years ago (Merchant et al. 2007; Yoon et al. 2004). Given these large evolutionary distances, there is a substantial sequence divergence within protein families and large differences in gene family sizes (One Thousand Plant Transcriptomes Initiative 2019) making the establishment of orthology relationships challenging.

The significant evolutionary divergence between *Chlamydomonas* and *Arabidopsis* presents obstacles in the identification of homologous domains. This requires a thorough investigation, which includes predicting protein structure and identifying homologous protein domains based on similarity in protein folds. Nonetheless, such an undertaking goes beyond the scope of the present study.

When using BLAST searches, with the identified phosphopeptide sites as query, we were not able to establish a relationship between our identified orthologous proteins (based on RBH, e value $< E-10$). In addition when comparing the phosphorylated sites identified in *Chlamydomonas* and the orthologous proteins previously shown to be differentially phosphorylated in *Arabidopsis* upon osmotic stress, we did not find any similarity besides the nature of the phosphorylated residue itself. As an example, we identified the following phosphopeptide in *Chlamydomonas* NAP; KSAAAGGEMsVGEAQQRT, while (Wang et al. 2013) (Wang et al. 2013) identified EEVIEEVAsPK. In both phosphopeptides “s” represents the phosphorylated serine, which is the only residue conserved.

Thus, we did not infer any functional conservation of phosphorylation sites nor proposed conserved functionality based on our phosphoproteomic results.

To clarify how we evaluated homology we edited the text in line 182-183;

We found several proteins with homologs (homology based on Reciprocal Best BLAST hit [RBH], see material and methods), that when mutated in *Arabidopsis*, have osmotic sensitive phenotypes such as;

3) The transcriptomic and proteomic raw data and database search files should be made available in a public repository.

Authors answer:

We uploaded the transcriptomic and phosphoproteomic raw data to public repositories and added a Data availability section in the Materials and Methods section lines 1210-1217

Data availability

All data are available in the manuscript, in the Supplementary material or in the following databases; high-throughput sequencing data sets are available through the National Center for Biotechnology Information Sequence Read Archive (SRA), GSE260814. Phosphoproteomic data sets are available through the Center for Computational Mass Spectrometry, Mass Spectrometry Interactive Virtual Environment (MassIVE) (<https://massive.ucsd.edu/ProteoSAFe/static/massive.jsp>), dataset ID: MSV000094492.

Minor comments:

4) Line 135. You refer here to Fig 1.b, Supplementary Fig. 1 a. Should this not be Supplementary Fig. 2 a?

We have now corrected this error and now cite Supplementary figure 2a instead of 1 a.

5) Supplemental Figure 3. What do the two bar plot columns represent? Please clearly label the bar plot or add a more comprehensive description in the figure legend text.

We added the following text to the legend, line 714-716:

Bars plots represent the percentage of genes differentially expressed upon osmotic stress or iron deficiency in *Chlamydomonas* with orthologous genes in *Saccharomyces* (orange bars) or *Arabidopsis* (blue) responding to the same stress.

6) Line 176 &193. Reference to Fig. 1c. This should be Fig. 2.

We have corrected the text and now cite Fig 2.

7) Line 201. Reference to Fig. 1 B. Labeling in Supplemental table S4 –tab Figure 1b. The Reference should be Fig. 2.

We have changed the text and now cite figure 2.

8) Line 1019, Line 1027. The reference for proteomic sample preparation #103 and Census software #104 seem to be shifted. Reference #103: Posas et al., The transcriptional response of yeast to saline stress (2000). #104: Hirasawa et al. Comparative analysis of transcriptional responses in saline stress in the laboratory and brewing strains of *Saccharomyces cerevisiae* with DNA microarray (2006)

I assume this should be a) Minkoff et al., A pipeline for 15N metabolic labeling and phosphoproteome analysis in *Arabidopsis thaliana* (2014) – currently #105, and b) Parke et al., A quantitative analysis software tool for mass spectrometry-based proteomics (2008) – currently #106

We updated the references and double-check all the references in the text.

9) Supplementary Figure 4. The color coding shows reciprocal ratios for 14N or 15N samples for quite some proteins. Could you double check that ratio calculation is always treatment vs control for both label-swap conditions? If this is the case, can you explain why you see such consistent different trends within the label replicates?

We double-checked the values and the ratio calculation was performed correctly. We did find variation between replicates. The amount of variation is normal when dealing with this type of data. The variability comes from the cumulative variability of an in vivo biological system responding to a stimulus, protein extraction, isolation, and digestion, phosphoenrichment, and some amount of instrument variability run to run.

10) Supplemental table S2. Tab “Pie chart osmotic orthology”. Values for “Pie chart Fe-deficiency orthology” missing.

We now include the values of Fe-deficiency orthology.

11) Supplemental table S3. Do you have an explanation for the high log(2) ratio (“Sam_Int”/“Ref_Int”; supplemental table S3) of CAS1_BOVIN and CAS2_BOVIN peptides in the phosphoproteome samples?

The presence of CAS1_BOVIN and CAS2_BOVIN is due to the spiked in casein being natural abundance 14N-containing protein. We do not have 15N-containing casein standards, and

so, in the quantification, we basically see massive abundance enrichment for the casein, since the fold change calculations here are $^{14}\text{N}/^{15}\text{N}$. It is used as a control for the phosphopeptide enrichment protocol, that we see it lends confidence that our enrichment worked.

12) Supplemental table S3. "Sequence" column. Could you add a legend or explanation in the table description for symbol meanings (#,@,*)?

We now added a legend in an extra tab called "legend".

A '@' indicates phosphorylation,
a '*' indicates an oxidized methionine,
and a '#' indicates a deamidated glutamine or asparagine.

13) Supplemental table S6. Tab "Highconfidencehits_FDR<0.3_FD". Remove internal comment in line 87/88. Tab "Highconfidence_geneannotation". Is Column A meant as internal comment?

We removed the comment. The tab contains two tables; on the left the average p-values and on the right the p-values for 2 mutant alleles. On Column A at the top of the table can read "Adjusted p-values (average)" that refers to the table on the left. We now formatted to better distinguish between the two tables.

REFERENCES

Merchant, Sabeeha S., Simon E. Prochnik, Olivier Vallon, Elizabeth H. Harris, Steven J. Karpowicz, George B. Witman, Astrid Terry, et al. 2007. "The Chlamydomonas

- Genome Reveals the Evolution of Key Animal and Plant Functions." *Science* 318 (5848): 245–50.
- One Thousand Plant Transcriptomes Initiative. 2019. "One Thousand Plant Transcriptomes and the Phylogenomics of Green Plants." *Nature* 574 (7780): 679–85.
- Wang, Pengcheng, Liang Xue, Giorgia Batelli, Shinyoung Lee, Yueh-Ju Hou, Michael J. Van Oosten, Huiming Zhang, W. Andy Tao, and Jian-Kang Zhu. 2013. "Quantitative Phosphoproteomics Identifies SnRK2 Protein Kinase Substrates and Reveals the Effectors of Abscisic Acid Action." *Proceedings of the National Academy of Sciences of the United States of America* 110 (27): 11205–10.
- Yoon, Hwan Su, Jeremiah D. Hackett, Claudia Ciniglia, Gabriele Pinto, and Debashish Bhattacharya. 2004. "A Molecular Timeline for the Origin of Photosynthetic Eukaryotes." *Molecular Biology and Evolution* 21 (5): 809–18.

REVIEWERS' COMMENTS

Reviewer #3 (Remarks to the Author):

authors have addressed my point from the first version and provided background on the role of actin with some interesting thoughts about biomechanics. I have no further requests.

Reviewer #4 (Remarks to the Author):

I would like to thank the authors for their effort to address the raised points. Two items still require correction.

1) The authors state in the rebuttal letter that “for convenience we now added an extra tab in the supplementary table 4 named “phosphorylation sites” where the phosphosites are added”. However, in the uploaded file the stated tab which I assume is the additional tab “Sheet1” only contains a list of identifiers with the heading “Chlamydomonas ID/ Treatment” but no phosphorylation site information. If this has been a mistake, please upload the correct table S4 file.

2) The manuscript now contains a data availability section. I apologize if I overlooked this information, but could you please provide the reviewer login details for the Massive ID MSV000094492?

REVIEWERS' COMMENTS

Reviewer #3 (Remarks to the Author):

authors have addressed my point from the first version and provided background on the role of actin with some interesting thoughts about biomechanics. I have no further requests.

Response

We appreciate the reviewer's input and thank for helping to improve our manuscript.

Reviewer #4 (Remarks to the Author):

I would like to thank the authors for their effort to address the raised points. Two items still require correction.

We appreciate the reviewer's input and thank for helping to improve our manuscript.

1) The authors state in the rebuttal letter that "for convenience we now added an extra tab in the supplementary table 4 named "phosphorylation sites" where the phosphosites are added". However, in the uploaded file the stated tab which I assume is the additional tab "Sheet1" only contains a list of identifiers with the heading "Chlamydomonas ID/ Treatment" but no phosphorylation site information. If this has been a mistake, please upload the correct table S4 file.

Response

We include the correct sites identified and upload the correct table S4. We renamed the tab containing the information "Phospho-sites". In addition, information about all sites can be find in table S3, with a legend for the indentified sites in the "Legend" tab.

2) The manuscript now contains a data availability section. I apologize if I overlooked this information, but could you please provide the reviewer login details for the Massive ID MSV000094492

Response

We made the data publicaly available and update the data availability section.

Data availability

All data are available in the manuscript, in the Supplementary material or in the following databases; high-throughput sequencing data sets generated in this study are available through the National Center for Biotechnology Information Sequence Read Archive (SRA), GSE260814 [https://www.ncbi.nlm.nih.gov/geo/query/acc.cgi?acc=GSE260814]. Phosphoproteomic data sets generated in this study are available through the Center for Computational Mass

Spectrometry, Mass Spectrometry Interactive Virtual Environment (MassIVE) (<https://massive.ucsd.edu/ProteoSAFe/static/massive.jsp>), dataset ID; MSV000094492. [<https://massive.ucsd.edu/ProteoSAFe/dataset.jsp?task=3e301feb87904983b4ccd5d9e949350b>].